# Incorporating the image formation process into deep learning improves network performance

Yue Li [1], Yijun Su[2,3,12], Min Guo [2], Xiaofei Han[2], Jiamin Liu[3], Harshad D. Vishwasrao[3], Xuesong Li[2,12], Ryan Christensen[2,12], Titas Sengupta[4], Mark W. Moyle[5], Ivan Rey-Suarez[6], Jiji Chen [3], Arpita Upadhyaya[6,7], Ted B. Usdin[8], Daniel Alfonso Colón-Ramos [9,10], Huafeng Liu[1,11]✉, Yicong Wu [2]✉ and Hari Shroff[2,3,10,12]

We present Richardson–Lucy network (RLN), a fast and lightweight deep learning method for three-dimensional fluorescence microscopy deconvolution. RLN combines the traditional Richardson–Lucy iteration with a fully convolutional network structure, establishing a connection to the image formation process and thereby improving network performance. Containing only roughly 16,000 parameters, RLN enables four- to 50-fold faster processing than purely data-driven networks with many more parameters. By visual and quantitative analysis, we show that RLN provides better deconvolution, better generalizability and fewer artifacts than other networks, especially along the axial dimension. RLN outperforms classic Richardson–Lucy deconvolution on volumes contaminated with severe out of focus fluorescence or noise and provides four- to sixfold faster reconstructions of large, cleared-tissue datasets than classic multi-view pipelines. We demonstrate RLN's performance on cells, tissues and embryos imaged with widefield-, light-sheet-, confocal- and super-resolution microscopy.

All fluorescence images are contaminated by blurring and noise, but this degradation can be ameliorated with deconvolution[1–3]. For example, iterative Richardson–Lucy deconvolution (RLD)[4,5] is commonly used in fluorescence microscopy, and is appropriate if the dominant noise source is described by a Poisson distribution. Unfortunately, RLD is computationally taxing for three-dimensional (3D) and 3D timelapse (4D) data, particularly if complex regularization[6,7] or large numbers of iterations are applied. To address this challenge, we recently proposed RLD variants[8] that can accelerate deconvolution speed by at least tenfold by reducing the number of iterations. Deploying these methods requires careful parameter optimization to avoid introducing artifacts.

[1]State Key Laboratory of Modern Optical Instrumentation, College of Optical Science and Engineering, Zhejiang University, Hangzhou, Zhejiang, China. [2]Laboratory of High Resolution Optical Imaging, National Institute of Biomedical Imaging and Bioengineering, National Institutes of Health, Bethesda, MD, USA. [3]Advanced Imaging and Microscopy Resource, National Institutes of Health, Bethesda, MD, USA. [4]Lewis-Sigler Institute for Integrative Genomics, Princeton University, Princeton, NJ, USA. [5]Department of Biology, Brigham Young University-Idaho, Rexburg, ID, USA. [6]Institute for Physical Science and Technology, University of Maryland, College Park, MD, USA. [7]Department of Physics, University of Maryland, College Park, MD, USA. [8]Systems Neuroscience Imaging Resource, National Institute of Mental Health (NIMH), National Institutes of Health (NIH), Bethesda, MD, USA. [9]Wu Tsai Institute, Department of Neuroscience and Department of Cell Biology, Yale University School of Medicine, New Haven, CT, USA. [10]MBL Fellows Program, Marine Biological Laboratory, Woods Hole, MA, USA. [11]Intelligent Optical and Photonics Research Center, Jiaxing Research Institute, Zhejiang University, Jiaxing, Zhejiang, China. [12]Present address: Janelia Research Campus, Howard Hughes Medical Institute (HHMI), Ashburn, VA, USA. ✉e-mail: liuhf@zju.edu.cn; yicong.wu@nih.gov

Parameter tuning is usually experience-dependent and time-consuming, and would ideally be automated. Deep learning offers one route to automation, as neural networks can automatically learn the mapping between the input data and desired output, given ample training data. Many deep learning models now show excellent capability in super-resolution, denoising and deconvolution applications, including content-aware image restoration networks (CARE)[9] based on the U-net architecture[10], residual channel attention networks (RCAN)[11,12], DenseDeconNet (DDN)[8] and the light-field reconstruction networks LFMNet[13] and HyLFM[14]. Drawbacks of these methods include poor network interpretability and their data-driven nature. The latter indicates that the quantity and quality of training data can drastically affect network performance. Another concern with deep learning methods is generalizability, that is, whether a network trained on one type of data can be used to make predictions on another data type.

Combining the interpretability of traditional model-based algorithms and the powerful learning ability of deep neural networks is a promising approach for avoiding tedious parameter tuning on the one hand and poor generalizability on the other. Algorithm unrolling[15] provides such a framework, using neural network layers to represent each step in traditional iterative algorithms (for example, ADMM-net[16] or ISTA-net[17], Deep-URL[18] and USRNet[19]). Passing input data through the unrolled network is equivalent to executing the iterative algorithm a finite number of times.

Inspired by RLD and algorithm unrolling, we propose a 3D microscopy deconvolution method that combines the forward/backward projector structure in RL deconvolution and deep learning, that is, Richardson–Lucy Network (RLN). We benchmarked the deconvolution capability of RLN against traditional RLD and purely data-driven networks including CARE, RCAN and DDN. We found that RLN causes fewer artifacts than purely data-driven network structures, providing better deconvolution and generalization capability. RLN contains less than 1/60th the number of learning parameters than CARE and RCAN, enabling at least fourfold improvement in processing time. Finally, RLN provides better axial resolution than RLD, even in the low signal-to-noise (SNR) ratio regime and when RLN is trained on synthetic data. We demonstrate the power of RLN on simulated phantoms and diverse samples acquired with widefield-, light-sheet-, confocal- and super-resolution microscopy.

## Results
### RLD motivates a new network architecture
The update formula in RLD (Extended Data Fig. 1a and Methods) can be decomposed into four steps:

$$(1)\, FP = E_k * f;\ (2)\, DV = I/\mathrm{FP};\ (3)\, BP = DV * b;\ (4)\, update = E_k \times BP.$$

Here * denotes convolution operation, the forward projector $FP$ function is the convolution of the current object estimate $E_k$ with the forward projector $f$, $DV$ indicates the division of the raw image $I$ by $FP$, the backward projector $BP$ function is the convolution of $DV$ with the backward projector $b$, and the estimate is updated by multiplying $E_k$

with BP. With appropriate design of $f$ and $b$, the speed of deconvolution can be improved[8]. However, the need to define parameters manually and the challenge of defining a stopping criterion[8] remain problematic. Given the reliance of RLD and fully convolutional networks on the convolution operation, we wondered whether the latter might be used to find the proper convolution parameters, thereby solving the projector design problem automatically.

We introduced $FP$, $DV$ and $BP$ functions into a convolutional network, creating a new 3D microscopy deconvolution network, RLN. The RLN structure consists of three core components: down-scale estimation (H1), original-scale estimation (H2) and merging (H3) (Fig. 1a, Extended Data Fig. 1b and Methods). H1 and H2 explicitly follow the RL deconvolution update formula (Methods), and H3 merges H1 and H2 with convolutional layers, providing the final deconvolved output. To enhance network efficiency, we designed H1 with smaller feature maps and more convolutional layers for processing downsampled input, and H2 with larger feature maps and fewer convolutional layers to process the original-size input. With this combination, H1 increases the field of view (FOV) accessed by each convolutional kernel and H2 mitigates information loss due to downsampling in H1. By using a synthetic phantom object consisting of mixtures of dots, solid spheres and ellipsoidal surfaces (Supplementary Fig. 1), we confirmed that the combination of H1 and H2 outperforms H1 or H2 alone (Supplementary Fig. 2a). Although RLN is conceptually motivated by algorithm unrolling similar to Deep-URL[18] and USRNet[19], RLN's new design offers distinct advantages over these methods, including the absence of the need to specify iteration number and the ability to rapidly process 3D data (Supplementary Note 1).

To provide further insight into the connection between RLN and traditional deconvolution, we examined the intermediate and final outputs of RLN and RLD on different phantoms and samples (Supplementary Table 1 and Supplementary Note 2). We studied network performance in conventional tests (where training and test data correspond to the same type of sample) and generalization tests (where the extent to which a model trains on one type of data generalizes to another, Fig. 1a). First, we evaluated network output on synthetic phantom objects consisting of mixed structures (Extended Data Fig. 2, Supplementary Fig. 1 and Methods). Second, we evaluated the generalization performance of the mixed structure model when applied to a human brain phantom (Supplementary Fig. 3a–f and Methods). Third, we evaluated single-view volumes of green fluorescent protein (GFP) -labeled cell membranes in a *Caenorhabditis elegans* embryo, imaged with dual-view light-sheet microscopy (diSPIM[20], Fig. 1b) under conventional testing (that is, the model was trained on similar single-view embryo data, using dual-view joint deconvolved results as ground truth). As shown in the lateral and axial views in all these examples, the intermediate output produced by RLN maintains the structure of the input data and resembles the output of RLD. For example, in both RLD and RLN, the FP results are blurry, which is expected as this step mimics the blurring introduced by imaging. For the simulated brain phantoms,

**Fig. 1 | RLN schematic and performance comparison with CARE, RCAN and DDN. a**, Schematic design of RLN consisting of three parts: H1, H2, and H3. *FP1, DV1, BP1, FP2, DV2, BP2* in H1/H2 follow the RL deconvolution iterative formula (Methods). **b**, *C. elegans* embryos expressing GFP-membrane marker, imaged with diSPIM, showing raw single-view input (left column), intermediate outputs (middle columns) and result (right column) of RLN (bottom row) versus RLD (top row). RLN was trained with dual-view deconvolved ground truth (GT). RLN: *FP1, BP1, FP2, BP2* are the steps in H1/H2. RLD: *FP1/BP1* and *FP2/BP2* are the forward/backward projection at iterations 1 and 5, respectively. Similarities between the RLN and RLD intermediates highlight RLN interpretability. **c**, Parameter number and testing runtime for a roughly 200 MB dataset, comparing RLN, DDN, CARE and RCAN. RLN offers the fewest parameters and a runtime roughly fourfold faster than CARE and roughly 50-fold than RCAN. **d**, Simulated noiseless spherical phantoms in lateral and axial views, comparing raw input,

GT and generalization predictions from the networks derived from a training dataset with mixed structures, emphasizing the generalization capability of RLN. RLN provides a prediction closest to GT, especially in axial views, whereas CARE/RCAN/DDN showed distorted shape or information loss (red/yellow arrows). **e**, Simulated noiseless human brain results in axial view, comparing raw input, GT (noiseless and without blur) and RLN prediction. **f**, Higher magnification view of red region in **e**, showing that RLN provides better restoration than RLD/CARE/RCAN/DDN predictions. The predictions rely on the same models as used for **d**, and thus underscore the generalization capability of RLN. **g**, Quantitative analysis (mean ± standard deviation, $n = 131$ *zy* slices for the brain data and $n = 12$ volumes for the beads data) with SSIM and PSNR for the predictions in **d** and **e**, confirming that RLN offers the closest match to GT. Scale bars, **b** 10 µm, **d** 5 µm, **e** 50 pixels and **f** 6 pixels. Experiments repeated four times for **b**, 12 times for **d** and once for **e**, representative data from single experiment are shown.

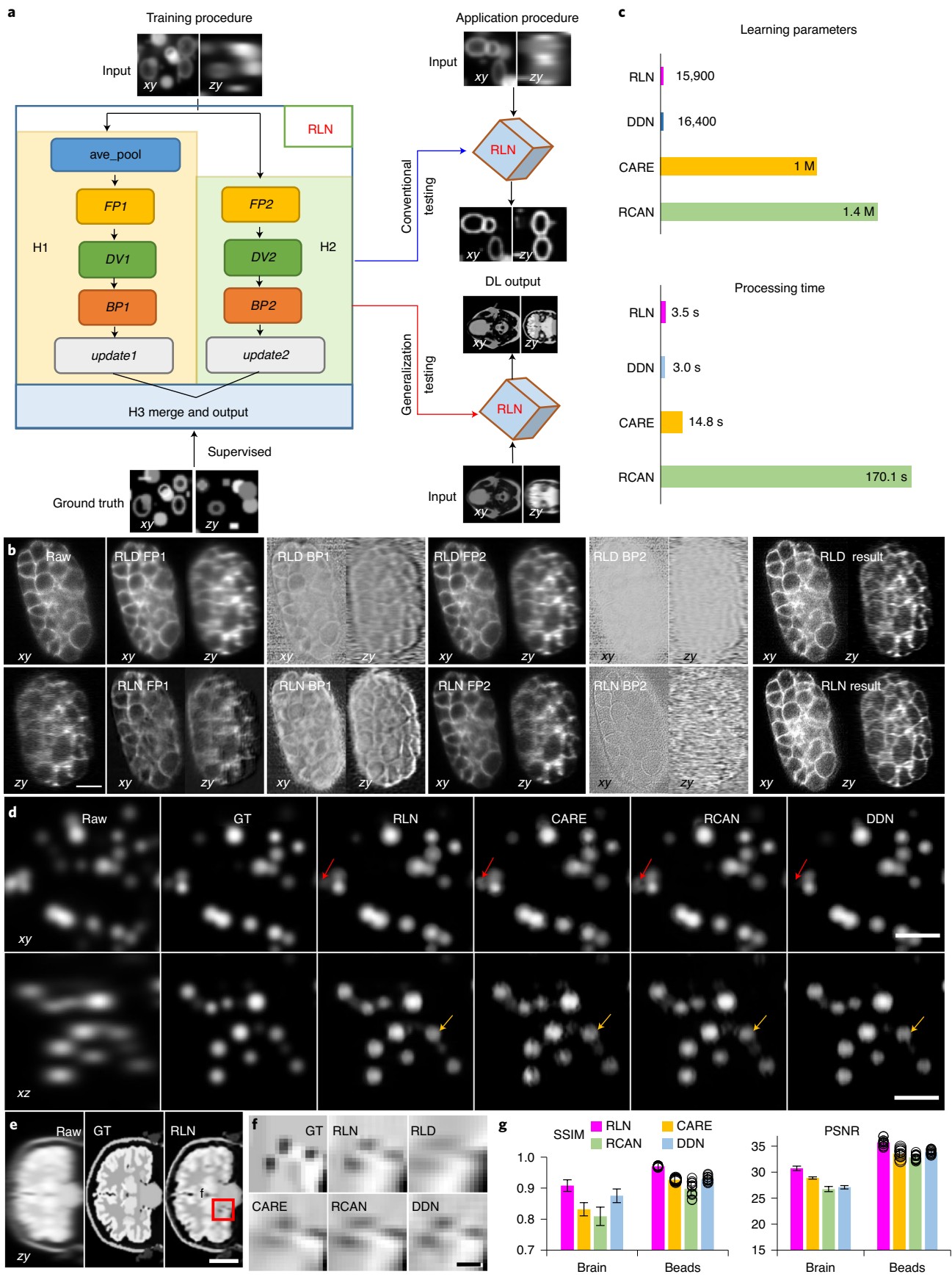

we found that the final RLN results (structural similarity index[21] (SSIM) 0.89, peak signal-to-noise ratio (PSNR) 24.4) outperformed RLD (SSIM 0.72, PSNR 16.9) (Supplementary Fig. 3), producing reconstructions closer to the ground truth. In all examples, we also noticed that RLN produced sharper axial views than RLD.

To study the effectiveness of the RL structure in RLN, we constructed an ablated version of RLN termed RLN-a (Extended Data Fig. 1c) by removing the DV and update steps. First, we compared intermediate and final outputs from RLD, RLN and RLN-a on a simulated bead dataset (Supplementary Fig. 2b–h), using the model trained on the synthetic mixed structures (Supplementary Fig. 1). The intermediate steps in RLN-a appeared visibly different from RLD and RLN (particularly the FP1 step), and the prediction from RLN-a (SSIM 0.94, PSNR 34.0) was noticeably further from the ground truth than RLN (SSIM 0.97, PSNR 35.7), suggesting that the DV and update steps in RLN are useful in network generalizability. Next, we compared the deconvolution ability of RLD, RLN and RLN-a in the presence of varying levels of Gaussian and Poisson noise by training models on mixed phantom structures with different input SNR levels (that is, one model for each SNR level, Supplementary Fig. 4). The performance of all methods degraded as the level of noise increased, although RLN performed better than RLN-a and both networks produced outputs visually and quantitatively closer to the ground truth than RLD at all noise levels. In summary, these results demonstrate the usefulness of the convolutional network structure in RLN-a and RLN in deconvolving noisy data, and that the additional structure in RLN further improves deconvolution output relative to RLN-a.

## Performance of RLN versus CARE, RCAN and DDN on simulated data

We compared the number of network parameters and the processing time of RLN with other state-of-the-art networks including CARE[9], RCAN[11] and DDN[8] (Fig. 1c). Both RLN and DDN are lightweight models, using less than 1/60th the number of learning parameters than CARE and RCAN. The time required to train an RLN model is comparable to CARE and DDN, but roughly three times faster than RCAN (for example, for 100 iterations with 64 × 64 × 64 voxels, RLN required 26.5 versus 29.2 s with CARE, 23.6 s with DDN and 90.9 s with RCAN). When applying the models to deconvolve sample volumes of roughly 200 MB size (1,920 × 1,550 × 20 voxels), RLN and DDN required roughly 3 versus 15 s with CARE and around 170 s with RCAN.

Next, we simulated noiseless spherical phantom datasets to examine (1) the difference in RLN reconstructions in conventional testing versus generalization applications (Supplementary Fig. 5) and (2) the output of RLN versus CARE, RCAN and DDN (Fig. 1d). Ground truth spherical structures were generated by ImgLib2 (ref. [22]) and blurred with a Gaussian kernel, and input data were generated by further blurring ground truth structures with the point spread function (PSF) of the 0.8/0.8 numerical aperture (NA) diSPIM (Methods). Generalization tests were conducted using the models trained from the synthetic mixed structures. RLN under both conventional testing

and generalization models recovered axial views distorted by the PSF and provided better linearity than RLD (Supplementary Fig. 5). Although the generalized RLN prediction was artificially sharpened compared to the conventional result, RLN still offered better generalization than CARE, RCAN and DDN, which all showed more obvious visual distortions (red and yellow arrows, Fig. 1d) and lower SSIM and PSNR (Fig. 1g).

Last, we applied the models trained from the synthetic mixed structures (Supplementary Fig. 1) to the synthetic human brain phantom (Fig. 1e and Supplementary Fig. 3g), which is visually very different from the structures in the training data. RLN and DDN generalization outputs more closely resembled the ground truth than RLD or CARE/RCAN generalization outputs (axial views shown in Fig. 1f, and lateral views shown in Supplementary Fig. 3g,h), a result consistent with SSIM and PSNR analysis (Fig. 1g and Supplementary Table 2).

## Class-leading performance of RLN on biological images

To demonstrate the deblurring capability of RLN for biological images, we used previously published images of live U2OS cells transfected with mEmerald-Tomm20, acquired with dual-view light-sheet fluorescence microscopy (0.8/0.8 NA diSPIM)[8,23,24]. Here we trained RLN, CARE, RCAN and DDN models to predict the dual-view, joint deconvolved results based on 12 randomly selected volumes from the time series. The training pairs consisted of raw single-view data paired with corresponding dual-view joint deconvolution ground truth (Supplementary Tables 3 and 4 provide additional information on the training parameters used in each model). RLN prediction showed clear improvements in resolution and contrast compared to the raw input (Fig. 2a), especially in the axial direction. Compared with other networks, mitochondrial details revealed by RLN were more similar to the ground truth in both lateral (Fig. 2b) and axial views (Fig. 2c), both visually and via quantitative assessment (Fig. 2f). We observed similar improvements when applying the RLN model to another live U2OS cell transfected with mEmerald-Tomm20, acquired with 0.8/0.8 NA diSPIM every 3 seconds, over 200 time points (Supplementary Video 1).

Next, we examined images of neurites in a slab of mouse brain, acquired by cleared-tissue diSPIM with 0.4/0.4 NA lenses[8]. The mouse brain sample was prepared using the iDISCO+ procedure[25], followed by immunolabeling with Alexa Fluor 555-conjugated secondary antibody against anti-tdTomato primary antibody. The entire brain volume after dual-view reconstruction spanned 10,280 × 5,160 × 1,400 voxels (corresponding to 4 × 2 × 0.5 mm³, 138.3 GB in 16-bit format)[26]. We trained on 12 randomly selected subvolumes of single-view data to predict the dual-view joint deconvolved results, each comprising 128 × 128 × 128 voxels, then applied the model to a held-out, larger scale subvolume from the same dataset spanning 1,184 × 1,184 × 1,218 voxels. RLN provided the best visual output of the neurites in both lateral and axial views (Fig. 2d), compared to RCAN, CARE and DDN (Fig. 2e), again confirmed quantitatively via PSNR and SSIM (Fig. 2f and Supplementary Table 2). In this example, we cropped the larger subvolume

**Fig. 2 | Deconvolution ability of RLN on thin or cleared biological samples. a**, Live U2OS cells transfected with mEmerald-Tomm20 were imaged with 0.8/0.8 NA diSPIM. Lateral maximum-intensity projections (MIP) of raw single-view and RLN prediction (conventional testing with single-view/joint deconvolution training pairs) are shown for a single time point. **b,c**, Higher magnification of solid line and rectangle in **a**, highlighting fine mitochondrial features (circular shape, red arrows; separated mitochondrial cross sections, yellow arrows) in lateral (**b**) and axial views (**c**), comparing raw single-view input, dual-view joint deconvolution ground truth (GT), predictions from RLN, CARE, RCAN and DDN. Visually, the RLN output most closely resembles GT. **d**, *xy* and *zy* MIP of sparsely labeled neurons in cleared brain tissue slab, acquired by 0.4/0.4 NA cleared-tissue diSPIM. Orange, raw data; green, RLN prediction. The inset shows the Fourier spectra of raw input and RLN output, indicating improvement in resolution after RLN. **e**, Higher magnification of yellow rectangle

in **d**, highlighting fine neurites, comparing raw, dual-view deconvolution GT and network predictions. RLN provides the most similar results to GT; other network outputs are blurrier (purple arrows) or lose information (blue arrows). **f**, Quantitative analysis (mean ± standard deviation) with SSIM and PSNR for 50 volumes from timelapse mitochondria data in **a** and 91 *xy* slices within the brain volume in **d**. **g**, Lateral and axial MIP from cleared brain tissue slab sparsely immunolabeled for axons, acquired with 0.7/0.7 NA cleared-tissue diSPIM and recovered with RLN. Images are depth coded as indicated. **h,i**, Higher magnification rectangular regions in **g**, comparing fine neurites in lateral (**h**) and axial (**i**) views with resolution estimates (Fourier spectra in the inserts). The RLN output closely resembles dual-view deconvolved GT with an SSIM of 0.97 ± 0.03, PSNR 49.7 ± 2.2 (*n* = 80 *xy* slices). Scale bars, **a,e** 10 μm, **b,c** 5 μm, **d** 100 μm, **g** 200 μm, **h,i** 50 μm. Experiments repeated four times for **a**, three times for **d** and twice for **g**, representative data from single experiment are shown.

into 25 batches, processed each batch with RLN and stitched the deep learning output to generate the final reconstruction (Methods). Cropping, RLN prediction and stitching took around 3 minutes. Scaling up

this RLN processing routine to the whole brain slab implies a time of roughly 2.2 hours with the RLN pipeline, a 5.5-fold speed up compared to the conventional processing pipeline described in our previous

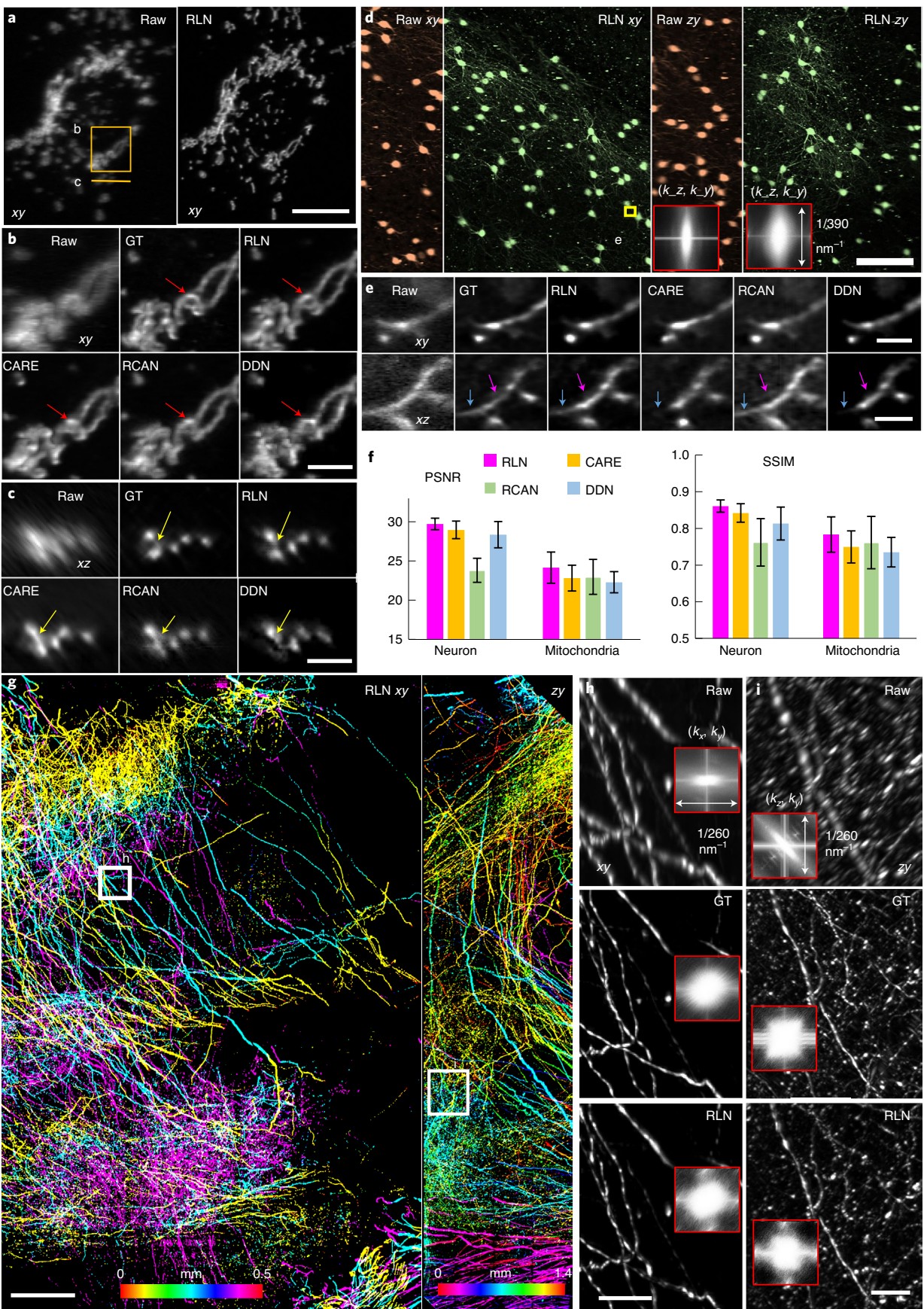

publication[8], which would otherwise take 12 hours including cropping, registration, joint RL deconvolution and stitching.

To further demonstrate that RLN can accelerate the restoration of large datasets, we imaged a large multi-tile image volume from another slab of cleared mouse brain, this time at higher resolution with a 0.7/0.7 NA cleared-tissue diSPIM (Methods). This brain expressed tdTomato in axonal projections from the site of a stereotactic adeno-associated virus injection. After fixing, clearing and sectioning the brain, we located and imaged a region with dense neurite labeling. The size of the brain volume after dual-view reconstruction spanned 5,432 × 8,816 × 1,886 voxels (roughly 1.4 × 2.3 × 0.5 mm$^3$, 168.2 GB in 16-bit format). From this dataset we randomly selected 40 subvolumes, each 256 × 256 × 256 voxels, pairing single-views (input) with dual-view joint deconvolutions (ground truth) to train the network, then applied the trained model to the entire dataset (Fig. 2g and Supplementary Video 2). Cropping the entire volume into 900 subvolumes (each 1,500 × 1,500 × 42 voxels), performing the RLN prediction on each subvolume, and stitching the results back together took roughly 2.7 hours on a single workstation equipped with a consumer-grade graphical processing unit (GPU) card. Compared to the single-view raw data, the RLN prediction displayed improved image resolution and contrast, closely resembling the joint deconvolution ground truth in both lateral (Fig. 2h) and axial (Fig. 2i) views (SSIM of 0.97 ± 0.03, $n = 80$ $xy$ planes). By contrast, it took approximately 11.5 hours to run the registration, joint deconvolution and stitching with the same GPU card as used above for applying RLN, or 3.5 hours on a cluster (Methods). We further tested the speed up of RLN processing time with different sizes of data (roughly 3–300 GB), confirming that RLN provides a 4–6-fold speed improvement (Extended Data Fig. 3a) over the previous processing pipeline[8] (that is, coarse registration, cropping, fine registration, joint deconvolution with a Wiener–Butterworth back projector and stitching) for cleared-tissue diSPIM data restoration. We also noted that the registration necessary for dual-view fusion failed on a small number of subvolumes in this example (for example, those with sparse signal), causing artifacts in the joint deconvolution result. As RLN is applied only on single-view input, it completely avoids errors of this kind (Extended Data Fig. 3b,c). Finally, we verified that RLN predictions displayed good performance over the entire volume (less than 10% variation: 688 ± 51 nm laterally, 701 ± 60 nm axially, assassed with decorrelation analysis[27] over $n = 300$ subvolumes, Extended Data Fig. 3d).

## RLN for volumes with scattering and spatially varying blur

Although the RLN prediction with single-view input (in the relatively thin or transparent samples studied thus far) closely resembled the dual-view ground truth, fine axial detail present in the ground truth was not fully recovered (for example, compare ground truth to RLN predictions in axial views, Fig. 2c,e,i). To address this issue and generate more nearly isotropic reconstructions, particularly in the context of densely labeled and scattering samples where additional views provide critical information lacking in any single view, we developed a dual-input implementation of RLN (Methods and Supplementary Fig. 6). We then tested this modified RLN with two inputs corresponding to the two raw registered views acquired with diSPIM. After training with 12 registered dual-view volumes acquired by imaging living GFP-histone-labeled *C. elegans* embryos with a 0.8/0.8 NA diSPIM, the dual-input RLN model produced better reconstructions than the single-view RLN model (Extended Data Fig. 4a,b,f and Supplementary Video 3). We observed similar improvements on embryos labeled with GFP-membrane markers (Extended Data Fig. 4c–f).

Next, we applied single- and dual-input RLN to the challenging case of images contaminated by a spatially varying blurring function. Reflective diSPIM[28] images samples deposited on reflective coverslips, enabling the collection of additional specimen views that boost collection efficiency and spatiotemporal resolution compared to conventional diSPIM. The raw reflective data are contaminated by substantial epifluorescence that varies over the imaging field. This contamination can be removed by incorporating a spatially varying PSF into RLD, at the cost of considerable computational expense. To train RLN for this application, we used a published reflective diSPIM dataset imaging muscle GCaMP3 expression in late stage *C. elegans* embryos (26 image pairs, raw specimen views as input data, deconvolution with a spatially varying Wiener–Butterworth back projector[8] as the ground truth). We found that single-input RLN handles this complex deconvolution task well, and that dual-input RLN provides even better deblurring quality on par with the ground truth (Extended Data Fig. 4g).

## RLN generalizes well on biological data

Having demonstrated RLN generalizability on simulated data (Fig. 1d,e and Supplementary Fig. 3), we next turned to biological data. We found that training models on simulated mixed structures that were blurred with the diSPIM PSF (Methods) generalized well when applied to *C. elegans* embryos labeled with nuclear and membrane markers and imaged with diSPIM (Extended Data Fig. 5). Although the generalization result was slightly inferior to the conventional test result (that is, training directly on diSPIM data), it still compared favorably against single-view RLD.

Next, we examined RLN generalizability on super-resolution data. We began by imaging (1) mitochondria labeled by mEmerald-Tomm20-C-10 (Mito) and (2) the endoplasmic reticulum (ER) labeled by ERmoxGFP in live U2OS cells with iSIM[29], a rapid super-resolution microscopy technique. We trained two RLN models with Mito and ER training data, respectively, and performed cross-validation testing: (1) the Mito data were predicted with the model trained with Mito and with the model trained with ER (Fig. 3a,b); (2) the ER data were similarly predicted by the two models (Fig. 3c). Similar to the deconvolved ground truth, RLN enabled crisper visualization of Mito and ER compared to the raw input. The predictions from different RLN models were nearly identical, both with SSIM higher than 0.96 and PSNR higher than 39 dB (Fig. 3d). These results indicate that the RLN predictions based on super-resolution input do not rely exclusively on image content, indicating that gathering ground truth data on a single type of structure is likely sufficient to predict another type of structure. Further cross-testing experiments on lysosomal and Golgi markers support this claim (Extended Data Fig. 6).

We then compared the generalization ability of RLN to other deep learning models (CARE, RCAN and DDN) on biological data. First, we found that RLN provides better deconvolution on Mito data than DDN, CARE and RCAN (Extended Data Fig. 7a–c and Supplementary Table 2), when using models trained on ER. Second, we compared the output of RLN, CARE, RCAN and DDN models trained exclusively on the synthetic mixed structures (Supplementary Fig. 1, blurred with the iSIM PSF[11], and Methods). When applying such models to the ER and lysosome biological data, we again found that RLN gave superior results, showing fewer artifacts and more closely resembling the ground truth than other networks (Fig. 3e, Extended Data Fig. 7d–f and Supplementary Table 2).

## RLN outperforms RLD on volumes with severe background/noise

Deconvolving volumes that are badly contaminated with background or noise is challenging. To illustrate the potential of RLN to address the former, we examined multiple samples imaged with widefield microscopy. First, we evaluated the generalization ability of RLN trained on purely synthetic data (Supplementary Fig. 1) to widefield images of fixed U2OS cells stained with Alexa Fluor 568 Phalloidin, marking actin, and fixed COS-7 cells immunolabeled with a primary mouse anti-Nup clone Mab414 and goat-anti-mouse IgG secondary antibody conjugated with Star635P, marking nuclear pore complexes (NPCs). We then compared the RLN predictions to the widefield input data, RLD on the widefield input, the Leica Thunder computational clearing method (a state-of-the-art commercial deconvolution software package) and

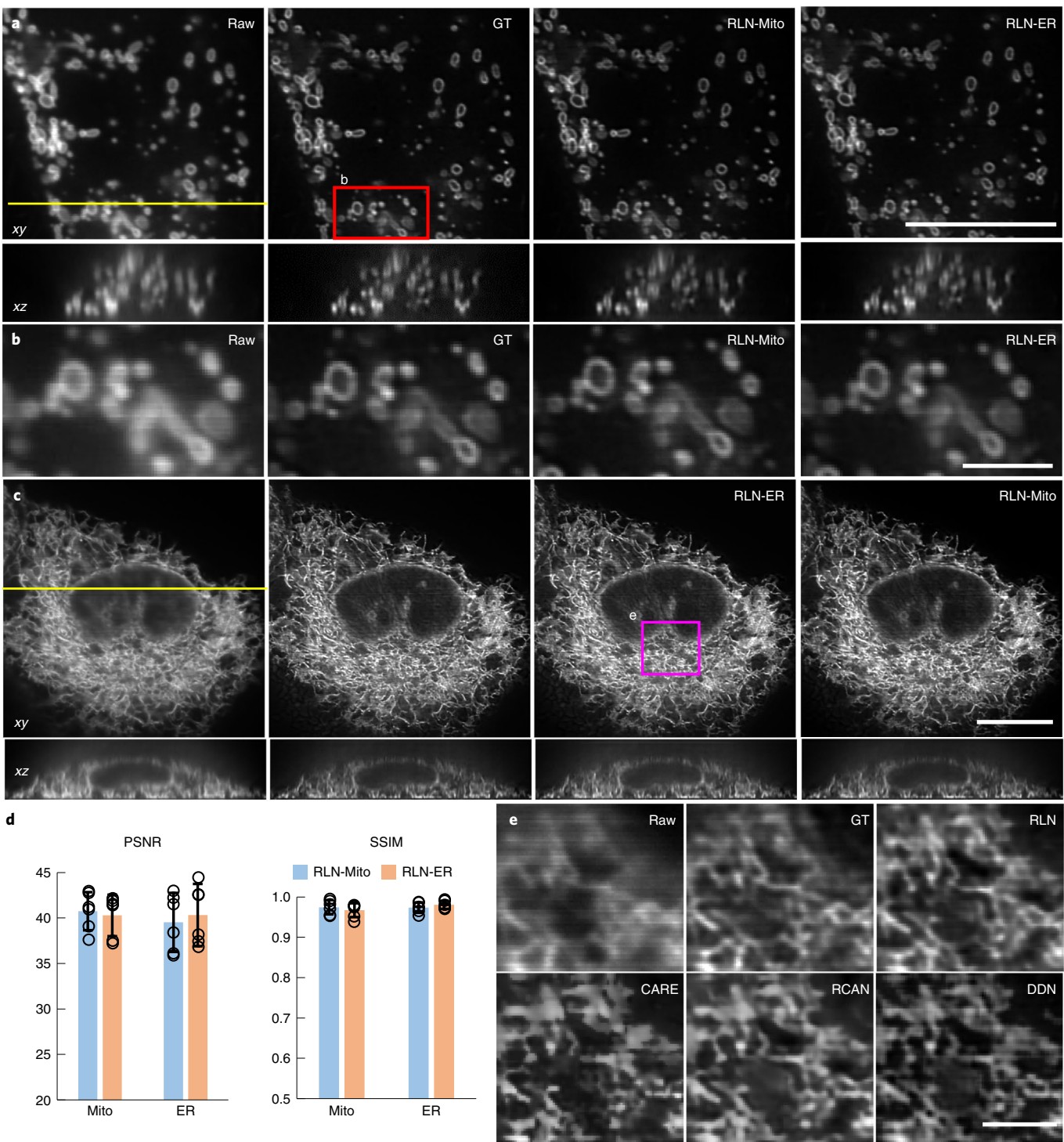

**Fig. 3 | RLN generalizes well on biological samples. a**, Lateral (top) and axial (bottom) views of live U2OS cells expressing mEmerald-Tomm20-C-10, acquired with iSIM, comparing the raw input (that is, without deconvolution), ground truth (the RL deconvolved result), the prediction from Mito-trained RLN (RLN-Mito) and the prediction from ER-trained RLN (RLN-ER). Axial views are taken across the yellow line. **b**, Higher magnification of red rectangle in **a**, further highlighting fine mitochondrial details. **c**, Lateral and axial views of live U2OS cells expressing ERmoxGFP, acquired with iSIM, comparing raw input, ground truth, predictions from Mito-trained RLN (RLN-Mito) and ER-trained RLN (RLN-ER). Axial views are taken across the yellow line. **d**, Quantitative analysis

with SSIM and PSNR for **a**–**c**, indicating the good generalizability of RLN. Means and standard deviations are obtained from *n* = 6 volumes (open circles indicate individual values) for Mito and ER. **e**, Magnified view of the magenta rectangle in **c**, comparing raw, RL deconvolved ground truth and predictions from RLN, CARE, RCAN and DDN. See Supplementary Table 2 for the quantitative SSIM and PSNR analysis of these network outputs. Note that for data shown in **e**, all network models were trained with a simulated phantom consisting of dots, solid spheres and ellipsoidal surfaces. Scale bars, **a**,**c** 10 μm, **b**,**e** 2 μm. Experiments repeated six times and representative data from single experiment are shown.

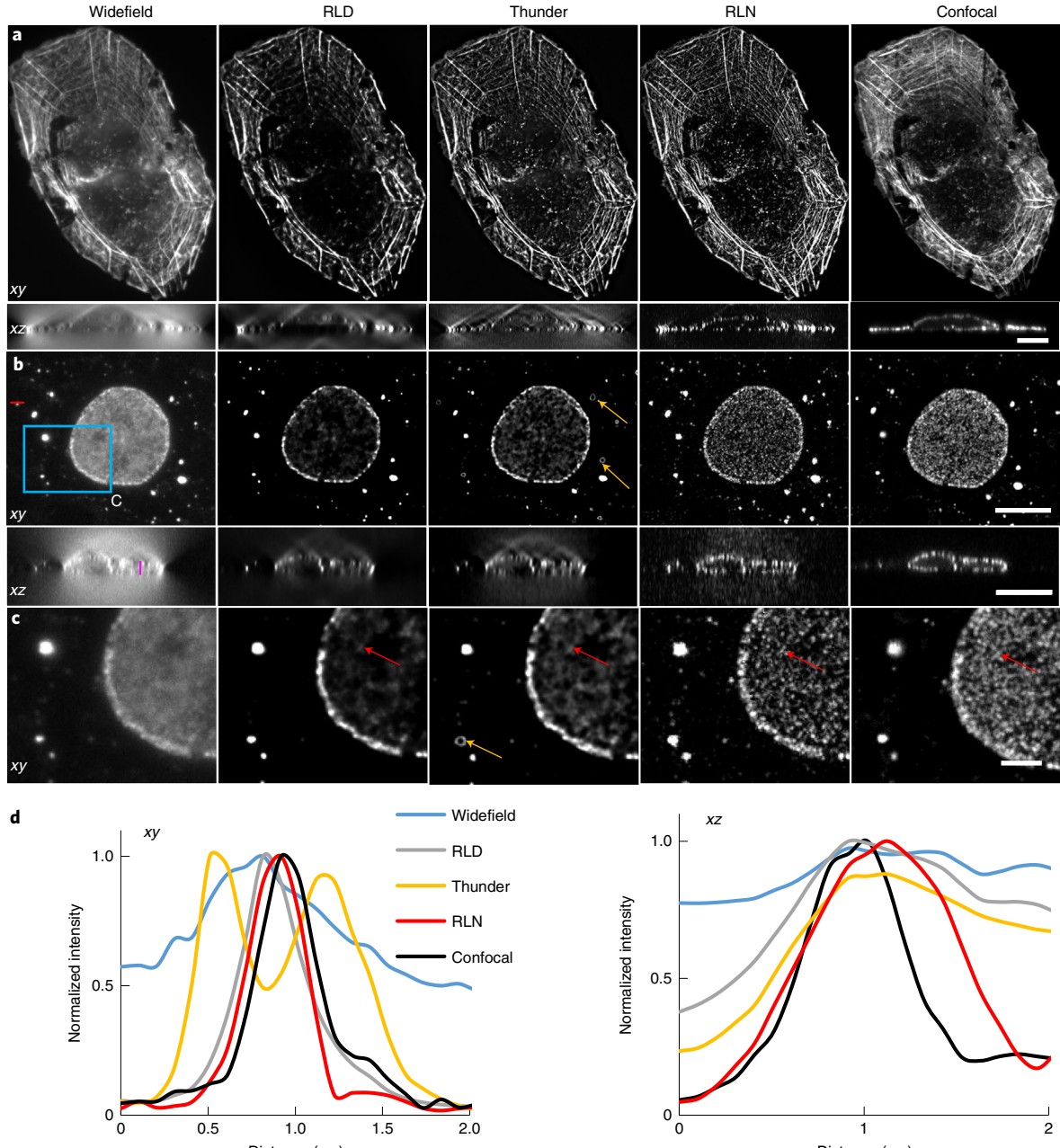

**Fig. 4 | Comparison between RLN, RLD and Leica Thunder computational clearing method. a**, Lateral and axial planes from images of a fixed U2OS cell stained with Alexa Fluor 568 Phalloidin, comparing widefield raw data, RLD with 100 iterations, Thunder output, RLN result and the registered confocal data as a ground truth reference. RLN was trained with synthetic mixed structures. RLN predictions show better restoration than RLD and Thunder, particularly along the axial dimension. PSNR and SSIM analysis using the confocal data as ground truth confirm this result (raw widefield, SSIM 0.57 ± 0.03; PSNR 28.7 ± 0.9; RLD, SSIM 0.67 ± 0.03, PSNR 30.0 ± 0.7; Thunder, SSIM 0.63 ± 0.05, PSNR 30.0 ± 0.7; RLN, SSIM 0.73 ± 0.02, PSNR 30.9 ± 0.9, n = 4 volumes). **b**, Lateral and axial planes of images of nuclei pore complexes in a fixed COS-7 cell immunolabeled with primary mouse anti-Nup clone Mab414 and goat-antimouse IgG secondary antibody conjugated with Star635P, comparing widefield input, RLD with 100

iterations, Thunder output, RLN prediction and registered confocal data as a ground truth reference. **c**, Magnified views of the blue rectangle in **b**. **d**, Line profiles across the red and magenta lines in the lateral and axial views in **b**. RLN was trained with synthetic mixed structures. Both visual analysis (for example, red arrows) and line intensity profiles demonstrate that RLN restoration outperform Thunder (obvious artifacts indicated by orange arrows) and RLD in both lateral and axial views, showing detail that approaches the confocal reference. PSNR and SSIM analysis using the registered confocal results as ground truth confirm this result (raw widefield input SSIM 0.78 ± 0.04, PSNR 34.5 ± 0.4; RLD SSIM 0.79 ± 0.02, PSNR 36.7 ± 0.5; Thunder SSIM 0.80 ± 0.04, PSNR 36.7 ± 0.4; RLN SSIM 0.86 ± 0.01, PSNR 37.5 ± 0.3, n = 4 volumes). Scale bars, **a** 10 μm; **b** 10 μm; **c** 3 μm. Experiments repeated four times for both **a** and **b**, representative data from single experiment are shown.

confocal images of the same structures to provide ground truth (Fig. 4). As shown, RLD, Thunder, and RLN all improve effective contrast and resolution when compared to the raw widefield input data. In some *xy* planes in the actin data (for example, Fig. 4a), RLD, Thunder and RLN provide visually similar output. However, RLD and Thunder both

produce obvious artifacts in the axial views (Fig. 4a) that are suppressed in RLN. Additionally, the Thunder imaging system also introduces artifacts in the NPC images (Fig. 4b–d) that are absent in RLN. The RLN output was visually closest to the confocal data, an impression confirmed with quantitative PSNR and SSIM analysis (Supplementary Table 2).

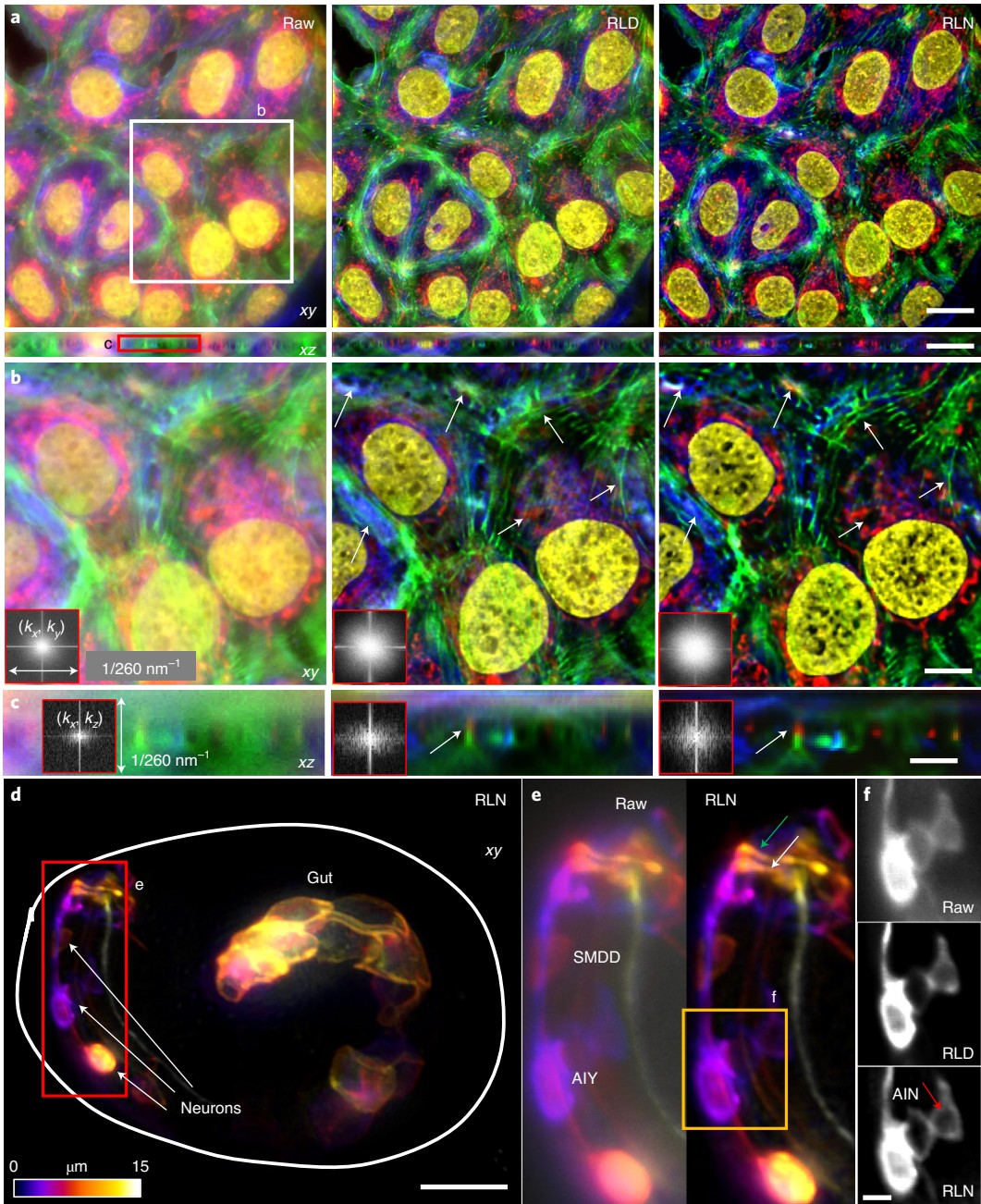

**Fig. 5 | RLN trained with synthetic mixed structures outperforms direct RLD on volumes contaminated by severe out-of-focus background. a**, Four-color lateral and axial maximum-intensity projections of a fixed U2OS cell, acquired by widefield microscopy, comparing the raw input, RLD and RLN prediction based on a model trained on synthetic mixed structures. Red, mitochondria immunolabeled with anti-Tomm20 primary antibody and donkey α-rabbit-Alexa-488 secondary; green, actin stained with phalloidin-Alexa Fluor 647; Blue, tubulin immunolabeled with mouse-α-tubulin primary and goat α-mouse-Alexa-568 secondary; yellow, nuclei stained with DAPI. **b,c**, Higher magnification views of white and red rectangular regions in **a** at a single slice, highlighting fine structures (white arrows) that are better resolved with RLN prediction than RLD in lateral (**b**) and axial views (**c**). Fourier spectra of the sum of all channels shown in the inserts also indicate that RLN recover resolution better than RLD.

**d**, Depth-coded image of a *C. elegans* embryo expressing ttx-3B-GFP, acquired by widefield microscopy, and predicted by RLN based on a model trained on the synthetic mixed structures. Solid line indicates the embryo boundary. **e**, Higher magnification of red rectangle in **d**, comparing the raw input and RLN prediction, showing neuronal cell bodies (AIY and SMDD) and neurites (the sublateral neuron bundle, green arrow; the amphid sensory neuron bundle, white arrow) are better resolved with RLN. **f**, Higher magnification of the orange rectangle in **e**, comparing the raw input, RLD and RLN predictions, highlighting AIY and AIN neurons. Red arrow highlights the interior of neuron, void of membrane signal and best resolved with RLN. Scale bars, **a** 20 µm, **b,d** 10 µm, **c,e** 5 µm, **f** 2 µm. Experiments repeated five times for both **a** and **d**, representative data from single experiment are shown.

Second, we used widefield microscopy to image fixed U2OS cells stained in four colors for actin, tubulin, mitochondria and nuclei (Methods and Fig. 5a). Then we trained an RLN model with synthetic mixed structures (Supplementary Fig. 1) and applied the model to these

widefield data. RLN outperformed RLD in lateral and axial views, sharpening nuclei, resolving more mitochondria, better separating actin and microtubule filaments, and recovering high spatial frequencies otherwise swamped by background (insets in Fig. 5b,c and Supplementary

Fig. 7). Third, we also applied the RLN model trained with synthetic data to a *C. elegans* embryo expressing ttx-3B-GFP, imaged with widefield microscopy (Methods and Fig. 5d). This marker labels neuronal membranes in the animal's head, and leaky expression from unc-54 3′ UTR likely labels membranes in gut cells. Compared to RLD, RLN better distinguished neuronal cell bodies and two functionally distinct nerve bundles: the sublateral and sensory neuron bundles within the nerve ring (main neuropil of *C. elegans*), which are challenging to distinguish due to the small size of the embryonic nerve ring (Fig. 5e,f). RLN also restored gut cell membranes better than RLD (Supplementary Fig. 8).

Another class of problematic samples for conventional RLD concerns those with poor SNR. RLN prediction is also influenced by the SNR of the input data. In addition to investigating the noise dependence of RLN on synthetic data (Supplementary Fig. 4 and Extended Data Fig. 8a), we also studied biological samples, imaging U2OS cells expressing ERmoxGFP with instant SIM[29]. When we trained an RLN model to deconvolve noisy data (input with an SNR of roughly five, ground truth with SNR of roughly 40), the prediction was visually improved compared to either the raw input or the RLD result, which were each dominated by noise (Extended Data Fig. 8b,c,e). Considering that using two or more networks sequentially can provide better restoration than a single network[11,30], we also performed a two-step deep learning strategy, first applying a denoising RCAN model to initially improve SNR, then applying a deconvolution RLN model for further improvement of contrast and resolution. In this two-step training scheme, the first step RCAN model was trained on pairs of low/high SNR raw data, whereas the second step RLN model was based on pairs of high SNR raw data and high SNR deconvolved data. We found the two-step prediction was noticeably closer to the ground truth and provided higher PSNR and SSIM than the single-step prediction with either RCAN or RLN model alone (Extended Data Fig. 8d–f).

## RLN's performance in super-resolution applications

Having demonstrated RLN's deconvolution capability using ground truth consisting of high SNR dual-view deconvolved light-sheet data, high SNR deconvolved iSIM data, high contrast confocal data and synthetic ground truth, we next evaluated the extent to which RLN can predict super-resolution images from diffraction-limited input. First, we evaluated confocal-to-stimulated emission depletion (STED) microscopy prediction[11,31], using 25 pairs of confocal/STED volumes of U2OS cells stained with a primary antibody against Tomm20 and an antirabbit secondary antibody conjugated with Alexa Fluor 594 (marking mitochondria) to train RLN and RCAN models (Extended Data Fig. 9). Although both networks improved the resolution of the confocal input, RLN quantitatively outperformed RCAN. Second, we performed conventional RLN testing based on Jurkat cells expressing EMTB-3XGFP (a microtubule marker) and generalization testing on U2OS cells expressing Lamp1-EGFP (a lysosomal marker). RLN models were trained with 46 EMTB-3XGFP volumetric datasets consisting of widefield input and 3D SIM ground truth. In both cases, output was visually and quantifiably closer to the 3D SIM ground truth than RLD (Extended Data Fig. 10 and Supplementary Table 2). Collectively, these results further demonstrate the power of RLN for additional applications on diverse microscopes. However, we also note that in all three examples, RLN did not restore input data to the extent of the ground truth. For example, dim microtubule filaments evident in the 3D SIM ground truth were not recovered using RLN (Extended Data Fig. 10a). This is unsurprising, given that the ground truth contains fine features difficult to recover given single diffraction-limited input images, and that RLN was designed primarily to deconvolve, not super-resolve, input data.

## Discussion

We designed RLN to mimic the forward/backward projector architecture of classic iterative deconvolution (Fig. 1a and Extended Data Fig. 1b), thereby improving network performance (Fig. 1b, Supplementary Figs. 3–5 and Supplementary Note 2). Distinct from previous methods based on algorithm unrolling[18,19], RLN enables 3D applications, spatially varying deconvolution and does not require an iteration number to be specified (Extended Data Fig. 4g and Supplementary Note 1). Since parameters are learned automatically, RLN has the potential to eliminate manual parameter selection in state-of-the-art deconvolution, as well as the burdensome and currently unsolved stopping criterion problem[8]. RLN is designed with around 16,000 parameters, roughly 60–90 fold fewer than purely data-driven network structures like RCAN and CARE (Fig. 1c). RLN also offers rapid runtime after training, more than fourfold faster than CARE, and almost 50-fold faster than 3D RCAN (Fig. 1c). With this advantage, it offers a 4–6-fold increase in speed (Extended Data Fig. 3a) compared to our previous processing pipeline for the reconstruction of large, cleared-tissue datasets with diSPIM[8] (Fig. 2d,e,g–i). Because the single-view RLN prediction showed improved resolution and contrast against the raw input, closely resembling the joint deconvolution ground truth (Fig. 2b,c,e,h,i), it can reduce the total amount of data required and bypass artifacts induced by poor registration (Extended Data Fig. 3b,c).

Compared with purely data-driven network structures, RLN shows better performance on both simulated data (Fig. 1d,e and Supplementary Fig. 3) and biological data derived from light-sheet (Fig. 2a–f) and super-resolution microscopes (Fig. 3e and Extended Data Fig. 7c,f). As expected, the deconvolution performance of RLN deteriorates in the presence of increasing noise, although RLN still outperforms RLD in the low-SNR regime (Extended Data Fig. 8e and Supplementary Fig. 4). Although it is not always possible to generate high SNR, high-quality deconvolved ground truth, the excellent generalization capability offered by RLN suggests that in these difficult cases it may be possible to use synthetic data (Supplementary Fig. 1) for training RLN and applying the model to the biological samples (Figs. 3e, 4 and 5a–c,e,f and Extended Data Figs. 5 and 7d–f). Further modification of the synthetic data would likely improve performance, perhaps by using a blurring kernel or noise level closer to the experimental test data or by incorporating more complex phantoms that better resemble real biological structures.

Like any denoising method, RLN's performance degrades if presented with ultralow SNR input data (Extended Data Fig. 8a,e), although a multistep network approach may help (Extended Data Fig. 8d,e). Also, although RLN can provide some resolution enhancement, it cannot restore fine detail to the extent present in the super-resolution ground truth (Extended Data Figs. 9 and 10).

We envision several extensions to our work. Since we have shown that we can successfully use synthetic data to train RLN, our method has the potential to aid in the deconvolution of any multi-view microscope system if the PSF can be defined (for example, as in our recently published multi-view confocal super-resolution microscopy method[30]). It would also be interesting to explore whether RLN trained on synthetic data blurred with a spatially varying PSF could generalize to real biological volumes that have been similarly contaminated[8,28]. Finally, RLN may offer speed or performance improvements over RLD and previous deep learning methods that have been used to reconstruct images acquired with light-field microscopy[14,32].

## Online content

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

## Methods

### RLN

RL deconvolution (equation (1)) has a compact update structure, needing only one formula to update each estimate:

$$
\begin{aligned}
E_0 &= I \\
&\text{for } k = 0, 1, \ldots N \,(\text{i.e. iteration number}) \\
E_{k+1} &= E_k \left\{ \frac{I}{E_k * f} * b \right\} \\
&\text{end}
\end{aligned} \tag{1}
$$

where * denotes convolution operation, $I$ and $E_k$ are the raw input and estimate of the $k$th iteration and $f$ and $b$ are the forward projector (system PSF) and backward projector, respectively. Traditionally $b$ is taken to be the transpose of $f$, but using unmatched back projectors (for example, Gaussian, Butterworth or Wiener–Butterworth filters)[8] can result in faster deconvolution by reducing the total number of iterations $N$ needed for achieving a resolution limited result.

The key procedure in RLD is convolution. Similarly, convolutional layers are integral to the architecture of deep learning networks, which can learn the convolution kernels automatically. This similarity inspired us to think of using convolutional layers to mimic the convolution with PSF kernels in RL deconvolution. RLN can be regarded as an algorithm unrolling method that uses convolutional layers in a fully convolutional network to represent the convolution steps in each RLD iteration, thereby mimicking the forward/back projection steps.

RLN consists of three parts: H1, H2, H3 (Fig. 1a and Extended Data Fig. 1b). H1 functions similarly to an early iteration in RLD, providing a rough estimate of the final output; H2 acts as a late iteration, using all the information in $I$ to refine the rough estimate and H3 is used to merge and integrate the information provided by H1 and H2. The architecture of H1 and H2 closely follow the RL deconvolution update formula, that is, they mimic the FP and BP steps with convolutional layers, additionally incorporating the division ($DV$, $\frac{I}{E_k * f}$), and update steps to learn the correction necessary for improving $I$. In RLD, $FP$ and $BP$ procedures use relatively large PSF kernels (for example, $128 \times 128 \times 128$ voxels for 0.8 NA/0.8 NA diSPIM). Applying such large kernels in a neural network would degrade training efficiency. Typically, deep learning networks use small convolutional kernels with several convolutional layers to extract features. For efficient operation, larger convolution kernels can be replaced by several smaller convolution kernels[33], for example a layer of $5 \times 5$ convolutions can be replaced by two layers each with $3 \times 3$ convolutions. To maintain network efficiency, H1 uses smaller feature maps and more layers, while H2 uses larger feature maps and less layers.

Because H1 only roughly estimates the ground truth, it starts with an average pooling layer to down-scale the input volume (that is, the normalized microscope acquisition $I$) by two in all dimensions to obtain $I_{ap}$ (average-pooled input). Although this step may cause information loss, it has the benefit of increasing the FOV, including more spatial information around each voxel, and decreasing computational cost. Following the RL iteration update process, $I_{ap}$ passes through three convolutional layers to construct the forward projection step. We use dense connections[34] among these convolution layers, that is, the outputs of the first two layers are concatenated along the channel direction to act as the input of the third layer, for efficient use of the feature maps. There is also a residual connection between the output feature maps of the third convolution layer and $I_{ap}$, and the result of this residual connection is denoted $FP1$. This residual connection has two functions: (1) the output of the forward projector $FP$ in RL deconvolution is a blurry copy of the current estimate, which approximates the microscope acquisition, and the residual connection acts similarly adding information learned by the network to the current estimate $I_{ap}$; and (2) it avoids the risk of dividing by zero in the following division step, which may introduce instability in training. All channels of the residual connection are merged by a channel-wise average (C_AVE)

producing $FP1$, and the quotient is computed as $DV1 = I_{ap}/FP1$. For the back projection step, RLN uses $DV1$ as the input to three densely connected convolutional layers to construct $BP1$. Because the final feature maps of H1 need to be restored to the original size, $BP1$ is up-scaled by a combination of an up-sampling layer and a convolutional layer to obtain the up-scaled $BP1$ ($BP1$up). All channels of $BP1$up are merged by a channel-wise average to obtain the correction, which is multiplied with $I$ to obtain the estimate $E1$.

H2 is constructed similarly to H1. The differences are that in the $FP$ and $BP$ steps, there are only two convolutional layers without dense connections; the input of H2 is the original-scale input $I$, that is, there is no upscaling procedure and the correction is applied to E1 to compute the second estimate $E2$. Since H1 already produces a rough estimate, H2 can use fewer parameters. We thus decreased the number of convolutional layers in H2 to improve memory efficiency.

H3 consists of three convolution layers and uses dense connections to merge and fine-tune $E1$ and $E2$. After the channel-wise average of the last layer's feature map, we obtain the final output $O$.

All convolution layers use $[3 \times 3 \times 3]$ kernels with $[1 \times 1 \times 1]$ strides, and are followed with batch normalizations[35] and softplus nonlinear activation functions[36]. The up-sampling consists of transpose convolution operations using $[2 \times 2 \times 2]$ kernels with $[2 \times 2 \times 2]$ stride, followed with batch normalizations and softplus. The softplus (SP) activation function is a smooth 'ReLU' function that ensures nonnegativity and avoids 'dead regions' where the gradient vanishes and parameters never update. For $DV1$ and $DV2$, we add a small constant $\alpha = 0.001$ in the denominator to prevent division by zero. In the unmatched forward/back projectors design[8], the choice of forward projector is set to the system PSF while the design of the back projector is more flexible, and should take noise amplification into account. Given that the design of the back projector is more complex, we set the number of output channels of the convolutional layers in the forward projector to four and in the backward projector to eight to place more weight on learning the back projectors. The total number of parameters in the RLN is roughly 16,000. For dual-view input, the dual-view information is registered with the ImageJ plugin diSPIM Fusion[8]. RLN merges these registered views by averaging before applying H1 (Supplementary Fig. 6).

To verify the effectiveness of the DV steps and update steps in RLN, we constructed an ablated version of RLN, named RLN-a (Extended Data Fig. 1c). RLN-a has the same convolutional layer design as the RLN but removes the $DV$ and update steps. It shares the same loss function and training parameters as RLN.

In the training procedure, the loss function is given by:

$$
L_{\text{total}} = L_{\text{aux}} + L_{\text{main}} \tag{2}
$$

where $L_{\text{aux}}$ is an auxiliary loss term used to guide H1 training, and $L_{\text{main}}$ is the main loss term used to guide training of the whole network.

As E1 is the rough estimate of the ground truth, it is expected to be sharper than the input volume I but blurrier than the ground truth GT. Thus, we define intermediate ITM as

$$
\text{ITM} = 0.8 \times \text{GT} + 0.2 \times I \tag{3}
$$

The ratio 0.8 versus 0.2 was empirically chosen, but we found that network output is only weakly dependent on this choice (for example, 0.9 versus 0.1 and 0.7 versus 0.3 work well also and are slightly better than 1.0 versus 0.0). $L_{\text{aux}}$ can be computed as the mean square error (m.s.e.) between E1 and ITM:

$$
L_{\text{aux}} = \text{m.s.e.}\,(\text{E1}, \text{ITM}) = \frac{1}{whd} \sum_{k=1}^{d} \sum_{j=1}^{w} \sum_{i=1}^{h} \left( \text{E1}\,(i,j,k) - \text{ITM}\,(i,j,k) \right)^2, \tag{4}
$$

where $d$, $w$, $h$ represents the depth, width and height of the ground truth, respectively.

$L_{main}$ includes two parts: the m.s.e. and SSIM[21] between the network output $O$ and GT:

$$L_{main} = \text{m.s.e.}(O, GT) - \ln((1 + SSIM(O, GT))/2) \qquad (5)$$

$$SSIM(O, GT) = \frac{(2\mu_O \mu_{GT} + C_1)(2\sigma_{O,GT} + C_2)}{(\mu_O^2 + \mu_{GT}^2 + C_1)(\sigma_O^2 + \sigma_{GT}^2 + C_2)}, \qquad (6)$$

where $\mu_{GT}$, $\mu_O$ are the mean values of the GT and $O$; $\sigma_{GT}^2$, $\sigma_O^2$ are the variances of the GT and $O$; $\sigma_{GT,O}$ is the covariance of GT and $O$; and $C_1$ and $C_2$ are small constants that prevent the denominator from becoming zero (here $C_1 = 1 \times 10^{-4}$ and $C_2 = 9 \times 10^{-4}$). A higher SSIM value means the network output is more similar to the ground truth. Because the SSIM value is smaller than 1, the $\ln(\cdot)$ operation is used to keep the loss positive. The m.s.e. term is similar to $L_{aux}$, ensuring that the difference between network outputs and ground truth is as small as possible, but using m.s.e. exclusively may lead to blurred output. SSIM is used to preserve the global structural similarity between $O$ and GT.

The solver method that is used to guide the parameter update is based on the 'adaptive moment estimation' (Adam) algorithm. The learning rate $r$ decays during the training procedure according to:

$$r = r_0 \times dr^{\frac{global\_step}{decay\_step}}, \qquad (7)$$

where $r_0$ is the start learning rate, dr is the decay rate, global_step represents the number of training iterations (updated after each iteration) and decay_step determines the decay period.

Gaussian filter kernels are used to initialize the convolutional layers in FP, which contain four output channels. Each channel is a Gaussian filter with standard deviation $\sigma = 0.5, 1, 1.5$ and 2, respectively:

$$\begin{aligned} & \text{Gaussian\_kernel}(i - c_i, j - c_j, k - c_k) \\ & = a \times \exp\left(-\frac{(i-c_i)^2 + (j-c_j)^2 + (k-c_k)^2}{2\sigma^2}\right), \end{aligned} \qquad (8)$$

where $(c_i, c_j, c_k)$ is the center coordinate of the kernel and $a$ is a random number to increase randomness (ranges from 0.5 to 1). Other kernels in the convolutional layers are randomly initialized with a Gaussian distribution (mean is 0, standard deviation is 1). Using our workstation (see below for details), training with 200 epochs usually takes 2–4 hours, with each epoch using 100 iterations.

Real microscopy volumes often exhibit isolated voxels with bright values that represent abnormal structures. Therefore, we adopted the percentile-based normalization as in CARE[9]:

$$N(u; p_{low}, p_{high}) = \frac{u - \text{percentile}(u, p_{low})}{\text{percentile}(u, p_{high}) - \text{percentile}(u, p_{low})}, \qquad (9)$$

where percentile($u, p$) is the $p$th percentile of all voxel values of data $u$. For real data, we set $p_{low} \in (0, 1)$ and $p_{high} \in (99.0, 100)$ according to the data quality. For simulated data, we set $p_{low} = 0$ and $p_{high} = 100$.

We adopted similar online data augmentation as used with 3D RCAN[11], which is a stochastic block selection process. For every training iteration, the batch size is set to four. The parameters of RLN and the size of selected blocks are summarized in Supplementary Table 3. For the comparison of RLN with RLD, we implemented both conventional RLD (Figs. 1b,f, 4 and 5, Extended Data Figs. 2, 5, 8 and 10 and Supplementary Figs. 2–5, 7 and 8) and RLD with an unmatched back projector (Fig. 2e,h and Extended Data Figs. 3b,c and 4g). Iteration numbers are included in Supplementary Table 3.

## RLN comparison with CARE, RCAN and DDN
We benchmarked the performance of RLN versus purely data-driven network structures including CARE, RCAN and DDN, which have demonstrated excellent performance in image restoration. The parameters used in training these neural networks are summarized in Supplementary Table 4.

The CARE implementation was downloaded from https://github.com/CSBDeep/CSBDeep and networks trained according to their instructions (http://csbdeep.bioimagecomputing.com/doc/). According to the default settings, the number of resolution levels of the U-net architecture was set to 2, each level in the down-scaling step and the upscaling step had two convolutional layers, the number of convolutional filters for first resolution level was set to 32 and the convolution kernel size was $(3 \times 3 \times 3)$. The total number of parameters is almost 1 million. During training, the training batch size was set to four.

For the studies using RCAN, we used our recently developed 3D RCAN model (https://github.com/AiviaCommunity/3D-RCAN), consisting of five residual groups with each residual group containing five residual channel attention blocks. As default, we used only two convolutional layers in each residual channel attention blocks. Since the convolution kernel size is $(3 \times 3 \times 3)$ and the convolution channel number is mostly set as 32, the total number of parameters is over 1 million.

For DDN, we used our published single-input neural network (https://github.com/eguomin/regDeconProject/tree/master/DeepLearning) based on three dense blocks. Here we improved the image preprocessing steps by adding online data augmentation and percentile-based normalization. During training, the training batch size was set to four.

## Training and testing
All networks (RLN, CARE, RCAN and DDN) were implemented with the Tensorflow framework v.1.14.0 and Python v.3.6.2 in the Ubuntu v.16.04.4 LTS operating system. Training and testing were performed on a computer workstation equipped with 32 GB of memory, an Intel(R) Core(TM) i7-8700K, 3.70 GHz CPU and two Nvidia GeForce GTX 1080 Ti GPUs, each with 24 GB of memory.

With this workstation, the maximum size of the input data that RLN can be applied to is 320 MB in 32-bit format. For input data sizes that exceed this limit (for example, the large cleared-tissue data shown in Fig. 2d,e,g–i), our Python-based processing code can automatically crop the volume into several subvolumes, feed them into the RLN network, and stitch the predictions back together. In detail, assuming a data with size $W \times H \times D$ voxels we first set the depth $d$ of the subvolume as:

$$d = \min(D, 1,600 \text{ pixels}), \qquad (10)$$

then calculate the width $w$ and height $h$ of the subvolume as:

$$w, h = \text{floor}\left(\text{sqrt}\left(\frac{\frac{320}{4} \times 1024 \times 1024}{d}\right)\right) \qquad (11)$$

We set the overlapping number voxels in neighboring subvolumes as 24 and use the linear_ramp function (NumPy function) to stitch the overlapped regions. This cropping and stitching procedure is the same as that used in 3D RCAN[11].

Since we did not have access to the true object structure when evaluating the performance of RLN, we used a variety of 'ground truth', consisting of high SNR dual-view deconvolved light-sheet data (Figs. 1b and 2 and Extended Data Figs. 3–5), synthetic ground truth (Fig. 1d,e, Extended Data Figs. 2 and 8a and Supplementary Figs. 2–5), high SNR deconvolved iSIM data (Fig. 3 and Extended Data Figs. 6, 7 and 8b), higher contrast confocal data (Fig. 4), super-resolution STED microscopy data (Extended Data Fig. 9) and super-resolution 3D SIM data (Extended Data Fig. 10). Further details (that is, training ground truth, training pair number, testing type) of training and test datasets are summarized in Supplementary Tables 1 and 3.

## Quantitative analysis

For all datasets, we selected several volumes or slices (4–131) to evaluate the SSIM and PSNR on normalized network outputs and ground truths with MATLAB (Mathworks. R2019b), and then computed the mean value and standard deviation of these volumes. Supplementary Table 2 summarizes these values.

The SNR of simulated noisy phantoms (represented as noiseless signal $S$+ different levels of noise $N_a$, Extended Data Fig. 8a and Supplementary Fig. 6) were computed as:

$$SNR_{simu} = 10 \times \log_{10} \frac{Var(S)}{Var(N_a)} \qquad (12)$$

Var $(.)$ was used to compute the variance of the volumes. The estimation of SNR of iSIM data (Extended Data Fig. 8b–e) is the same as used in our earlier 3D RCAN work[11]:

$$SNR_{iSIM} = S_{iSIM}/\sqrt{S_{iSIM} + N_r^2}, \qquad (13)$$

where $S_{iSIM}$ is the observed, background-corrected signal in photoelectrons (0.46 photoelectrons per digital count) and $N_r$ is the read noise (1.3 electrons according to the manufacturer).

Lateral and axial resolution estimates for the data shown in Extended Data Fig. 3d were based on decorrelation analysis[27] of subvolume $xz$ maximum-intensity projections (mean and standard deviations derived from 300 subvolumes, each $600 \times 600 \times 600$ voxels), using a sectorial mask to capture spatial frequencies predominantly along the $x$ and $z$ dimensions, respectively.

## Sample preparation

Human osteosarcoma (U2OS, ATCC HTB-96), human T lymphocyte (Jurkat E6-1, ATCC TIB-152, gift from L.E. Samelson, NIH) and African green monkey kidney fibroblast-like cell lines (COS-7, commercially provided by Leica Microsystems) were used in this study.

For live cell imaging, the U2OS cells were cultured and maintained at 37 °C and 5% $CO_2$ on a BIO-133 bottomed-well plate[24] for diSPIM imaging (Fig. 2a–c), on a no. 1.5 coverslip (VWR, 48393-241) for diSPIM imaging (Supplementary Video 1) or on glass bottom dishes (Mattek, P35G-1.5-14-C) for iSIM imaging (Fig. 3 and Extended Data Figs. 6–8), in 1 ml of DMEM medium (Lonza, 12-604F) containing 10% fetal bovine serum. At 40–60% confluency, cells were transfected with 100 µl of 1× PBS containing 2 µl of X-tremeGENE HP DNA Transfection Reagent (Sigma, 6366244001) and 2 µl of plasmid DNA (300–400 ng µl⁻¹) and then maintained at 37 °C, 5% $CO_2$ for 1–2 days before image acquisition. Cell ER was labeled by ERmoxGFP (Addgene, 68072), mitochondria labeled by mEmerald-Tomm20-C-10 (Addgene, 54281), Golgi apparatus labeled by GalT-GFP (plasmid was a gift from the Patterson Laboratory, NIH, NIBIB) and lysosomes labeled by Lamp1-EGFP (plasmid a gift from the Taraska Laboratory, NIH, NHLBI). Coverslips were prepared as previously described[30].

For widefield/confocal fixed cell imaging (Fig. 4a), U2OS cells were cultured on a glass bottom dish and fixed in 4% paraformaldehyde/PBS mixture at room temperature for 15 min, then permeabilized by 0.1% Triton X-100/PBS solution at room temperature for 2 min. Cells were rinsed three times by 1× PBS and labeled with 1:100 Alexa Fluor 568 Phalloidin (Thermo Fisher Scientific, A12380). For fixed COS-7 cell imaging (Fig. 4b), the samples (a gift from Leica Microsystem) were immunolabeled with 500 µl of 1:1,000 anti-NUP primary antibody (clone Mab414) and 1:200 goat-antimouse IgG secondary antibody conjugated with STAR635P.

For widefield fixed cell imaging (Fig. 5a–c and Supplementary Fig. 7), U2OS cells were cultured and fixed as above. Fixed cells were rinsed three times by 1× PBS and labeled with 500 µl of 1:100 anti-alpha tubulin primary antibody (Thermo Fisher Scientific, 322500), 1:200 anti-Tomm20 primary antibody (Abcam, 78547) and 1:100 Alexa Fluor

647 Phalloidin (Thermo Fisher Scientific, A22287) in 1× PBS at room temperature for 1 h. Labeling mixture was washed away in 1× PBS three times for 1 min for each time. Cells were then labeled with 500 µl of 1:500 Alexa-488 conjugated goat-antimouse secondary antibody (Invitrogen, A11011), 1:500 Alexa-568 conjugated goat-antirabbit secondary antibody (Invitrogen, A-110036) and 1 µg ml⁻¹ DAPI (Thermo Fisher Scientific, D1306) in 1× PBS at room temperature for 1 h. After immunolabeling, cells were washed three times (1 min for each time) in 1× PBS.

For confocal/STED imaging (Extended Data Fig. 9), U2OS cells were cultured and fixed as above. Fixed cells were immunolabeled with 1:200 anti-Tomm20 primary antibody (Abcam, 78547) and 500 µl of 1:500 donkey antirabbit secondary antibody conjugated with Alexa Fluor 594 (Jackson ImmunoResearch, 711-587-003). For 3D SIM imaging of Jurkat T cells expressing EMTB-3XGFP (Extended Data Fig. 10a), E6-1 Jurkat cells were cultured in RPMI 1640 supplemented with 10% fetal bovine serum and 1% pen-strep antibiotics. For transient transfections, cells were electroporated using the Neon transfection system (Thermo Fisher Scientific). Briefly, $2 \times 10^5$ cells were resuspended in 10 µl of R-buffer with 0.5–2 µg of EMTB-3XGFP (Addgene plasmid 26741) and exposed to three pulses of 1,325 V amplitude and 10 ms in duration. Cells were imaged 48 h posttransfection. Coverslips attached to eight-well Labtek chambers were incubated in 0.01% w/v poly-L-lysine (Sigma-Aldrich, P4707) for 10 min. Poly-L-lysine was aspirated and the slide left to dry for 1 h at 37 °C. T cell-activating antibody coating was performed by incubation of slides in a 10 µg ml⁻¹ solution of anti-CD3 antibody (Thermo Fisher Scientific, 14-0039-82) for 2 h at 37 °C or overnight at 4 °C. Excess anti-CD3 was removed by washing with L-15 imaging medium immediately before the experiment. For 3D SIM imaging of fixed U2OS cell expressing Lamp1-EGFP (Extended Data Fig. 10b), cell cultures at roughly 50% confluency were transfected using xTremegene HP DNA Transfection Reagent (Sigma, 6366236001). The transfection mixture contained 100 ml of 1× PBS, 2 ml of Transfection Reagent and 1 mg plasmid DNA. LAMP1-EGFP plasmid DNA (gift from G. Patterson's Laboratory) was used to label lysosomes.

The mouse brain sample imaged with 0.4/0.4 NA diSPIM (Fig. 2d) was prepared using the iDISCO+ procedure and published previously[8]. The tissue section was dissected from an 8-week old male mouse with vasopressin receptor 1B Cre X Ai9 provided by the NIMH (T.B. Usdin and S. Young). The sample was fixed by trans-cardiac perfusion with 4% paraformaldehyde, then dehydrated through a methanol series, rehydrated, immunolabeled with an antibody for tdTomato (Rabbit anti-red fluorescent protein, Rockland Antibodies and Assays, 600-401-379) and an Alexa Fluor 555 secondary antibody (Invitrogen, A27039). Before imaging with cleared-tissue diSPIM[8], the tissue slab was dehydrated with a methanol series and dichloromethane before equilibration in dibenzyl ether (Sigma, 108014).

For the cleared mouse brain samples (Fig. 2g, Extended Data Fig. 3b,c and Supplementary Video 2), fixed adult mouse brain expressing tdTomato in axonal projections from the area of a stereotaxic injection of adeno-associated virus was cleared using SDS and equilibrated in CUBIC-R[37]. Next, 2 mm thick coronal slabs were sectioned and held in a sample chamber custom designed for the CT-diSPIM. The animal rooms were on a 12-h light cycle, a temperature range of 21–23 °C (70–74 °F) and a humidity range of 30–70%. All animal studies were performed in a manner consistent with the recommendations established by the Guide for the Care and Use of Laboratory Animals (NIH), and all animal protocols were approved by the Animal Care and Use Committees in NIMH.

Nematode strains included BV24 ((*ltIs44* (*pie-1*p-mCherry::PH(PLC1delta1) + *unc-119*(+)); *zuIs178* ((*his-72* 1 kb::HIS-72::GFP); *unc-119*(+)) V), Extended Data Figs. 4a,b and 5b, Supplementary Video 3), od58 (*ltIs38* (*pie-1p*::GFP::PH(PLC1delta1) + *unc-119*(+))*, Fig. 1b and Extended Data Figs. 4c–f and 5a), AQ2953 ljIs 131((*myo-3p*::GCaMP3-SL2-tagRFP-T), Extended Data Fig. 4g) and DCR6268 ((*pttx-3b*::SL2::Pleckstrin homology domain::GFP::unc-54 3′ UTR + *pelt-7*::mCh::NLS::*unc-54* 3′ UTR)), Fig. 5d–f and Supplementary

Fig. 8). All worms were cultivated at 20 °C on nematode growth medium plates seeded with a lawn of *Escherichia coli* strain OP50. Embryos were dissected from gravid adults, placed on poly-L-lysine-coated coverslips and imaged in M9 buffer, as previously described[38].

### Simulation of phantom objects

To evaluate the quality and performance of our network, we generated 3D phantom objects consisting of three types of structure in MATLAB (Mathworks, R2019b, with the Imaging Processing Toolbox) for ground truth: dots, solid spheres and ellipsoidal surfaces (Supplementary Fig. 1). Each phantom was composed of 100 solid spheres, 100 ellipsoidal surfaces and 400 dots, randomly located in a 128 × 128 × 128 volume. The 100 solid spheres were generated with random intensity (50–850 counts) and random diameter (4–8 voxels). The 100 ellipsoidal surfaces were generated with random intensity (50–850 counts), random diameter along different axes (4–8 voxels) and random thickness (1–2 voxels); the 400 dots were generated with random intensity (50–850 counts) and random extent along each direction (1–3 voxels). The background value was set to a constant at 30 counts.

Noiseless input volumes were generated by convolving the ground truth data with different PSFs (Supplementary Fig. 1). Five types of PSF were used, including: the system PSF for the 0.8/0.8 NA diSPIM that has threefold larger axial extent compared to its lateral extent[23] for the generalization test on embryo nuclei and membrane data Extended Data Fig. 5); the system PSF of iSIM[11] for the generalization test of ER volumes (Fig. 3e and Extended Data Fig. 7d–f), the system PSFs of the widefield microscope (Leica, LAS X, DM18, ×63/1.40 OIL ultraviolet) for the generalization test of the fixed U2OS cells and fixed COS-7 cells (Fig. 4), the system PSFs of the widefield microscope (Olympus, UPLXAPO60XO, ×60, NA of 1.42 oil objective) for the generalization test of the four-color fixed U2OS cells (Fig. 5a–c), and the system PSF of the widefield microscope (Olympus UPLSAPO60XWPSF, ×100, NA of 1.35 silicon oil lens) for the generalization test of *C. elegans* embryo expressing ttx-3B-GFP (Fig. 5d–f). Noisy images were then obtained by adding different levels of Gaussian and Poisson noise.

The 3D human brain phantom was downloaded from the Zubal Phantom website[39] (http://noodle.med.yale.edu/zubal/data.htm, Fig. 1e,f and Supplementary Fig. 3). The simulated spherical phantoms ground truths were generated with ImgLib2 (ref. [22]) and blurred with a 3D Gaussian kernel with standard deviation set to 2 pixels, the maximum radius of the spheres was set at seven pixels and the intensity range to 80–255 (Fig. 1d and Supplementary Figs. 2 and 5). Network inputs of these structures (Fig. 1d,e) and their corresponding training data were blurred with the system PSF of the 0.8/0.8 NA diSPIM.

### DiSPIM data acquisition and processing

A fiber-coupled diSPIM[20] with two ×40, 0.8 NA water objectives (Nikon catalog no. MRD07420), resulting in a pixel size of 162.5 nm, was used to image the U2OS cell transfected with mEmerald-Tomm20-C-10 (Fig. 2a and Supplementary Video 1), transgenic embryos strain od58 expressing GFP-membrane (Fig. 1b and Extended Data Figs. 4c–f and 5a) and BV24 expressing GFP-nuclei (Extended Data Figs. 4a,b and 5b and Supplementary Video 3). For cellular imaging, 50–200 dual-view volumes (60 planes, 1 μm interplane spacing in each view) were acquired with 3 s intervals; for embryo imaging, dual-view stacks (50 planes at 1 μm spacing per view) were acquired at 1-min intervals for 291 min. Dual-view data were registered and jointly deconvolved with the diSPIM Fusion ImageJ plugin[8] to generate ground truth, using ten iterations for joint deconvolution.

### DiSPIM cleared-tissue acquisition and processing

Cleared-tissue image data in Fig. 2d was acquired on a fiber-coupled diSPIM that was modified for cleared-tissue imaging by incorporating elements of the commercially available ASI DISPIM and DISPIM for

Cleared Tissue (CT-DISPIM)[8]. We used a pair of Special Optics 0.4-NA multi-immersion objectives (ASI, 54-10-12). The cleared mouse brain volumes were acquired by moving the stage (2 μm step size, total 4,800 frames with 2,048 × 2,048 pixels) in a raster pattern with the aid of the ASI diSPIM Micromanager plugin (http://dispim.org/software/micro-manager). Image data for Fig. 2g were acquired on a dedicated, commercial ASI CT-DISPIM equipped with a pair of Special Optics 0.7-NA multi-immersion objectives (ASI, 54-12-8). Using the DISPIM plugin in Micromanager, we set up a multi-position acquisition in light-sheet mode with unidirectional stage scan. Image FOV was set to 1,536 × 1,536 pixels to avoid geometric distortions near the edge of the full FOV (2,048 × 2,048). Five *y* positions and two *z* positions were acquired with 15% overlap, each position was a stack of 1,573 images with a stage step of 1.414 μm.

Dual-view data were registered and jointly deconvolved based on Wiener–Butterworth filter back projector (one iteration) for the ground truth, using MATLAB (Mathworks, R2019b, with the Imaging Processing and Parallel Computation Toolboxes)[8] on a computer workstation equipped with Intel(R) Xeon(R) W-2145 CPU at 3.70 GHz and Nvidia Quadro P6000 with 24 GB memory.

For joint deconvolution of cleared mouse brain samples in Fig. 2g, running on NIH Biowulf cluster, we modified the code to meet the high-performance computing at NIH requirements for job scheduling. The 28-core 'gpu' queue for Biowulf (28 × 2.4 GHz Intel E5-2680v4 processor, four NVIDIA P100 GPUs, 16 GB VRAM, 3,584 cores) was used for computing.

### iSIM data acquisition and processing

A home-built iSIM system[29] with a ×60, 1.2 NA water objective (Olympus UPLSAPO60XWPSF) and an sCMOS camera (PCO, Edge 5.5), resulting in a pixel size of 55 nm, was used to image the U2OS cells (Fig. 3 and Extended Data Figs. 6–8). All raw volumes were background subtracted and deconvolved using the RL algorithm with 15 iterations to generate ground truth.

### Widefield data acquisition and processing

Widefield fixed U2OS and COS-7 cell images (Fig. 4) were acquired with a Leica widefield microscope (LAS X, DM18, ×63/1.40 OIL ultraviolet, 102 nm pixel size) and processed with the Leica Thunder computational clearing method (commercial deconvolution software designed for deblurring widefield volumes, using the small volume computational clearing method with default settings, Strategy: Adaptive; Thunder Strength: 60; Thunder Regularization 5.05 × 10[6]), then the same samples were acquired with Leica confocal microscopy (HC PL APO CS2 ×63/1.40 OIL, a pixel size of 102 nm). The same FOV of widefield and confocal data were found manually, then finely registered with an affine transformation[8]. Widefield fixed U2OS images (Fig. 5a–c and Supplementary Fig. 7) were acquired by a home-built widefield microscope with a ×60, 1.42 NA oil objective (Olympus, UPLXAPO60XO) and a pixel size of 266 nm. Widefield *C. elegans* embryos (Fig. 5d–f and Supplementary Fig. 8) were acquired with a ×100, 1.35 NA silicon oil lens (Olympus UPLSAPO60XWPSF) and a pixel size of 111 nm.

### Reflective diSPIM data acquisition and processing

The geometry of the diSPIM (0.8/0.8 NA) used for reflective imaging has been previously described[28]. Glass coverslips were sputtering a 150-nm-thick aluminum film over their entire surface and then protecting them with a 700-nm-thick layer of SiO2 (Thin Film Coating). During reflective imaging, four views (direct fluorescence and mirror images) were simultaneously collected in stage scanning mode with the same detection optics. The exposure time for each plane was 5 ms. The ground truth consisted of deconvolving the registered input using a spatially varying PSF and the Wiener–Butterworth unmatched back projector[8] with two iterations (Extended Data Fig. 4g).

## Confocal and STED data acquisition and processing

A commercial Leica STED system (HC PL APO CS2 ×100/1.40 OIL) and a Leica confocal microscopy (HC PL APO CS2 ×63/1.40 OIL) were used to acquire the confocal/STED training and testing datasets (Extended Data Fig. 9).

## 3D SIM data acquisition and processing

A home-built 3D SIM system inspired by previous designs[40–43] with a ×60, 1.27 NA water objective was used to image fixed Jurkat T cells expressing EMTB-3XGFP that had settled on anti-CD3 coated coverslips (Extended Data Fig. 10a) and fixed U2OS cells expressing Lamp1-EGFP (Extended Data Fig. 10b). The raw 15 input images had a pixel size of 82 nm, and were (1) summed to form diffraction-limited widefield images, then interpolated by a factor of two as input data for RLN training and (2) used to generate 3D SIM reconstructions as ground truth, via a generalized Wiener filter[41] with final pixel size of 41 nm. The microscope will be fully described in a forthcoming publication.

## Reporting summary

Further information on research design is available in the Nature Research Reporting Summary linked to this article.

## Data availability

The data that support the findings of this study are included in the Extended Data Figures and Supplementary Videos, with some representative source data for the main figures (Figs. 1d, 2a,d, 3c, 4b and 5a) publicly available at https://zenodo.org/record/7023909#.Ywl-QI3HMJaR. The 3D human brain phantom can be downloaded from the Zubal Phantom website (http://noodle.med.yale.edu/zubal/data.htm). Other datasets are available from the corresponding author upon reasonable request.

## Code availability

Code for the simulation of 3D mixture phantoms, generation of simulated input data and RLN training/prediction (with a small test dataset) are available at https://github.com/MeatyPlus/Richardson-Lucy-Net. DiSPIM acquisition software was developed in Micromanager v.1.4 (http://dispim.org/software/micro-manager). Other microscope acquisition code was written in Python v.3.7.0 and MATLAB v.2019b and is available upon request. The RLD and Wiener–Butterworth deconvolution algorithms were written in MATLAB 2019b and are available at https://github.com/eguomin/regDeconProject/tree/master/WBDeconvolution.

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

## Acknowledgements

This research was supported by the intramural research programs of the National Institute of Biomedical Imaging and Bioengineering, NIH. Y.L. and H.L. acknowledge support from the National Key Research and Development Program of China (grant no. 2020AAA0109502), the National Natural Science Foundation of China (U1809204, 61701436) and the Talent Program of Zhejiang Province (grant no. 2021R51004). T.B.U. acknowledges support from the intramural research programs of NIMH, NIH (grant no. ZIC MH002963-05). We thank the Marine Biological Laboratories (MBL) for providing meeting and brainstorming platform. H.S. and D.A.-C-R. acknowledge the Whitman and Fellows program at MBL for providing funding and space for discussions valuable to this work. Research in the D.A.C.-R. laboratory was supported by NIH grant no. R24-OD016474. M.W.W. was supported by NIH grant no. F32-NS098616. A.U. acknowledges support from grant nos. NIH R01 GM131054 and NSF PHY 1806903. We thank H. Eden and Patrick La Riviere for a careful read of and valuable feedback on the manuscript. We thank L. Elliot and J. Marr from Leica Microsystems and M. Smelkinson from the NIAID imaging core, for providing the NPC sample (Fig. 4b) and for supporting the use of Leica commercial systems and Thunder software. We thank Z. Bao for providing the OD58 *C. elegans* strain used in Fig. 1b and Extended Data Fig. 4, J. Bui, G. Kroeschell and M. Chaw for maintaining the worm strains, and W.S. Young (NIMH) for providing early access to the V1b mouse line. We thank G. Patterson's Laboratory for providing the LAMP1-EGFP plasmid DNA used in Extended Data Fig. 10b. This work used the computational resources of the NIH HPC Biowulf cluster (http://hpc.nih.gov). This work was supported by the Howard Hughes Medical Institute. This article is subject to HHMI's Open Access to Publications policy. HHMI laboratory heads have previously granted a nonexclusive CC BY 4.0 license to the public and a sublicensable license to HHMI in their research articles. Pursuant to those licenses, the author-accepted manuscript of this article can be made freely available under a CC BY 4.0 license immediately on publication. The NIH and its staff do not recommend or endorse any company, product or service.

## Author contributions

Y.W. and H.L. conceived the project. Y.L. and Y.W. developed and tested the deep learning algorithms and software. Y.W., Y.S., X.H., H.D.V., J.C. and H.S. designed and performed experiments. Y.W., M.G. and H.D.V. built the instrumentation. T.S., M.W.M., I.R.-S., A.U., T.B.U. and D.C.-R. contributed reagents and advice on biological questions and interpretations. Y.L., Y.S., M.G., J.L., X.L., T.S., R.C., Y.W. and H.S. processed and analyzed data. Y.L., Y. W. and H.S. wrote the manuscript. H.S. supervised the research.

## Competing interests

The authors declare no competing interests.

## Additional information

**Extended data** is available for this paper at

**Supplementary information** The online version
contains supplementary material available at

**Correspondence and requests for materials** should be addressed to
Huafeng Liu or Yicong Wu.

**Peer review information** *Nature Methods* thanks the anonymous
reviewers for their contribution to the peer review of this work.Primary
team. Peer reviewer reports are available.

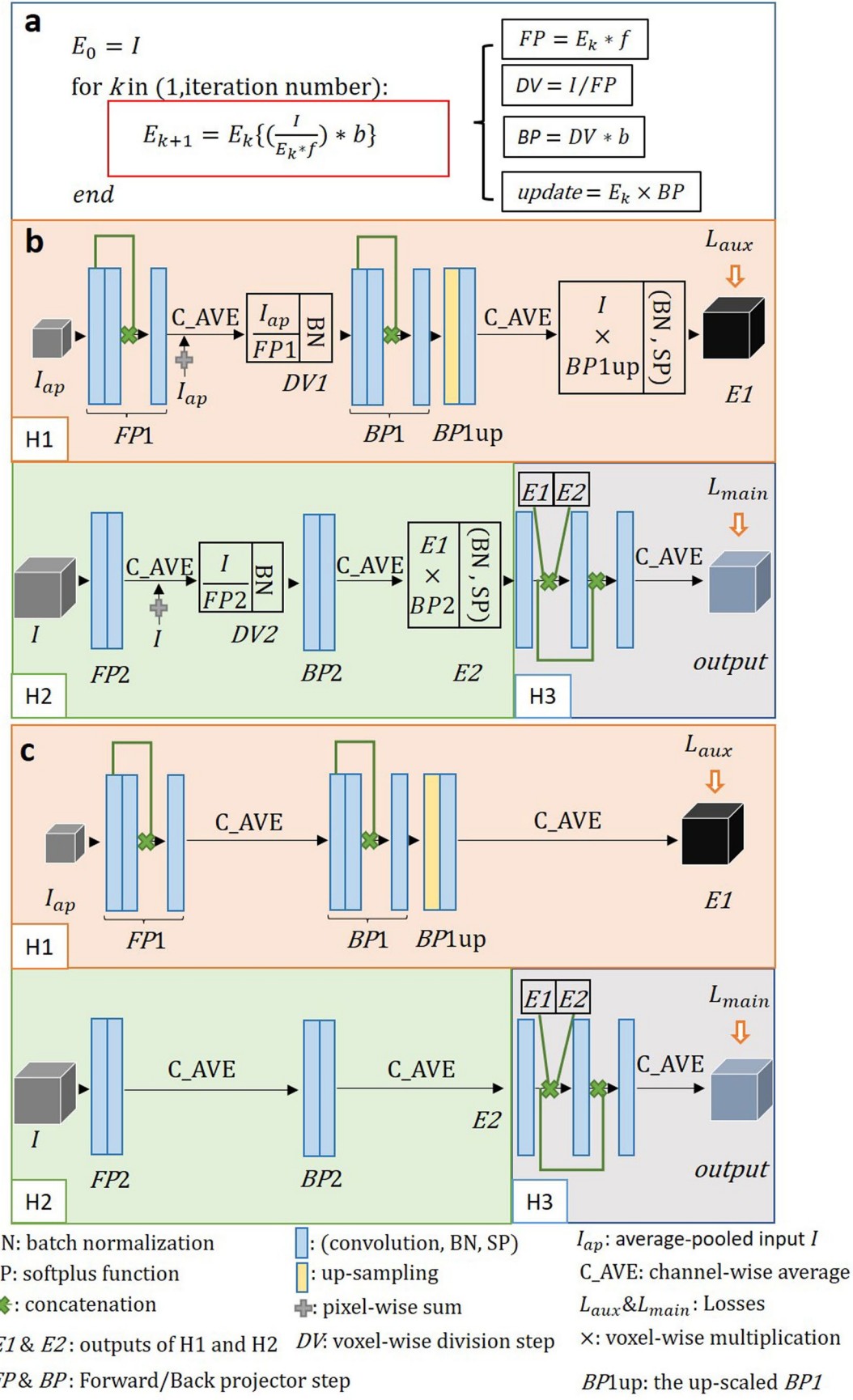

$$E_0 = I$$

for $k$ in (1, iteration number):

$$E_{k+1} = E_k\{(\frac{I}{E_k * f}) * b\}$$

end

$FP = E_k * f$

$DV = I/FP$

$BP = DV * b$

$update = E_k \times BP$

BN: batch normalization
SP: softplus function
❌: concatenation
E1 & E2: outputs of H1 and H2
FP & BP: Forward/Back projector step

▮: (convolution, BN, SP)
▮: up-sampling
➕: pixel-wise sum
DV: voxel-wise division step

$I_{ap}$: average-pooled input $I$
C_AVE: channel-wise average
$L_{aux}$ & $L_{main}$: Losses
×: voxel-wise multiplication
BP1up: the up-scaled BP1

**Extended Data Fig. 1 | See next page for caption.**

**Extended Data Fig. 1 | Decomposition of RL deconvolution iteration and the internal structure of RLN and RLN-a. a**) RL deconvolution can be decomposed into four parts: forward projector (*FP*) function, division (*DV*) step, back projector (*BP*) function and *update* step. **b**) Schematic of RLN consisting of three parts: down-scale estimation starting from the average-pooled input image, H1; original-scale estimation starting with original-scale input image, H2; and merging/fine-tuning, H3. H1 and H2 are inspired by the RL deconvolution update formula, which mimic the unmatched forward/back projection steps. **c**) Schematic of RLN-a, without the *DV* and *update* steps in RLN. See Methods for more detail.

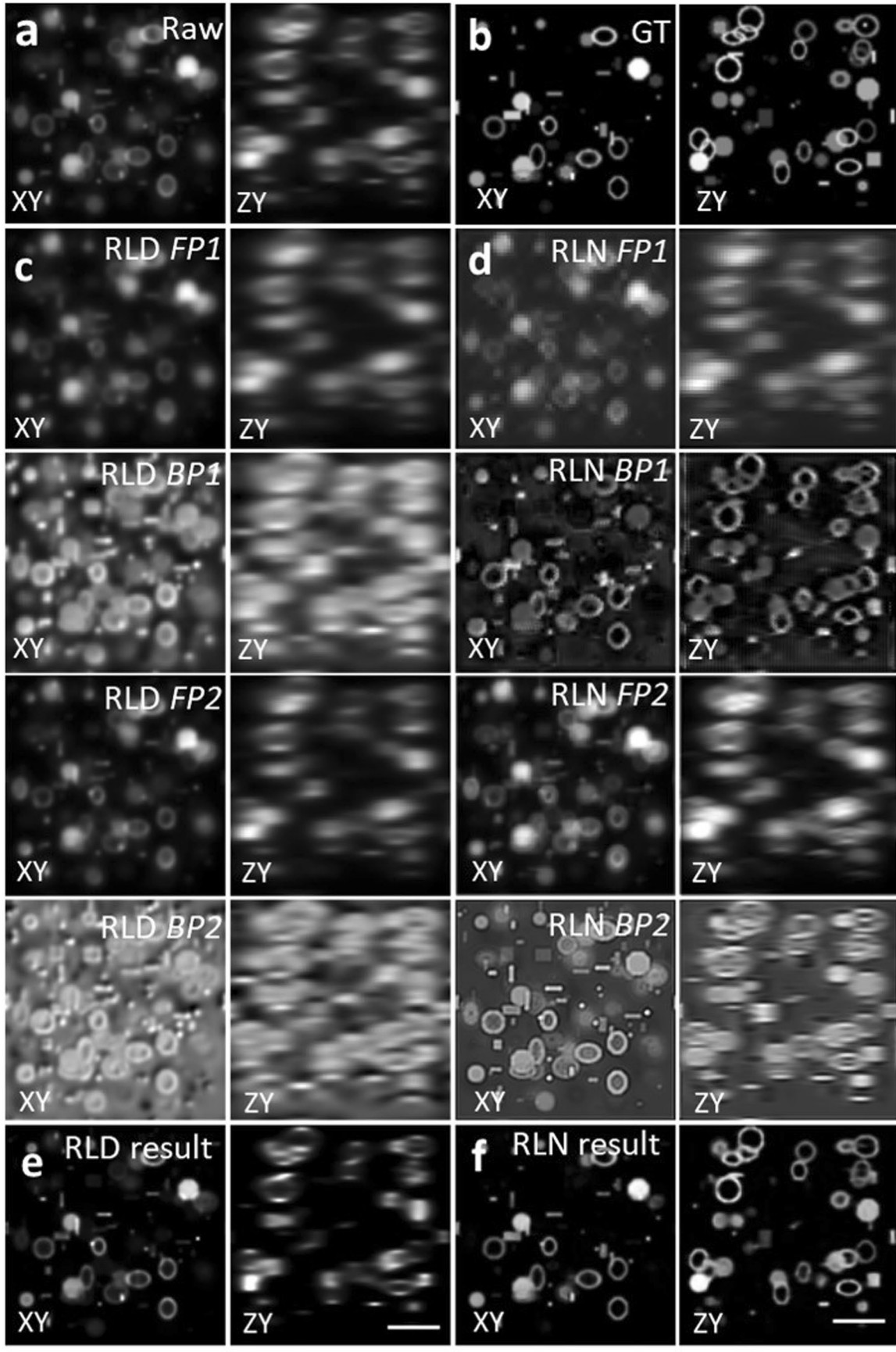

**Extended Data Fig. 2 | See next page for caption.**

**Extended Data Fig. 2 | Comparing RLD and RLN on a phantom object consisting of dots, solid spheres, and ellipsoidal surfaces.** Lateral (XY) and axial (ZY) views are presented in each case. **a**) Raw input, that is, blurry image. **b**) Ground truth object. **c**) Intermediate output of RLD. *FP1* and *BP1* are the forward projector function and backward projector function at iteration 1, and *FP2, BP2* at iteration 20. **d**) Intermediate steps of RLN. *FP1, BP1, FP2, BP2* are the steps in H1 and H2 shown in the neural network (Extended Data Fig. 1). The similarities between the RLD and RLN intermediate steps indicates the RLN internal feature maps are interpretable. **e**) Final RLD result after 40 iterations. **f**) RLN result, which is much closer to the ground truth (SSIM 0.95, PSNR 31.1) than RLD (SSIM 0.67, PSNR 19.4). Scale bars: 5μm.

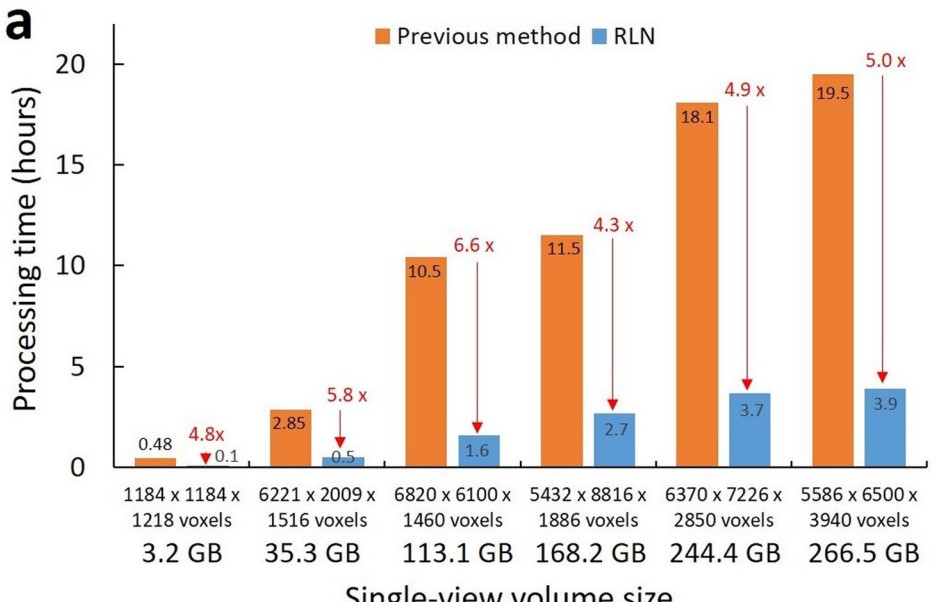

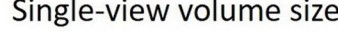

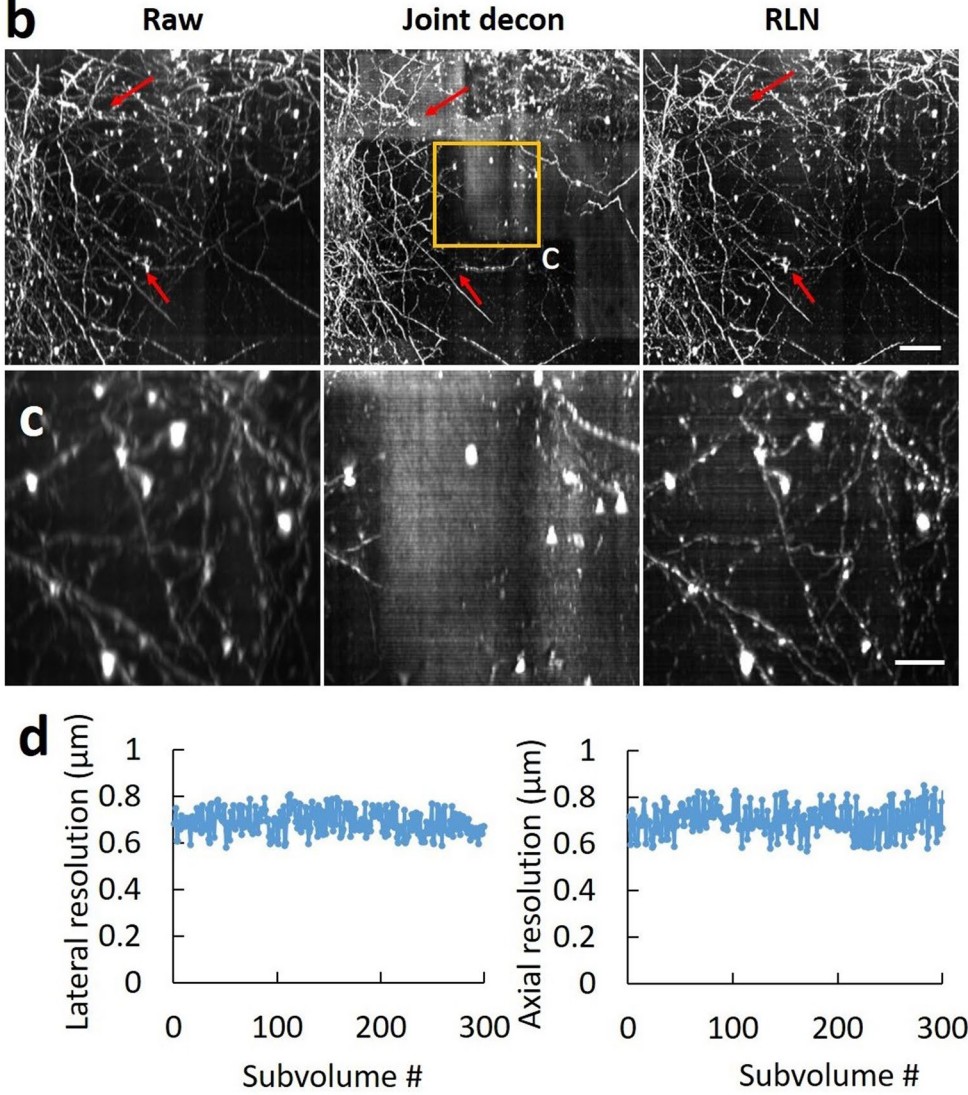

**Extended Data Fig. 3 | See next page for caption.**

**Extended Data Fig. 3 | RLN outperforms the previous processing pipeline for reconstructions of large, cleared tissue datasets imaged with diSPIM.**
**a**) Processing times for RLN vs. previous pipeline on differently sized volumes. In the RLN pipeline, the large raw single-view input volume is cropped into subvolumes, RLN is applied to each subvolume, and the predictions on each subvolume are stitched back into a single volume. The previous pipeline (1) downsamples the large raw dual view volumes, performing affine registration of the downsampled volumes to obtain a transformation matrix; (2) crops the raw dual view volumes into multiple subvolumes using the transformation matrix to guide the cropping coordinates; (3) applies an affine registration to the dual-view subvolumes; (4) performs joint deconvolution on the registered dual-view subvolumes with the Wiener–Butterworth backprojector; (5) stitches the deconvolved subvolumes back into a single volume. Raw input data are 16-bit. The 3.2 GB and 168.2 GB data were from the cleared brain tissue shown

in Fig. 2, others were from the previously published cleared tissue samples and reprocessed with RLN. As shown, RLN accounts for a 4- to 6-fold speed improvement. **b**) Comparisons of single-view raw, dual-view joint deconvolution, and RLN predictions from the subvolume shown in Fig. 2g. Red arrows indicate artifacts in dual-view joint deconvolution result, relative to the raw input and RLN prediction. We suspect these artifacts are likely due to failures of registration between the two raw views. **c**) Magnified view of rectangular regions in **b**), further demonstrating the poor joint deconvolution results. **d**) Estimating variability of resolution in RLD predictions over 300 subvolumes (each 600 ×600 x 600 voxels) cropped from the large, cleared tissue dataset used in Fig. 2g. Shown are lateral and axial resolution estimates from decorrelation analysis, with variation within 10% (688 ± 51 nm laterally, 701 ± 60 nm axially, N = 300 subvolumes). Scale bars: **b**) 50 μm; **c**) 20 μm.

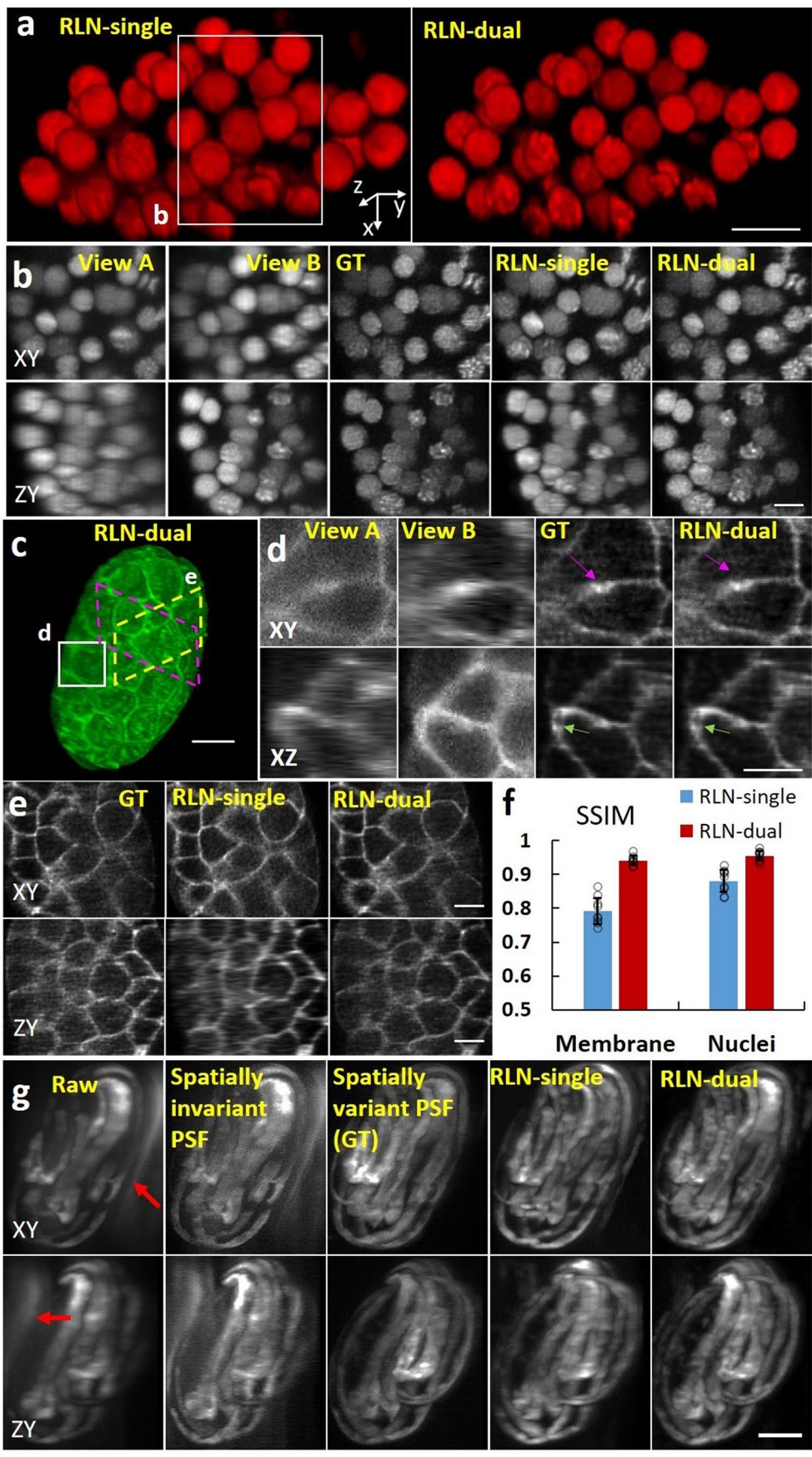

**Extended Data Fig. 4 | See next page for caption.**

**Extended Data Fig. 4 | Dual-input RLN improves axial resolution relative to single-input RLN on scattering samples. a**) 3D rendering of nuclei expressed in live *C. elegans* embryo collected by diSPIM and processed with single-input RLN and dual-input RLN. **b**) Higher magnification of white rectangle in **a**), comparing raw views A/B, joint deconvolution of views A/B (GT), single- and dual-input RLN predictions. The dual-input RLN uses the information available in both views to achieve resolution isotropy, providing near-identical output to the ground truth. **c**) 3D rendering of membrane marker in live *C. elegans* embryo, raw data collected by diSPIM and processed with dual-input RLN. **d**) Higher magnification of white rectangle in **c**), comparing raw views A/B joint deconvolution (GT), and dual-view RLN predictions. Dual-input RLN uses complementary dual-view information to enable near-isotropic resolution, providing results near-identical to the ground truth (magenta and green arrows). **e**) Higher magnification of dashed yellow and magenta parallelograms in **c**), comparing dual-view deconvolution ground truth, single- and dual-view RLN predictions. Dual-input RLN better recovers axial resolution than single-input RLN. **f**) SSIM analysis for data displayed in **b**),

**e**), showing that RLN-dual better recovers the signals than RLN-single when the views are contaminated by scattering. Individual values (open circles), means, and standard deviations from N = 9 volumes are shown for membranes and nuclei. **g**) Maximum-intensity projections of raw images of *C. elegans* embryo expressing GCaMP3 acquired with reflective diSPIM, deconvolution based on spatially invariant PSF, deconvolution based on spatially variant PSF (ground truth 'GT'), single-input RLN, and dual-input RLN predictions (using two orthogonal views as input). Raw input is contaminated with spatially varying epifluorescence background (red arrows). Spatially variant deconvolution, single-input RLN, and dual-input RLN remove associated epifluorescence contamination, enhancing resolution and contrast. PSNR and SSIM analysis confirms this result (Raw: PSNR 22.65 ± 1.46, SSIM 0.52 ± 0.04; spatially invariant deconvolution: PSNR 26.61 ± 1.53, SSIM 0.62 ± 0.02; single-input RLN: PSNR 31.56 ± 0.70, SSIM 0.84 ± 0.02; dual-input RLN: PSNR 34.55 ± 0.61, SSIM 0.87 ± 0.01; N = 12 volumes). Scale bars: **a, b, d, e**) 5 µm, **c, g**) 10 µm.

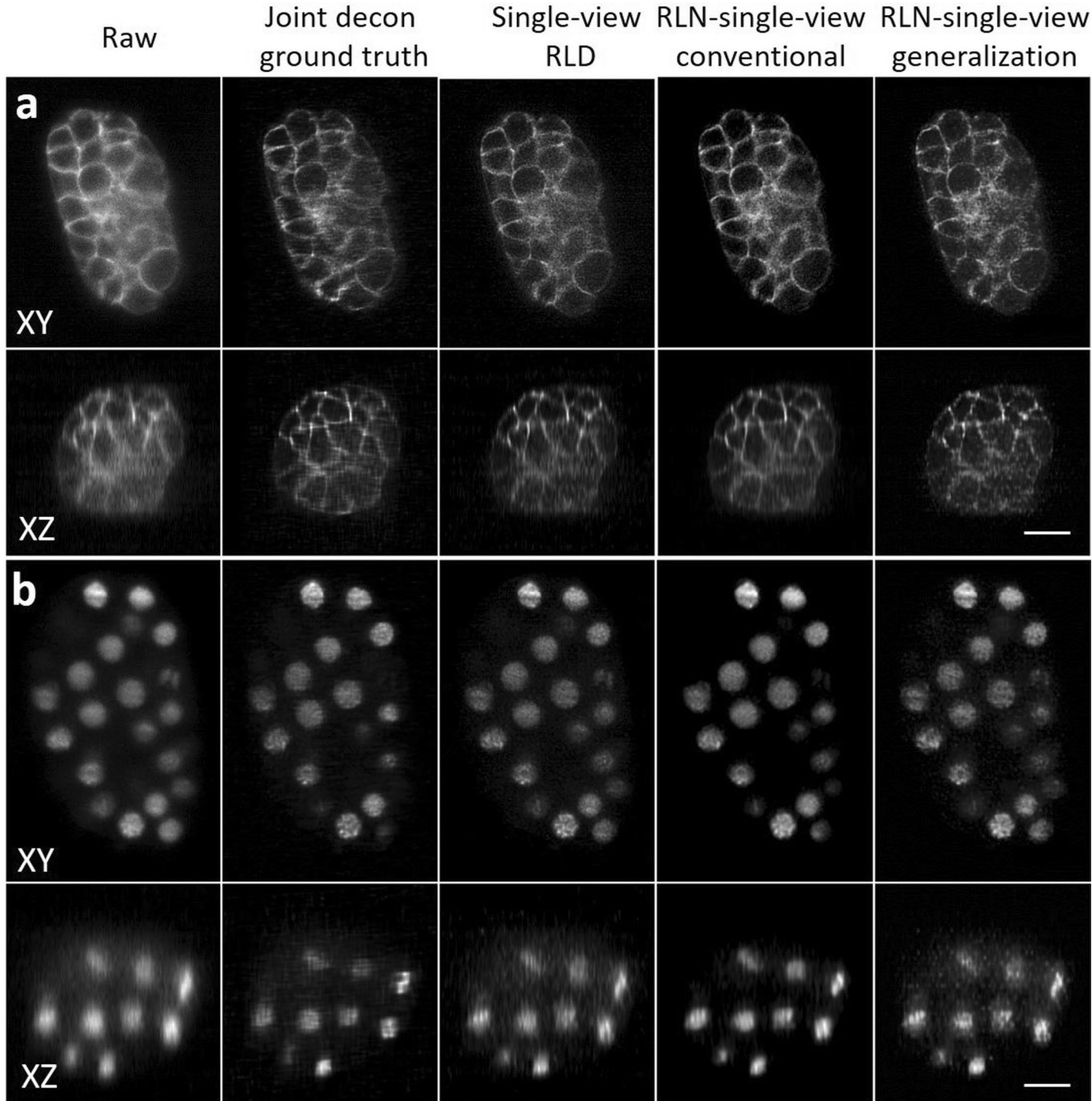

**Extended Data Fig. 5 | RLN models trained on synthetic mixed data generalize well to images of *C. elegans* embryos acquired with diSPIM. a)** Membrane and **b)** nuclei results, comparing raw input, dual-view joint deconvolution ground truth, single-view RLD, and predictions from RLN with single-view input under conventional testing (trained on similar images to test data) vs. generalization (models trained with mixed phantoms of dots, solid spheres, and ellipsoidal surfaces). Outputs of single-view RLD, conventional testing, and generalization show close visual resemblance to each other. Quantitative assessments show that the conventional testing results (SSIM-membrane: 0.80, SSIM-nuclei: 0.85, PSNR-membrane: 28.9, PSNR-nuclei: 27.0) is closest to the dual-view joint deconvolution ground truth, while the generalization results (SSIM-membrane: 0.75, SSIM-nuclei: 0.76, PSNR-membrane: 27.3; PSNR-nuclei: 26.1) compare favorably against single-view RLD (SSIM-membrane: 0.74, SSIM-nuclei: 0.73, PSNR-membrane: 27.1, PSNR-nuclei: 25.9). Scale bars: 10 µm.

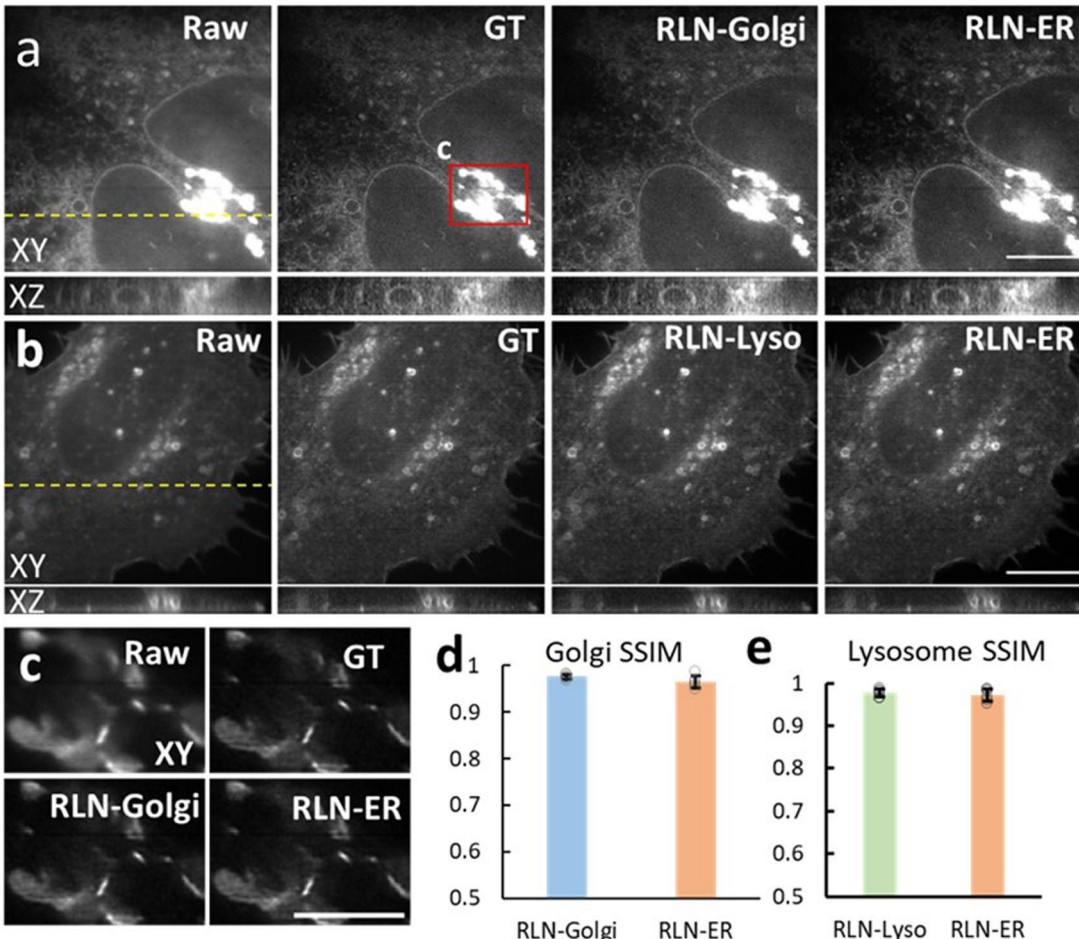

**Extended Data Fig. 6 | Generalization ability of RLN tested with ER, Golgi, and Lysosome markers, imaged with iSIM. a**) Lateral and axial views of live U2OS cells expressing GalT-GFP, acquired with iSIM, comparing the raw input, deconvolved iSIM ground truth, predictions from Golgi-trained RLN (RLN-Golgi) and ER-trained RLN (RLN-ER). Yellow dashed line indicates the Y position of the XZ plane. **b**) Lateral and axial views of live U2OS cells expressing Lamp1-EGFP, acquired with iSIM, comparing the raw input, ground truth, predictions from lysosome-trained RLN (RLN-Lyso) and ER-trained RLN (RLN-ER). Axial view is taken along yellow dashed line. **c**) contrast-adjusted higher magnification view of red rectangular region in **a**). **d**) Quantitative SSIM analysis for data shown in **a**), individual values (open circles), means, and standard deviations are shown from N = 6 volumes. **e**) Quantitative SSIM analysis for data shown in **b**), individual values (open circles), means and standard deviations are shown from N = 6 volumes. Scale bars: **a, b**) 10 μm, **c**) 5 μm.

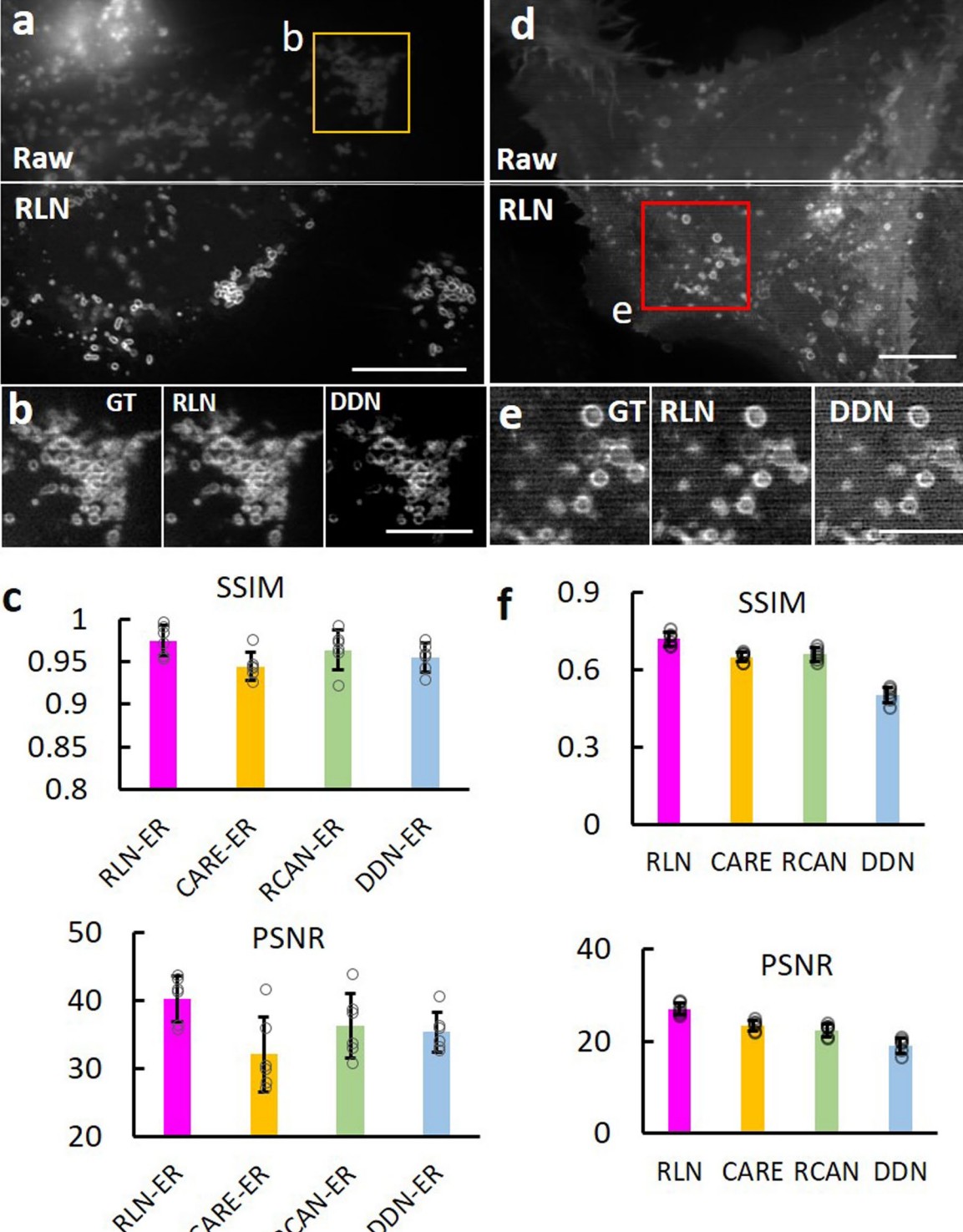

**Extended Data Fig. 7 | RLN provides better generalization on super-resolution data than other networks. a**) Super-resolved images of live U2OS cells expressing mEmerald-Tomm20-C-10, acquired with iSIM. Top: raw input; bottom: RLN output. **b**) Higher magnification view of yellow rectangular region in **a**), comparing ground truth, RLN output, and DDN output. The models were trained with ER datasets. **c**) SSIM and PSNR measurements for RLN, CARE, RCAN and DDN for data shown in **a**), means and standard deviations are obtained from N = 6 volumes. **d**) Super-resolved images of live U2OS cells expressing Lamp1-EGFP, acquired with iSIM. Top: raw input; bottom: RLN output. **e**) Higher magnification of rectangular regions in **d**), comparing ground truth, RLN output, and DDN output. Models were trained with phantom objects consisting of dots, solid spheres, and ellipsoidal surfaces. **f**) SSIM and PSNR measurements, comparing RLN, CARE, RCAN and DDN for data shown in **d**), means and standard deviations are obtained from N = 6 volumes (open circles indicate individual values). Scale bars: **a, d**) 5 μm, **b, e**) 2 μm.

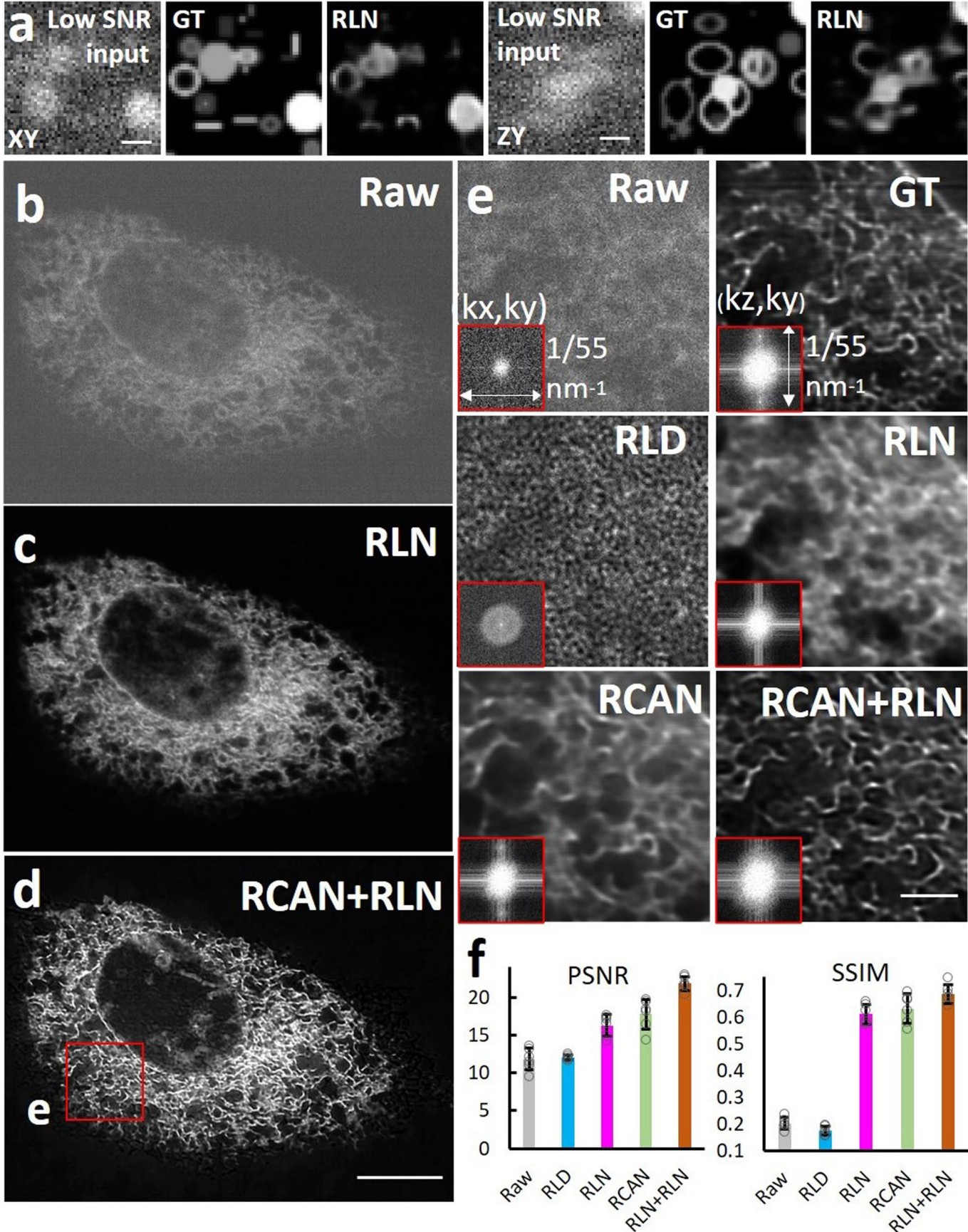

**Extended Data Fig. 8 | See next page for caption.**

**Extended Data Fig. 8 | RLN's performance on volumes with very noisy input.**
**a**) An example of the failure of RLN when challenged with very noisy input. RLN was trained with noisy (SNR 2.45 dB) synthetic data (Supplementary Fig. 1) and tested on similar structures. Lateral (XY) and axial (ZY) views are shown. RLN results show obvious artifacts. **b-f**) Comparison of RLD and RLN on super-resolved, low-SNR data from a live U2OS cell expressing ERmoxGFP, acquired with iSIM. **b**) Low SNR input, XY view. **c**) RLN output with one step training, that is, applying a model trained with low SNR raw input and high SNR deconvolved ground truth. **d**) RLN output with two-step deep learning, by first applying a denoising RCAN model, then applying an RLN model to deconvolve the output from the first step. The RCAN model was based on pairs of low/high SNR raw data, the RLN model was based on pairs of high SNR raw data and high SNR deconvolution data. **e**) Higher magnification of red rectangle in **d**), comparing raw input, high SNR deconvolved ground truth, RLD on low SNR raw input, one-step RLN, one-step RCAN (same training data as with one-step RLN, that is, input is the low SNR raw input and the high SNR deconvolved result is the ground truth), and two-step deep learning with RCAN for denoising and RLN for deconvolution. Insets show Fourier transforms of the data. **f**) Quantitative analysis with PSNR and SSIM for the raw input, RLD, one-step RLN, one-step RCAN, and two-step RCAN + RLN result, open circles, means, and standard deviations are obtained from N = 6 volumes. Both one-step RCAN and one-step RLN outperform RLD, and the two-step methods further boosts resolution and contrast, indicated by the Fourier spectra shown in the inserts. Scale bars: **a**) 10 pixels, **b-d**) 10 μm, **e**) 2 μm.

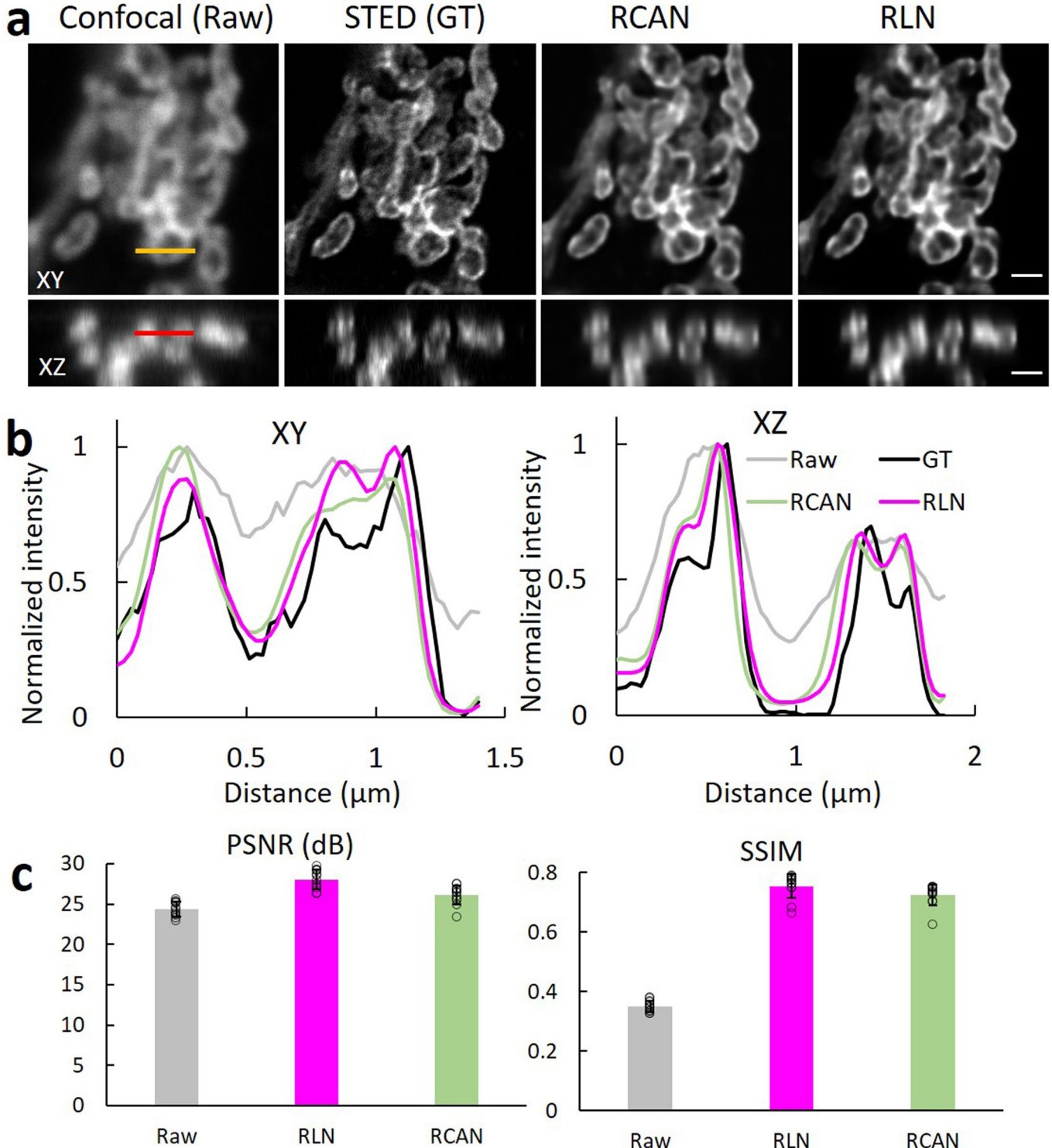

**Extended Data Fig. 9 | RLN outperforms RCAN when attempting confocal-to-STED cross modality prediction.** a) Lateral views (top) and axial views (bottom) of U2OS cells immunolabeled with a primary antibody against Tomm20 and an anti-rabbit secondary antibody conjugated with Alexa Fluor 594, comparing the raw input collected by confocal microscopy, STED microscopy images (ground truth), RCAN output, and RLN output. b) Line profiles across the yellow and red lines in the lateral view and axial view in a, demonstrating that RCAN and RLN improve resolution compared to the input, yet not to the extent of the ground truth. c) SSIM and PSNR analysis for data shown in a), means and standard deviations are obtained from N = 10 slices. Scale bars: 4 µm.

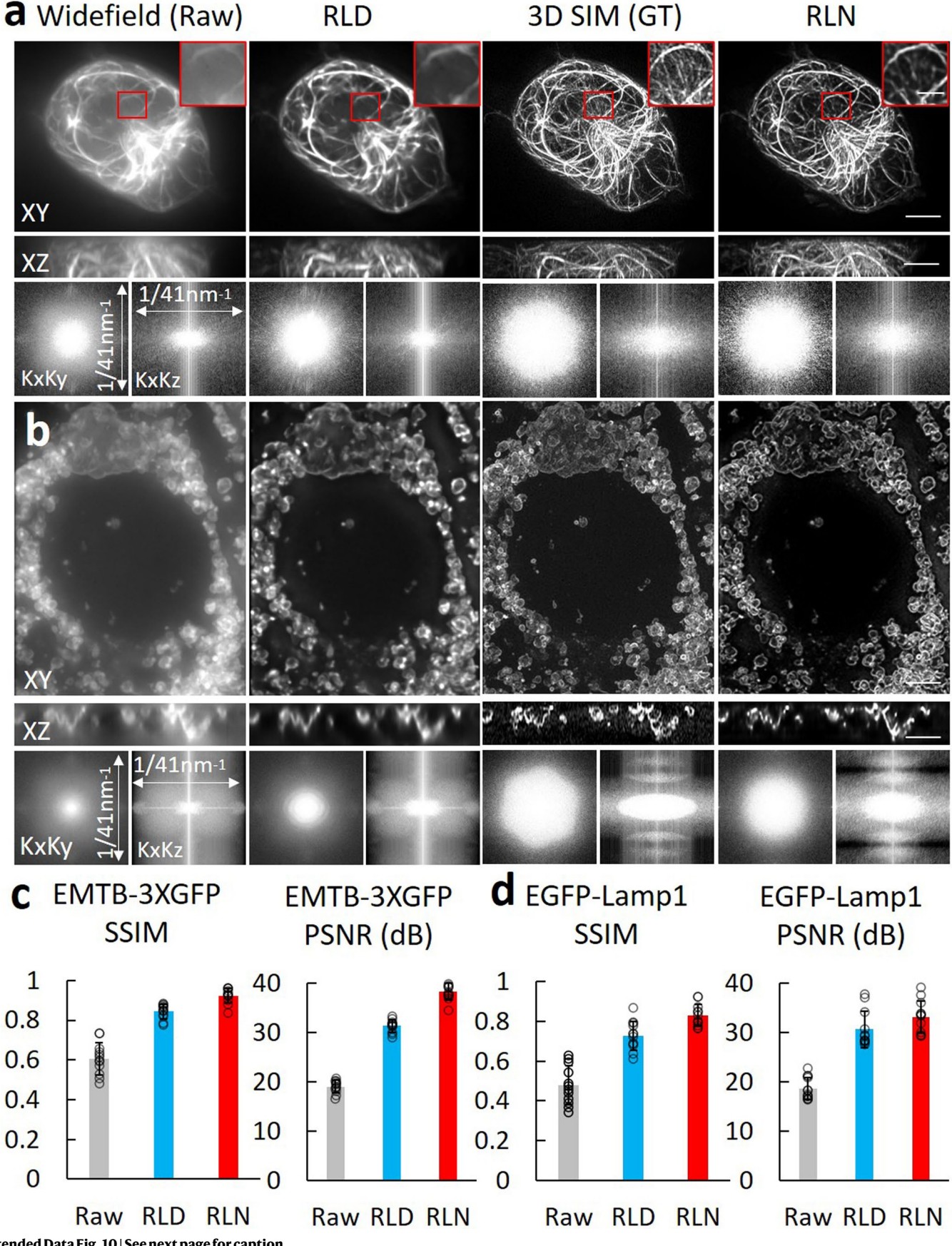

**Extended Data Fig. 10 | See next page for caption.**

**Extended Data Fig. 10 | RLN outperforms RLD when attempting to restore widefield data based on 3D SIM ground truth, under both conventional and generalization testing. a**) Lateral maximum projection (top), axial maximum projection (middle), and Fourier spectra (bottom) of Jurkat cells expressing EMTB-3XGFP, comparing the raw input collected by widefield microscopy, RLD with 40 iterations, 3D SIM images (that is, the ground truth), and conventional test result of RLN. Decorrelation analysis shows the resolution improvement of RLN compared to RLD (raw: 348 ± 20 nm laterally, 798 ± 133 nm axially; RLD: 272 ± 9 nm laterally, 621 ± 120 nm axially; RLN: 156 ± 7 nm laterally, 414 ± 11 nm axially; GT: 117 ± 1 laterally, 385 ± 10 nm axially, N = 15 slices). Magnified insets corresponding to the red rectangular regions indicate that RLN output fails to predict some dim filaments. **b**) Lateral maximum intensity projection (top), axial maximum intensity projection (middle), and Fourier spectra (bottom) of U2OS cells expressing Lamp1-EGFP, comparing the raw input collected by widefield microscopy, RLD with 40 iterations, 3D SIM images (ground truth), and the generalization test result of RLN (that is, the model is the same one as used in **a**), trained on Jurkat cells expressing EMTB-3XGFP). Quantitative resolution analysis by decorrelation methods shows the resolution improvement of RLN compared to RLD (raw: 368 ± 22 nm laterally, 837 ± 66 nm axially; RLD: 245 ± 7 nm laterally, 641 ± 109 nm axially; RLN: 146 ± 7 nm laterally, 387 ± 20 nm axially; GT: 118 ± 2 laterally, 368 ± 29 nm axially, N = 10 slices). **c, d**) Quantitative SSIM and PSNR analysis for data shown in a) and b), respectively; means and standard deviations obtained from N = 12 subvolumes. Scale bars: 1 μm in the magnified insets in **a**), others are 3 μm.

