## [Peer Review File · Nature Methods]

Peer Review Information

Manuscript Title: Incorporating the image formation process into deep learning improves network performance

Corresponding author name(s): Yicong Wu, Huafeng Liu

Editorial Notes: None

Reviewer Comments & Decisions:

Decision Letter, initial version:

Dear Yicong; dear Hari,

Your Article entitled "Incorporating the image formation process into deep learning improves network performance in deconvolution applications" has now been seen by three reviewers, whose comments are attached. While they find your work of some potential interest, they have raised concerns which in our view are sufficiently important that they preclude publication of the work in Nature Methods.

We will consider looking at a revised manuscript only if further experimental data allow you to address all the major criticisms of the reviewers (unless, of course, something similar has by then been accepted at Nature Methods or appeared elsewhere). This includes submission or publication of a portion of this work somewhere else.

The required new experiments and data include, but are not limited to, better justification/clarification of your method design, showing more general applicability beyond LSFM data, and improved artifact assessment. It would also be nice to have a better sense of what really is interpretable about the approach. With regards to concerns about novelty, we ask that you ensure all relevant literature is appropriately cited and discussed, and the conceptual and practical benefits of your approach clearly delineated. We hope you understand that until we have read the revised paper in its entirety we cannot promise that it will be sent back for peer-review.

If you are interested in revising this manuscript for submission to Nature Methods in the future, please contact me to discuss your appeal before making any revisions. Otherwise, we hope that you find the reviewers' comments helpful when preparing your paper for submission elsewhere.

Sincerely,
Rita

Rita Strack, Ph.D.
Senior Editor
Nature Methods

Reviewers' Comments:

Reviewer #1:

Remarks to the Author:

The manuscript proposes an architecture that enables fast 3D deconvolution of fluorescent microscopy images. This is achieved by means of a modified Richardson Lucy (RL) deconvolution algorithm, where the basic operations required in the algorithm like forward/backward projection and division are substituted by convolutional layers and other blocks. This allows shorter computation times (4-50 fold decrease) compared to traditional RL deconvolution, and robustness on samples from other domains, for which traditional data-driven approaches typically fail.

Comments:

Overall, this manuscript demonstrates an interesting contribution to deconvolution in microscopy, namely improving computation times by using fast inferencing of neural network-based algorithms. While the particular proposed method appears to contain novel contributions, the overall idea has been implemented before [1,2]. As such I do not judge this manuscript to be of sufficient novelty for the targeted journal Nature Methods.

[1] DEEP-URL: A Model-Aware Approach to Blind Deconvolution Based on Deep Unfolded Richardson-Lucy Network, Agarwal et al., 2020

[2] Deep Unfolding Network for Image Super-Resolution, Zhang et al., 2020

Major comments:

1. The authors mention parameter tuning as a disadvantage of data-driven methods and imply that the proposed algorithm does not suffer from this. However, as their method is built on a data-driven architecture (e.g. convolutional layers, activation functions, etc.), there are indeed parameters to be tuned, such as learning rate, batch size, decay rate/step, etc. Which are mentioned but not ablated in the manuscript.

2. In the general design criteria, strong assumptions are made which should have a quantitative impact analysis on the quality of the reconstructions. Such as:

- The impact of using H1 and H2 in the reconstruction process is not analyzed. Could their individual usage outperform the final model in time/quality. The authors did a proper ablation for other design criteria, like, comparing against the RLN-a, where the division and update steps were removed, showing their importance to the algorithm.
- The weights chosen for mixing the GT and the image in eq. (3) seem arbitrary. From a human point of view it makes sense, but without a quantitative analysis a 0.8 and 0.2 weight choice is unfounded.

Minor comments:

3. The architecture figures (Supplementary Fig. 1 and 9) suffer from:

- Text overlapping lines.
- Lines going behind objects. (See E1 and E2 in H3)
- Mixture of fonts and sizes used (C_AVE vs I_{ap} for example)
- The concatenation operation is confusing, as it should be the input to the third layer, and in the image seems like a second input coming from the top.

4. Supplementary Fig. 2 could be improved by adding a convolution symbol indicating the convolution of the PSF and the input object.

5. Figures with black and white images could be improved by using text in a color other than Black/white. Currently the text gets lost in some cases. Also, a text-box could be employed.

6. Missing references regarding deep learning models for super-resolution, denoising, and deconvolution (Isotropic reconstruction of 3D fluorescence microscopy images using convolutional neural networks, Weigert et al., Deep learning-enhanced light-field imaging with continuous validation, Wagner et al., Learning to Reconstruct Confocal Microscopy Stacks From Single Light Field Images, Page et al., etc.).

7. What does: "although the generalized prediction was artificially sharpened compared to the

conventional result. Notably" mean in section "Performance of RLN vs. CARE, RCAN and DDN on simulated data"?

Reviewer #2:

Remarks to the Author:

In this manuscript Li et al. demonstrate a novel approach to image deconvolution in which a convolutional neural network is applied to the forward and backward projection processes used in classical iterative deconvolution. As compared with conventional iterative deconvolution techniques this approach has the advantage of eliminating the need for manual selection of parameters, and arbitrary stopping criteria.

The authors present a very rigorous, thoroughly documented case for the superiority of the "Richardson-Lucy Network" (RLN) over conventional Richardson-Lucy Deconvolution (RLD), with respect resolution, signal linearity and generalization. Results are shown for both synthetic data as well real image volumes collected by single-view lightsheet microscopy, where results are compared with "ground truth" images obtained by deconvolved dual-view lightsheet microscopy. While the improvement frequently appears subtle in the images, the authors include clever quantitative metrics demonstrating the improvement obtained from the RLN approach.

Comparisons of the RLN technique with other deep-learning deconvolution approaches are more of a mixed bag. With the exception of an analysis of a human brain phantom, differences in deconvolution results are generally subtle, even in the quantifications. Training time is much reduced in RLN as compared with RCAN, but similar to DDN and CARE.

The use of images obtained using dual view lightsheet for ground truth of images collected by single-view is clever, but raises a few questions. First, nearly all of the results presented here were obtained from images obtained by light-sheet microscopy, a technique that is not widely used. While a few examples of results obtained from wide-field microscopy are shown, no rigorous analysis was applied. How well will the deconvolution approaches demonstrated here for light-sheet microscopy images work for images collected by other, more conventional approaches? Second, using deconvolutions of images obtained by dual-view lightsheet microscopy as ground truth for evaluations of results obtained by different deconvolution sounds circular, particularly when used as ground truth for the dual-RLN approach applied for deconvolution of dual-view images. Please explain. As these "ground truth" volumes are not used for quantitative comparisons, it may suffice to simply identify them by a different term. Also, please provide clear descriptions for how all "ground truth" images are obtained for real microscopy image volumes (legends to Figures 2 and 3, Supplementary Figures 10, 12, 13 and 16).

Minor points –

1. Figure 8a shows a comparison of processing times obtained using RLN versus the “previous method”. Please clarify in the figure legend and in the accompanying text the specifics of each pipeline.
2. Can the authors comment on the performance of the RLN in large, cleared volumes. How variable are the results spatially? Does the RLN approach help to address the problem of spatial variability in the point-spread function?
3. How does one evaluate quality in absence of ground truth. In some cases, “sharper” is interpreted as better, but the authors also refer to a result that is “over-deconvolved”.
4. In the course of characterizing different strengths, the manuscript presents an enormous number of images. While including all of the images demonstrates impressive rigor, the number of “uncompelling” images detracts from the main points of the manuscript. The authors should consider reducing the number of images to those that best demonstrate the salient points, and provide better explanations of the images in the legends and text.

Reviewer #3:

Remarks to the Author:

The authors present a deep learning based solution for deconvolution, which falls under the scope of physics-based/physics-aided artificial intelligence. They show extensive results on a variety of samples and good generalizability across samples and microscopes. Supplementary figures are vital for appreciating the advantage on a variety parameters such as noise, etc.

The scope of creating impact through physics-based artificial intelligence is well-tapped here. Therefore, even though the idea of using deep learning for any image translation problem seems a trivial trend today, the approach of including physics in the neural network is indeed interesting, even if not categorically novel. The authors themselves, unsurprisingly, refrain from claiming strong novelty and instead focus on the impact of the method (less number of parameters, generalizability, etc.). The extent of results provided, both synthetic and experimental, is impressive to convey it is a large and painstakingly created body of work.

However, despite all the positive qualities mentioned here, I am afraid that I have major concerns about the impact of the work matching the expectations of Nature Methods. I present the scientific concerns below in a numbered list. After that, I present other concerns.

1. The authors used RLD deconvolution (if I understood correctly) as the ground truth for training, except in the case of synthetic datasets. Therefore, this method is only as good as RLD deconvolution for experimental datasets. The authors have not provided justification or insight into why this was used as the ground truth for the experimental data. Could authors have explored other ground truth mechanisms? Or, maybe the objective is to perform Richardson Lucy Deconvolution without using it and not necessarily to obtain sharper or more optically cleared images. If this is the case, there is not a flaw in the objective except a significantly limited impact. It also conflicts with the authors claiming that RLN performs better than RLD for synthetic datasets. On the other hand, it is commendable that authors have explored using synthetic datasets for training, but both synthetic and experimental datasets for testing. Why not then use simulation/synthetic data-supervision as the core solution, in order to be not limited by the lack of ground truth in the experimental data?

2. No benchmarking beyond simulation results, RLD, and data driven methods. We do not know what artifacts RLN produces, which ones are attributed to the quality of ground truth, which ones to the architecture, and which ones to the learning process. We also do not see a discussion on the limitation of the technique or failure analysis in order to understand the scope in which the method creates utility. Experimental benchmarking using methods that support super-resolution like structured illumination microscopy and/or methods known as gold standard for benchmarking such as SEM is needed, even if for thin samples.

3. The recently released Thunder Imaging system of Leica also claims superior deconvolution using deep learning (in fact better deconvolution or optical clearance in comparison to conventional RLD, if I understood this correctly). How does RLN compare with Thunder Imaging in terms of quality of results and/or approach?

4. The authors discuss a lot about interpretability in the introduction, but do not address the interpretability of the method beyond using algorithm unrolling. Indeed the outputs of RLN FP1, BP1, FP2, BP2 are given in multiple figures and discussed in one paragraph on page 3, it is not a sufficient insight into interpretability. It would be more useful to explain how learning progresses through training, and what gets learnt or unlearnt in the process, how dependent it is on the data used for training, and why it generalizes over samples and imaging systems.

5. A minor comment on readability. This manuscript is replete with acronyms, right from the abstract. Quite a few acronyms are so close to each other, the readability is compromised in discerning one acronym from the other and recalling which one meant what.

While deconvolution is an important topic, especially for thick samples, I believe that the scope of the work is quite narrow, the novelty limited, and the validity and reliability needing more work. The robustness is demonstrated and the principle behind the work is interesting. I do acknowledge the

extensive efforts of the authors, but I feel that this work is more suited to other avenues, which are looking for new ways of making deep learning more powerful and application specific. My own suggestions would be Nature Machine Intelligence and/or IEEE TMI/TCI/TIP.

Author Rebuttal to Initial comments

Dear Reviewers,

Thank you for your helpful feedback and comments. We are glad that you recognize the novelty, impact, and rigor of our work. We appreciate your constructive suggestions, which we have thoroughly addressed with responses to your specific points followed by the **Re:** callouts and highlighted the changes with **dark red**. In the revised manuscript, we also mark the changes with **dark red**.

Thank you for reconsidering our revised manuscript.

Sincerely,

Yicong Wu and Hari Shroff (on behalf of all the authors)

NIBIB/NIH

Reviewers' Comments:

Reviewer #1:

Remarks to the Author:

The manuscript proposes an architecture that enables fast 3D deconvolution of fluorescent microscopy images. This is achieved by means of a modified Richardson Lucy (RL) deconvolution algorithm, where the basic operations required in the algorithm like forward/backward projection and division are substituted by convolutional layers and other blocks. This allows shorter computation times (4-50 fold decrease) compared to traditional RL deconvolution, and robustness on samples from other domains, for which traditional data-driven approaches typically fail.

Comments:

Overall, this manuscript demonstrates an interesting contribution to deconvolution in microscopy, namely improving computation times by using fast inferencing of neural network-based algorithms. While the particular proposed method appears to contain novel contributions, the overall idea has been implemented before [1,2]. As such I do not judge this manuscript to be of sufficient novelty for the targeted journal Nature Methods.

[1] Deep-URL: A Model-Aware Approach to Blind Deconvolution Based on Deep Unfolded Richardson-Lucy Network, Agarwal et al., 2020

[2] Deep Unfolding Network for Image Super-Resolution, Zhang et al., 2020

Re: We are glad the reviewer recognizes that our work contains novel contributions, and we appreciate the pointer to this prior and related work which we now cite in the revised manuscript. In the revision, we have included **Supplementary Note 1** that compares RLN, Deep-URL (method used in the first reference) and USRNet (method used in the second reference), clarifying the key advantages and differences of RLN vs. this prior work. In the discussion, in lines 343-345 on pages 10, we have added a sentence, “**Distinct from previous methods based on algorithm unrolling^{18, 19}, RLN enables 3D applications, spatially varying deconvolution, and does not require an iteration number to be specified (Supplementary 10g, Supplementary Note 1).**” In more detail:

Both USRNet and Deep-URL do share similarities with RLN: (1) all three approaches are based on ‘deep unfolding’, i.e., the algorithm unrolling framework; (2) they all incorporate model-based formulae into the learning-based method to attempt to bridge the gap between learning-based methods and model-based methods. However, we also believe that RLN is quite different from the two prior methods, and its architecture possesses intrinsic novelty, enabling us for example to demonstrate class leading performance in 3D applications, which is not demonstrated with the prior methods, perhaps due to memory issues with these architectures.

USRNet is a single end-to-end trained network designed to handle super-resolution tasks with different scale factors, blur kernels, and noise levels. As we summarize in Figure 1, the architecture of USRNet includes three core components: data modules, prior modules, and a hyper-parameter module. The USRNet alternates between the data module and the prior module using K deconvolution iterations, where K was empirically set to 8 to balance speed and accuracy. Unlike RLN, we emphasize that K is a parameter that must be determined in advance for each application – like classic deconvolution. The data module contains no trainable parameters, but requires a scale factor (i.e., another parameter) and a blurring kernel as input. The prior module integrates residual blocks into a U-Net architecture and uses the result of the data module and the noise level map as input. The hyper-parameter module consists of three fully connected layers and controls the outputs of the data module and prior module by varying the scale factor and noise level.

This architecture leads to clear disadvantages compared to RLN. First, USRNet is only designed for 2D images. It would be challenging to extend USRNet to 3D, which would require more GPU memory and computational overhead. Since USRNet is heavily reliant on FFT and inverse FFT operations in each data module, using a prior module based on the UNet architecture will introduce considerably more computational burden if using a 3D UNet. For example, in our manuscript we showed that the content-aware image restoration network (CARE) architecture (based on a 3D UNet and including two downsample levels) is 4x slower than RLN. USRNet requires three downsample levels, suggesting the prior module in 3D USRNet will be even more computationally expensive than CARE, and thus much slower than RLN. Second as mentioned above, USRNet requires a tunable parameter K to provide reasonable results. Third,

the blurring kernel must be known and fed into the network, limiting the usage of USRNet to applications in which the PSF is constant over the field of view ('spatially invariant PSF'). Last, the computational burden even for 2D applications is high, for example 2 days were required to train a 2D network model using 4 Nvidia Tesla V100 GPUs, for the parameters used in the published work (training iterations $\sim 2 \times 10^5$, training images size of 96×96 , and batch size set to 128). By contrast, RLN performs 3D image deconvolution without needing to explicitly provide (or tune) either a blurring kernel or an iteration number, and has achieved rapid training and processing time much faster than USRNet on 3D data (e.g., ~ 2 -3 hours using a single Nvidia GeForce GTX 1080 Ti GPU, training iteration number is $\sim 2 \times 10^4$, training volumes sized of $64 \times 64 \times 64$ with batch size set to 4), confirming this reviewer's comment that "RLN....., improves computation times by using fast inferencing of neural network-based algorithms".

Deep-URL is designed for 2D blind deconvolution, aiming to predict both the deconvolved image and the blurring kernel. Deep-URL (Figure 2) is motivated by the Richardson-Lucy algorithm like RLN (Figure 3), but Deep-URL defines the number of network layers exactly as the deconvolution iteration number K (e.g., 2, 5 in the published work) in the Richardson-Lucy formula at iteration k ($k = 1, 2, \dots, K$):

$$H^{k+1} = \sigma \left(\text{ReLU} \left(\left[\frac{y}{\text{ReLU}(x^k * W_H^k)} \right] * x^{k\dagger} \right) \times W_H^k \right)$$

$$x^{k+1} = \sigma \left(\text{ReLU} \left(\left[\frac{y}{\text{ReLU}(W_x^k * H^{k+1})} \right] * H^{k+1\dagger} \right) \times W_x^k \right).$$

The learning parameters W_H^k and W_x^k in the k -th layer (deconvolution iteration) have the same size as the blurring kernel H^{k+1} and input image x^{k+1} . Although the computational burden is never specified in the paper, we suspect the large size of these parameters combined with the necessary convolutions very likely contribute to a long training time, perhaps infeasibly long for 3D applications. Another concern is the application of Deep-URL in tasks that require a spatially varying PSF, as Deep-URL outputs only a single PSF. In addition, since Deep-URL was only demonstrated on simulated data; its performance on experimental data is unknown. RLN differs from Deep-URL in the following aspects: (1) for most fluorescence microscopes, the point spread function can be measured or modelled, and thus RLN does not need to predict the blur kernel explicitly, which simplifies network architecture; (2) RLN was motivated by our improved Richardson-Lucy algorithm, i.e., using an unmatched backprojector [3], and can achieve resolution-limited results with only 1 deconvolution iteration so there is no need to specify or tune an deconvolution iteration number K ; (3) RLN makes use of many small $[3 \times 3 \times 3]$ convolution kernels to perform learning, thereby rapidly and effectively performing 3D deconvolution.

In summary, we emphasize that the architecture of RLN is unique compared to USRNet and Deep-URL, and that this unique architecture provides notable advantages including extension to 3D imaging, the absence of a need to specify an explicit deconvolution iteration number in the network design, and considerably less computational burden than the other two networks, despite handling larger (3D) data. In addition, we have now confirmed that that RLN can handle images with spatially varying blurring (Figure 4 below, **Supplementary Fig. 10g** in the revision, please see more details on our response to Reviewer #2 about this capability), unlike USRNet and Deep-URL. Beyond the novelty of the RLN architecture, we also

think there is novelty in the biological applications we demonstrate, which are absent in USRNet and Deep-URL. Unlike these approaches, we have demonstrated generalization, interpretability, and clear improvements in deconvolution compared to RLD. We suspect the ability to rapidly process large cleared-tissue data is especially important, given its increasing prevalence in biological studies.

Figure 1. The overall architecture of USRNet with the data module, prior module, and hyper-parameter module emphasized. In addition to the typical parameters required for deep learning (e.g., learning rate, batch size, decay rate, etc.), this architecture also requires the following tunable parameters (red): the deconvolution iteration number K for constructing the data and prior module, the blurring kernel, noise level, and scale factor. $k = 1, 2, \dots, K$ represent intermediate iterations.

Figure 2. The architecture of Deep-URL for model-aware blind deconvolution. Given a blurred image y and initial estimates of the clean image x^0 and blurring kernel H^0 , the model updates x^k and H^k . The deconvolution iteration number K (red) must be chosen as it is used to determine how many layers are needed. $k = 1, 2, \dots, K$ represent intermediate iterations.

Figure 3. The architecture of RLN for 3D deconvolution, consisting of three parts: down-scale estimation starting from the average-pooled input image, $H1$; original-scale estimation starting with original-scale input image, $H2$; and merging/fine-tuning, $H3$. $H1$ and $H2$ are inspired by the RL deconvolution update formula, which mimic the unmatched forward/back projector steps. $H1$ is used to increase the field of view and decrease the computational burden. $H2$ is used to provide information to assist $H1$. RLN only needs the learning-based parameters typical for any data-driven network; there are no model parameters to adjust.

Figure 4. (now **supplementary Fig. 10g** in the revision) *C. elegans* embryos expressing GCaMP3 from a *myo-3* promoter (labeling muscles) were imaged by reflective diSPIM [4]. In this technique, multiple views of the sample are acquired simultaneously, but each raw view is contaminated by spatially varying blur (left), which can be removed by modeling the spatially variant PSF in the deconvolution process. From left to right: maximum-intensity projections of raw data, deconvolution based on spatially invariant PSF, deconvolution based on spatially variant PSF (i.e., the ground truth 'GT'), single-input RLN, and dual-input RLN predictions (using two orthogonal views as input) are shown for lateral (top) and axial (bottom) views. The raw input is contaminated with spatially varying epifluorescence background (red arrows). As shown, spatially variant deconvolution, single-input RLN and dual-input RLN remove associated epifluorescence contamination, enhancing resolution and contrast. PSNR and SSIM analysis confirmed this result (Raw: PSNR 22.65 ± 1.46 , SSIM 0.52 ± 0.04 ; spatially invariant deconvolution: PSNR 26.61 ± 1.53 , SSIM 0.62 ± 0.02 ; single-input RLN: PSNR 31.56 ± 0.70 , SSIM 0.84 ± 0.02 ; dual-input RLN: PSNR 34.55 ± 0.61 , SSIM 0.87 ± 0.01 ; N = 12 volumes). Scalebar: 10 μm .

[3] Guo, M. et al. Rapid image deconvolution and multiview fusion for optical microscopy. *Nature Biotechnology* 38, 1337-1346 (2020).

[4] Wu, Y. et al. Reflective imaging improves spatiotemporal resolution and collection efficiency in light sheet microscopy. *Nat. Commun.* 8, 1452 (2017).

Major comments:

1. The authors mention parameter tuning as a disadvantage of data-driven methods and imply that the proposed algorithm does not suffer from this. However, as their method is built on a data-driven architecture (e.g. convolutional layers, activation functions, etc.), there are indeed parameters to be tuned, such as learning rate, batch size, decay rate/step, etc. Which are mentioned but not ablated in the manuscript.

Re: Respectfully, we think the reviewer might have misunderstood our meaning about parameter tuning. We did not once mention parameter tuning as a disadvantage of the *data-driven* (i.e., deep learning) methods. Rather, we emphasized the value of using data-driven methods as a means of bypassing the parameter tuning that is necessary in physics and model-based deconvolution methods (e.g., deconvolution iteration number, regularization parameters, and backprojectors that can dramatically change the deconvolution results and introduce artifacts if not tuned correctly). Indeed, in incorporating the physics-based RLD method into RLN, we carefully designed the latter so that it avoids the use of iteration number (note that Deep-URL and USRNet require this parameter). Like any data-driven architecture, RLN still requires some parameters (e.g., convolutional layers, activation functions, learning rate, batch size, decay rate/step) which mostly affect the speed of convergence during the training. For example, RLN model performance is only weakly dependent on the number of training iterations (Figure 5). We note that these data-driven parameters are determined once for training, and remain unchanged during application (i.e., conventional testing or generalization testing) of RLN models.

Figure 5. Quantitative analysis of different training iterations, using conventional testing with synthetic mixed structures ($N = 12$ volumes). The differences that result from using different iteration numbers are minor.

2. In the general design criteria, strong assumptions are made which should have a quantitative impact analysis on the quality of the reconstructions. Such as:

- The impact of using H1 and H2 in the reconstruction process is not analyzed. Could their individual usage outperform the final model in time/quality. The authors did a proper ablation for other design criteria, like, comparing against the RLN-a, where the division and update steps were removed, showing their importance to the algorithm.

- The weights chosen for mixing the GT and the image in eq. (3) seem arbitrary. From a human point of view it makes sense, but without a quantitative analysis a 0.8 and 0.2 weight choice is unfounded.

Re: We agree with the reviewer's point that our paper would be strengthened by better investigating the role of H1 and H2. We designed H1 to process downsampled input, thereby increasing the field of view accessed by each convolutional kernel. However, downsampling causes information loss – this point motivated us to use H2, working directly on the original-size input. Intuitively, the combination of H1 and H2 can thus provide better performance than only H1 or H2. To demonstrate this assertion, we used the synthetic mixed structures in a conventional testing comparison (training on similar structures as the test data, Figure 6). Visually, the combination of H1 and H2 outperforms either network alone. This is consistent with quantitative analysis on the conventional testing result: H1 alone, PSNR 26.4 ± 0.5 , SSIM 0.87 ± 0.03 ; H2 alone, PSNR 22.4 ± 0.4 , SSIM 0.74 ± 0.01 ; RLN, PSNR 30.2 ± 0.3 , SSIM 0.93 ± 0.01 , $N = 12$ volumes. Therefore, we conclude the combination of H1 and H2 (i.e., RLN) outperforms either network alone. In the revised manuscript, we have added a discussion in lines 94-103 on page 3 and included this result as **Supplementary Fig. 3a**. The new paragraph read as “To enhance network efficiency, we designed H1 with smaller feature maps and more convolutional layers for processing downsampled input, and H2 with larger feature maps and fewer convolutional layers to process the original-size input. With this combination, H1 increases the field of view accessed by each convolutional kernel, and H2 mitigates information loss due to downsampling in H1. By using a synthetic phantom object consisting of mixtures of dots, solid spheres, and ellipsoidal surfaces (**Supplementary Fig. 2**), we confirmed that the combination of H1 and H2 outperforms H1 or H2 alone (**Supplementary Fig. 3a**).”

Figure 6. (now **supplementary Fig. 3a** in the revision) Comparing networks using only H1, only H2, or RLN on synthetic mixed structures (conventional testing). Lateral (top) and axial (bottom) views are shown. The combination of H1 and H2 provide better performance than using H1 or H2 alone (yellow arrows). Scalebar: 10 pixels.

Regarding the weights chosen for mixing the GT and the image in Eq. (3), we agree that it is helpful to clarify this choice. The weight ratio 0.8 vs 0.2 is indeed empirically chosen, but it is not the only choice that yields good results. We used generalization testing (training dataset on synthetic mixed structures and testing on a simulated human brain phantom) to quantify the differences among different weight selections (Figure 7). From the PSNR and SSIM values, we found the differences between different selections are small, but combinations of ground truth and input including 0.9 / 0.1, 0.8 / 0.2, and 0.7 / 0.3, provide slightly better performance than other combinations. In the revision, in lines 651-653 on page 24, we have added a sentence “The ratio 0.8 vs. 0.2 was empirically chosen, but we found that network output is only weakly dependent on this choice (e.g., 0.9 vs. 0.1 and 0.7 vs 0.3 work well also and are slightly better than 1.0 vs. 0.0).”

Figure 7. Quantitative analysis of using the combination of ground truth and input with different weighting ratios to supervise H1 learning. Some weight ratios (e.g., 0.9 vs 0.1, 0.8 vs 0.2, 0.7 vs 0.3) provide better results than other weight ratios, but the difference is not large. Experiments are repeated 3 times.

Minor comments:

3. The architecture figures (Supplementary Fig. 1 and 9) suffer from:

- Text overlapping lines.
- Lines going behind objects. (See E1 and E2 in H3)
- Mixture of fonts and sizes used (C_AVE vs I_[1] for example)
- The concatenation operation is confusing, as it should be the input to the third layer, and in the image seems like a second input coming from the top.

Re: Thanks for pointing out the issues. In the revision, we have modified **Supplementary Figs. 1 and 9** as suggested.

4. Supplementary Fig. 2 could be improved by adding a convolution symbol indicating the convolution of the PSF and the input object.

Re: Thanks for the suggestion. In the revision, we have added the convolution operation and rearranged **Supplementary Fig. 2**.

5. Figures with black and white images could be improved by using text in a color other than Black/white. Currently the text gets lost in some cases. Also, a text-box could be employed.

Re: Thanks for the suggestion. In the revision, we have modified some text and changed the color so that the display has been improved, e.g., **Supplementary Figs. 3, 10.**

6. Missing references regarding deep learning models for super-resolution, denoising, and deconvolution (Isotropic reconstruction of 3D fluorescence microscopy images using convolutional neural networks, Weigert et al., Deep learning-enhanced light-field imaging with continuous validation, Wagner et al., Learning to Reconstruct Confocal Microscopy Stacks From Single Light Field Images, Page et al., etc.).

Re: Thanks for the suggestion. Regarding the paper (Isotropic reconstruction of 3D fluorescence microscopy images using convolutional neural networks, Weigert et al, <https://arxiv.org/pdf/1704.01510.pdf>), in our previous version we cited their peer-reviewed publication in *Nature Methods* (Weigert, M. et al. Content-aware image restoration: pushing the limits of fluorescence microscopy. *Nature Methods* **15**, 1090-1097, 2018). In the revision, we have cited the other two references in line 54-55 on page 2 (references # 13-14).

7. What does: "although the generalized prediction was artificially sharpened compared to the conventional result. Notably" mean in section "Performance of RLN vs. CARE, RCAN and DDN on simulated data"?

Re: We meant that in the study of linearity (**Supplementary Fig. 7**), the generalized prediction results on the simulation beads data were visually sharper than the ground truth. To clarify our assertion, in lines 153-158 on page 5 in the revision, we have changed the sentences to “**RLN under both conventional testing and generalization paradigms recovered axial views distorted by the PSF and provided better linearity than RLD (Supplementary Fig. 7). Although the generalized RLN prediction was artificially sharpened compared to the conventional result, RLN still offered better generalization than CARE, RCAN and DDN, which all showed more obvious visual distortions (red and yellow arrows, Fig. 1d) and lower SSIM and PSNR (Fig. 1g).**”

Reviewer #2:

Remarks to the Author:

In this manuscript Li et al. demonstrate a novel approach to image deconvolution in which a convolutional neural network is applied to the forward and backward projection processes used in classical iterative deconvolution. As compared with conventional iterative deconvolution techniques this approach has the advantage of eliminating the need for manual selection of parameters, and arbitrary stopping criteria.

Re: We thank the reviewer for the useful summary of our work.

The authors present a very rigorous, thoroughly documented case for the superiority of the “Richardson-Lucy Network” (RLN) over conventional Richardson-Lucy Deconvolution (RLD), with respect resolution, signal linearity and generalization. Results are shown for both synthetic data as well real image volumes collected by single-view light sheet microscopy, where results are compared with “ground truth” images obtained by deconvolved dual-view light sheet microscopy. While the improvement frequently appears subtle in the images, the authors include clever quantitative metrics demonstrating the improvement obtained from the RLN approach.

Re: We thank the reviewer for the summary of our results. We would like to point out that in our original manuscript, we also showed RLN’s application on image volumes collected by widefield microscopy (**Fig. 4** in the original submission, now **Fig. 5** in the revision, **Supplementary Figs. 14, 15**) and iSIM, a high-speed structured illumination microscope with optical sectioning similar to spinning disk confocal microscopy (**Fig. 3, Supplementary Figs. 12, 13, 16**). In the revision, we have provided even more evidence that RLN offers class leading performance in a greater variety of applications and microscopes:

(1) RLN performance given reflective light sheet microscopy data blurred with a spatially varying PSF (Figure 4 in this response, **Supplementary Fig. 10g** in the revised manuscript); in the revision we have added a paragraph in lines 235-245 on page 7: “Next, we applied single- and dual-input RLN to the challenging case of images contaminated by a spatially varying blurring function. Reflective diSPIM²⁸ images samples deposited on reflective coverslips, enabling the collection of additional specimen views that boost collection efficiency and spatiotemporal resolution compared to conventional diSPIM. The raw reflective data are contaminated by substantial epifluorescence that varies over the imaging field. This contamination can be removed by incorporating a spatially varying PSF into RLD, at the cost of considerable computational expense. To train RLN for this application, we used a published reflective diSPIM dataset imaging muscle GCaMP3 expression in late stage *C. elegans* embryos (26 image pairs, raw specimen views as input data, deconvolution with a spatially varying Wiener-Butterworth backprojector⁸ as the ground truth). We found that single-input RLN handles this complex deconvolution task well, and that dual-input RLN provides even better deblurring quality on par with the ground truth (**Supplementary Fig. 10g**).”

(2) RLN predictions on confocal input, after training on STED microscopy ground truth (Figure 8 in this letter, **Supplementary Fig. 17** in the revised manuscript); RLN predictions on widefield input, after training on 3D structured illumination microscopy (3D SIM) ground truth (Figure 9 in this letter, **Supplementary Fig. 18** in the revised manuscript). In the revision, in lines 320-338 on page 9, we have added a section “**RLN’s performance in super-resolution applications**” that describes the potential of RLN in applications with such super-resolution ground truth: “Having demonstrated RLN’s deconvolution capability using ground truth consisting of high SNR dual-view deconvolved light-sheet data, high SNR deconvolved iSIM data, high contrast confocal data, and synthetic ground truth, we next evaluated the extent to which RLN can predict super resolution images from diffraction-limited input. First, we evaluated confocal-to-STED

prediction^{11, 31}, using 25 pairs of confocal/STED volumes of U2OS cells stained with a primary antibody against Tomm20 and an anti-rabbit secondary antibody conjugated with Alexa Fluor 594 (marking mitochondria) to train RLN and RCAN models (**Supplementary Fig. 17**). Although both networks improved the resolution of the confocal input, RLN quantitatively outperformed RCAN. Second, we performed conventional RLN testing based on Jurkat cells expressing EMTB-3XGFP (a microtubule marker) and generalization testing on U2OS cells expressing Lamp1-EGFP (a lysosomal marker). RLN models were trained with 46 EMTB-3XGFP volumetric datasets consisting of widefield input and 3D SIM ground truth. In both cases, output was visually and quantifiably closer to the 3D SIM ground truth than RLD (**Supplementary Fig. 18, Supplementary Table 2**). Collectively, these results further demonstrate the power of RLN for additional applications on diverse microscopes.

(3) RLN predictions on widefield data (Figure 12 in this letter, **Fig. 4** in the revised manuscript), showing clear performance improvements against state-of-the-art commercial deconvolution software, the Leica Thunder computational clearing method – as verified using confocal microscopy ground truth. The latter result is particularly exciting given that RLN was trained only on synthetic data. Please see more details from our response to Reviewer #3, or **Fig. 4**, lines 276-291 on page 8, and lines 887-893 on page 30 in the revised manuscript.

Comparisons of the RLN technique with other deep-learning deconvolution approaches are more of a mixed bag. With the exception of an analysis of a human brain phantom, differences in deconvolution results are generally subtle, even in the quantifications. Training time is much reduced in RLN as compared with RCAN, but similar to DDN and CARE.

Re: We agree with the assessment of training time, but disagree that performance is a mixed bag. In all comparisons across multiple data types, RLN performance outperforms other methods quantitatively, as shown in **Supplementary Table 2**. Even with the new tests included in this rebuttal, confocal improvement based on training with STED microscopy ground truth (Figure 9 in this letter, **Supplementary Fig. 17** in the revised manuscript), RLN outperforms RCAN. In addition, RLN has the advantage of being more interpretable than other deep learning approaches, which are purely ‘black boxes’.

The use of images obtained using dual view lightsheet for ground truth of images collected by single-view is clever, but raises a few questions. First, nearly all of the results presented here were obtained from images obtained by light-sheet microscopy, a technique that is not widely used. While a few examples of results obtained from wide-field microscopy are shown, no rigorous analysis was applied. How well will the deconvolution approaches demonstrated here for light-sheet microscopy images work for images collected by other, more conventional approaches?

Re: We disagree with some of the assertions in this statement. Light sheet microscopy is becoming increasingly prevalent, and is now often used to image live samples as well as large, cleared tissue

samples, which are both examples that we highlighted here to showcase the potential of RLN. We also note that we devoted an entire figure to data collected with iSIM, a rapid structured illumination microscope with optical sectioning capability (**Fig. 3**), as well as another figure on widefield microscopy (**Fig. 4** in the original submission, which is **Fig. 5** in the revised manuscript). For the iSIM volumes, the SSIM/PSNR analyses of RLN predictions vs. deconvolved, high SNR ground truth confirm that RLN performs well when attempting to deconvolve raw iSIM data. For the widefield volumes, although there was no ground truth, it was visually apparent that RLN outperformed direct Richardson-Lucy deconvolution. This is now quantitatively supported by new results (Figure 12 in this letter, **Fig. 4** in the revision), in which we verify that RLN applied to widefield data produces a prediction that is quantitatively closer to the confocal ground truth than RLD, and Thunder, a state-of-the-art commercial deconvolution package from Leica.

All this said, we appreciate the reviewer's broader question about how generally applicable the method is. To address this point, we have performed more tests on different biological structures that were imaged with additional types of microscopes. First, we use RLN to predict GCaMP3 fluorescence in muscle cells in *C.elegans* embryo acquired with reflective light sheet microscopy. The raw volumes are contaminated with a spatially varying PSF, but RLN can handle this difficult deconvolution task and deblurs the resulting volumes similarly to ground truth that has been deconvolved with a spatially varying PSF (Figure 4 in this letter, **Supplementary Fig. 10g** in the revised manuscript, and please see additional detail and discussion inline 235-245 on page 7 in the revised manuscript). Second, we evaluated the extent to which RLN can learn to predict super-resolution 3D SIM reconstructions from low resolution widefield data (Figure 8 in this letter, **Supplementary Fig. 18** in the revised manuscript, details in the "RLN's performance in super-resolution applications" section, page 9, lines 320-338). Specifically, we performed conventional RLN testing based on Jurkat cells expressing EMTB-3XGFP (a microtubule marker) and generalization testing on U2OS cells expressing Lamp1-EGFP (a lysosomal marker). RLN were trained with 46 EMTB-3XGFP datasets (widefield input and 3D SIM as ground truth) using 18400 iterations. In both cases, RLN produced an output that was visually and quantifiably more like the 3D SIM ground truth than RLD. Third, we evaluated the extent to which RLN can predict super resolution STED microscopy data from confocal input (Figure 9 in this letter, **Supplementary Fig. 17** in the revised manuscript). In these data (samples were U2OS cells stained with a primary antibody against Tomm20 and an anti-rabbit secondary antibody conjugated with Alexa Fluor 594, marking the outer mitochondrial membrane), RLN was trained with 25 confocal/STED pairs using 10000 iterations, and RCAN was trained using the same data, with 5 residual groups and 5 residual blocks. Although both networks improve resolution of the confocal input, quantitative analysis demonstrates that RLN outperforms RCAN. Collectively, these results further support the power of RLN and demonstrate its application to more microscopy types than the widefield, light-sheet, and iSIM data demonstrated in our initial manuscript.

Regarding the reviewer's comment on the dearth of 'rigorous analyses' in the widefield data (**Fig. 4** in the original submission, now **Fig. 5** in the revised manuscript), while no ground truth was available, we did show using both real and frequency space measures that we obtained quantifiably better resolution/clearer identification of known structures than RLD (e.g., frequency domain images shown in **Fig. 5** and **Supplementary Fig. 14**, and intensity profiles shown in **Supplementary Fig. 15**). To further

support our claim that RLN outperforms RLD in widefield imaging, we have acquired two new datasets (U2OS cells stained with Alexa Fluor 568 Phalloidin, marking actin; and Cos-7 cells immunolabeled with primary mouse anti-Nup clone Mab414 and goat-anti-mouse IgG secondary antibody conjugated with Star635P, marking nuclear pore complexes). We imaged the same samples first with a Leica widefield microscope and then with a Leica confocal microscope, both with the same 63x/1.40 oil immersion objective lens. We registered and compared the raw widefield images, RLN's generalization test results (the model was trained on synthetic data and never 'saw' the experimental data during training), and the confocal ground truth (Figure 12 in this letter, **Fig. 4** in the revised manuscript). We also compared the RLN results with the Leica Thunder computational clearing method. Visual inspection and quantitative analysis of SSIM and PSNR demonstrate RLN outperforms RLD and Thunder, especially in the axial dimension, providing an output closer to the confocal ground truth. Please see this discussion in lines 276-291 on page 8 in the revision.

Figure 8. (now **supplementary Fig. 17** in the revision) RLN outperforms RCAN when attempting confocal-to-STED cross modality prediction. a) Lateral views (top) and axial views (bottom) of U2OS cells immunolabeled with a primary antibody against Tomm20 and an anti-rabbit secondary antibody conjugated with Alexa Fluor 594, comparing the raw input collected by confocal microscopy, STED microscopy images (ground truth, GT), RCAN output, and RLN output. b) Line profiles across the yellow and red lines in the lateral view and axial view in a, demonstrating that RCAN and RLN improve resolution compared to the input, yet not to the extent of the ground truth. c) SSIM and PSNR analysis for data shown in a), means and standard deviations are obtained from $N = 10$ slices. Scale bars: $4\ \mu\text{m}$.

Figure 9. (now **supplementary Fig. 18** in the revision) RLN outperforms RLD when attempting to restore widefield data based on 3D SIM ground truth, under both conventional and generalization testing. a) Lateral maximum projection (top), axial maximum projection (middle), and Fourier spectra (bottom) of Jurkat cells expressing EMTB-3XGFP, comparing the raw input collected by widefield microscopy, RLD with

40 iterations, 3D SIM (i.e., the ground truth), and the conventional test result of RLN. Decorrelation analysis confirms the resolution improvement of RLN compared to RLD (raw: 348 ± 20 nm laterally, 798 ± 133 nm axially; RLD: 272 ± 9 nm laterally, 621 ± 120 nm axially; RLN: 156 ± 7 nm laterally, 414 ± 11 nm axially; GT: 117 ± 1 nm laterally, 385 ± 10 nm axially, $N = 15$ slices). Magnified insets corresponding to the red rectangular region indicate that RLN output fails to predict some dim filaments. b) Lateral maximum intensity projection (top), axial maximum intensity projection (middle), and Fourier spectra (bottom) of U2OS cells expressing Lamp1-EGFP, comparing the raw input collected by widefield microscopy, RLD with 40 iterations, 3D SIM (ground truth), and the generalization test result of RLN (i.e., the model is the same one as used in a), trained on Jurkat cells expressing EMTB-3XGFP). Quantitative resolution analysis by decorrelation methods shows the resolution improvement of RLN compared to RLD (raw: 368 ± 22 nm laterally, 837 ± 66 nm axially; RLD: 245 ± 7 nm laterally, 641 ± 109 nm axially; RLN: 146 ± 7 nm laterally, 387 ± 20 nm axially; GT: 118 ± 2 nm laterally, 368 ± 29 nm axially, $N = 10$ slices). c, d) Quantitative SSIM and PSNR analysis for data shown in a) and b), respectively; means and standard deviations are obtained from $N = 12$ subvolumes. Scale bars: 1 μm in the magnified insets in a), others are 3 μm .

Second, using deconvolutions of images obtained by dual-view lightsheet microscopy as ground truth for evaluations of results obtained by different deconvolution sounds circular, particularly when used as ground truth for the dual-RLN approach applied for deconvolution of dual-view images. Please explain. As these “ground truth” volumes are not used for quantitative comparisons, it may suffice to simply identify them by a different term. Also, please provide clear descriptions for how all “ground truth” images are obtained for real microscopy image volumes (legends to Figures 2 and 3, Supplementary Figures 10, 12, 13 and 16).

Re: We thank the reviewer for raising this point. When performing dual-view light sheet imaging, we did use the high SNR dual-view deconvolved as “ground truth” in evaluating the success of RLN, since we did not have access to the true object structure. We don’t think that doing so is circular. Given the dual-view deconvolved ground truth (produced by RLD), we were able to verify the extent to which RLN can reproduce dual-view deconvolution produced by RLD, given either single- or dual-view inputs to the network. For single-view input, the RLN prediction displayed better resolution in the axial view than direct single-view Richardson-Lucy deconvolution, and closely resembled the dual-view ground truth when the sample was relatively thin or transparent. For densely labeled and scattering samples, we found that dual-input RLN generates more nearly isotropic reconstructions. These conclusions were assessed visually, but also by computing the SSIM and PSNR values between the RLN results and dual-view joint deconvolution “ground truth” – which are quantitative comparisons either presented in the main text or in figure legends to **Fig. 2** and **Supplementary Figs. 10, 11**.

Nevertheless, we do agree that our description of “ground truth” ought to be clarified further – particularly as it varies for different results presented in this paper. The “ground truth” used in the original manuscript consisted of high SNR dual-view deconvolved light-sheet data, high SNR deconvolved iSIM

data, and synthetic ground truth. In the new results provided in this revision, we gathered higher contrast or higher resolution ground truth (e.g., confocal, 3D SIM, and STED microscopy data). These provide closer approximations to the true object structure than the input data, so we used them to train the models and evaluate the RLN predictions in the new data we present here. To better describe the “ground truth”, in the revision, we have expanded **Supplementary Table 1** to clearly delineate what “ground truth” was used in training for every result presented in the paper and also added a text description in **Methods** (lines 729-735 on page 26), “Since we did not have access to the true object structure when evaluating the performance of RLN, we used a variety of “ground truth”, consisting of high SNR dual-view deconvolved light-sheet data (**Fig. 1b, Fig. 2, Supplementary Figs. 8, 10, 11**), synthetic ground truth (**Fig. 1d, e, Supplementary Figs. 3-7, 16a**), high SNR deconvolved iSIM data (**Fig. 3, Supplementary Figs. 12, 13, 16b-f**), higher contrast confocal data (**Fig. 4**), super-resolution STED microscopy data (**Supplementary Fig. 17**), and super-resolution 3D SIM data (**Supplementary Fig. 18**). Further details (i.e., training ground truth, training pair number, testing type) of training and test datasets are summarized in **Supplementary Tables 1 and 3.**”

We also checked the descriptions in the legends to the figures: (1) in the legend to **Fig. 2**, we mentioned that the raw input images were of live U2OS cells transfected with mEmerald-Tomm20, acquired with symmetric 0.8/0.8 NA diSPIM, and cleared brain tissue slab, acquired by cleared-tissue immersion 0.4/0.4 NA diSPIM, and specified that the “ground truth” were dual-view deconvolved light-sheet data; (2) in the legend to **Fig. 3**, the raw input images were of live U2OS cells expressing mEmerald-Tomm20-C-10, acquired with iSIM, and the ground truth were deconvolved iSIM data; (3) in the legend to **Supplementary Fig. 16**, we mentioned that the raw input images were of live U2OS cells expressing ERmoxGFP, acquired with iSIM, and the ground truth was high SNR deconvolved data; (4) in the legend to **Supplementary Fig. 10**, the images were of a live *C. elegans* embryo collected by diSPIM and processed with single-input RLN and dual-input RLN. In the revision, we have clarified that the ground truth was the dual-view deconvolved result; (5) in the legend to **Supplementary Figs. 12, 13**, the images were of a U2OS cell expressing GalT-GFP, mEmerald-Tomm20-C-10, and Lamp1-EGFP, acquired with iSIM, and in the revision we have clarified that the ground truth was the deconvolved iSIM data.

Minor points –

1. Figure 8a shows a comparison of processing times obtained using RLN versus the “previous method”. Please clarify in the figure legend and in the accompanying text the specifics of each pipeline.

Re: In the revision, we have changed the sentence in the legend to **Supplementary Fig. 8** from “a) Processing time of RLN vs. previous processing pipeline (i.e., cropping, registration, joint deconvolution, and stitching) on differently sized volumes.” to “a) Processing time of RLN vs. previous processing pipeline on differently sized volumes. In the RLN pipeline, the large raw single-view input volume is cropped into subvolumes, RLN is applied to each subvolume, and the predictions on each subvolume are stitched back into a single volume. The previous pipeline (1) downsamples the large raw dual view volumes, performing affine registration of the downsampled volumes to obtain a transformation matrix; (2) crops the raw dual

view volumes into multiple subvolumes using the transformation matrix to guide the cropping coordinates; (3) applies an affine registration to the dual-view subvolumes; (4) performs joint deconvolution on the registered dual-view subvolumes with the Wiener-Butterworth backprojector; (5) stitches the deconvolved subvolumes back into a single volume.” In the main text, in lines 213-214 on page 6, we changed the description to “... over the previous processing pipeline⁸ (i.e., coarse registration, cropping, fine registration, joint deconvolution with Wiener-Butterworth backprojector, and stitching) for cleared-tissue diSPIM data restoration.”

2. Can the authors comment on the performance of the RLN in large, cleared volumes. How variable are the results spatially? Does the RLN approach help to address the problem of spatial variability in the point-spread function?

Re: We thank the reviewer for mentioning our work on large, cleared volumes, which we believe demonstrates the promise of RLN. Processing large, cleared tissue volumes using physical model-based methods (i.e., Richardson-Lucy deconvolution) is a time-consuming task. As we demonstrated in our manuscript, RLN can reduce the processing time by ~4-6 fold while simultaneously reducing artifacts induced by registration failures. To evaluate the variability of the RLN prediction in this large dataset, we cropped the entire dataset (**Fig. 2g** in the main text) into 300 subvolumes (each 600 x 600 x 600 voxels) and used decorrelation analysis to calculate the lateral and axial resolutions of each volume (Figure 10 in this response, **Supplementary Fig. 8d** in the revised manuscript). Although the lateral and axial resolution estimated by this method fluctuate (688 ± 51 nm laterally, 701 ± 60 nm axially, $N = 300$ subvolumes), the variation is within 10%. In the revision, in lines 218-220 on page 6, we have added a sentence: “Finally, we verified that RLN predictions displayed good performance over the entire volume (less than 10% variation: 688 ± 51 nm laterally, 701 ± 60 nm axially, as assed with decorrelation analysis²⁷ over $N = 300$ subvolumes, **Supplementary Fig. 8d**).”

We also examined a distinct example in which the PSF obviously varies across the field of view, finding that RLN can handle this case also. Reflective diSPIM [4] enables improvements in spatiotemporal resolution compared to conventional diSPIM, but the raw reflective data are contaminated by substantial epifluorescence that varies over the imaging field and that can (using classic methods) only be removed by incorporating a spatially varying PSF into the deconvolution. Here we used 26 training data pairs (the raw specimen views as inputs and the previously published Wiener-Butterworth deconvolution [3] result as the ground truth) to train a single-input RLN model and a dual-input RLN model with 39000 iterations (~8h of training time), holding out 12 volumes for testing. As shown in Figure 4 in this response (**Supplementary Fig. 10g** in the revised manuscript), the output from single-input RLN and dual-input RLN both outperform conventional RLD with a spatially invariant PSF, providing visually and quantitatively improved deconvolution closer to the ground truth (Raw data: PSNR 22.7 ± 1.5 , SSIM 0.52 ± 0.04 ; spatially invariant deconvolution: PSNR 26.6 ± 1.5 , SSIM 0.62 ± 0.02 ; single-input RLN: PSNR 31.6 ± 0.7 , SSIM 0.84 ± 0.02 ; dual-input RLN: PSNR 34.6 ± 0.6 , SSIM 0.87 ± 0.01 ; $N = 12$ volumes). More details can be found in

the section “RLN for deconvolution on volumes contaminated with scattering and spatially varying blur” (lines 235-245 on page 7) in the revised manuscript.

Figure 10. (now **supplementary Fig. 8d** in the revision) Estimating variability of resolution in RLD predictions from over 300 subvolumes (each 600 x 600 x 600 voxels) cropped from the large, cleared tissue in main text Fig. 2g dataset. (a) Lateral and (b) axial resolution estimates from decorrelation analysis.

3. How does one evaluate quality in absence of ground truth. In some cases, “sharper” is interpreted as better, but the authors also refer to a result that is “over-deconvolved”.

Re: We agree that this is a fair point. In the absence of ground truth (e.g., the widefield data shown in **Fig.**

4 in the original submission, now **Fig. 5** in the revision), we evaluated image quality visually (e.g., less out of focus contamination), via line intensity profiles, and via spatial frequency content. We do interpret “sharper” as better, if the recovered structures are expected and within the resolution limit. For example, in the main text, in line 121-122 on page 4 (3rd paragraph on page 3 in the original submission), we stated that “In all examples, we also noticed that RLN produced sharper axial views than RLD.” These examples include **Fig. 1b** and **Supplementary Figs. 4,5**. In **Fig. 1b**, the axial resolution of the network output was improved relative to RLD, producing more nearly isotropic reconstructions closer to the high SNR dual-view deconvolved light-sheet data (i.e., the ground truth used in training the network). Similarly, in **Supplementary Figs. 4, 5**, RLN produced a ‘sharper’ axial view, more similar to the isotropic resolution of the ground truth used in training the network than the RLD output.

We used ‘over-deconvolution’ once: in the first paragraph on page 4 in the original submission, we stated that “RLD also produced ‘over-deconvolved’ reconstructions relative to RLN and RLN-a”. This refers to **Supplementary Fig. 6**. In this case, RLD produced noisy reconstructions with artificially thin structures in the lateral view (i.e., thinner than the ground truth), while completely failing to recover fine structure in the axial views – a phenotype that often occurs if deconvolution is run too long (thus: ‘over-deconvolved’). We agree that this is jargon and, to avoid any confusion, we have removed this sentence in the revision.

4. In the course of characterizing different strengths, the manuscript presents an enormous number of images. While including all of the images demonstrates impressive rigor, the number of “uncompelling” images detracts from the main points of the manuscript. The authors should consider reducing the number of images to those that best demonstrate the salient points, and provide better explanations of the images in the legends and text.

Re: We appreciate the suggestion – but without reviewer or editorial feedback on precisely what constitutes ‘uncompelling’, we are reluctant to cut any data. We also note that both this and the other reviewers have asked us for more data and results - not less. As noted above, we agree that our legends could be improved, particularly in the context of clarifying what is meant by ‘ground truth’. In the revision, we expanded **Supplementary Table 1** to clearly delineate what “ground truth” was used in training for every result presented in the paper and added a text description in Methods (page 26, lines 729-735).

Reviewer #3:

Remarks to the Author:

The authors present a deep learning based solution for deconvolution, which falls under the scope of physics-based/physics-aided artificial intelligence. They show extensive results on a variety of samples and good generalizability across samples and microscopes. Supplementary figures are vital for appreciating the advantage on a variety parameters such as noise, etc.

The scope of creating impact through physics-based artificial intelligence is well-tapped here. Therefore, even though the idea of using deep learning for any image translation problem seems a trivial trend today, the approach of including physics in the neural network is indeed interesting, even if not categorically novel. The authors themselves, unsurprisingly, refrain from claiming strong novelty and instead focus on the impact of the method (less number of parameters, generalizability, etc.). The extent of results provided, both synthetic and experimental, is impressive to convey it is a large and painstakingly created body of work.

Re: We appreciate the authors' summary of our work.

However, despite all the positive qualities mentioned here, I am afraid that I have major concerns about the impact of the work matching the expectations of *Nature Methods*. I present the scientific concerns below in a numbered list. After that, I present other concerns.

Re: We hope that our responses below help to convince this reviewer that our paper now matches the expectation of *Nature Methods*.

1. The authors used RLD deconvolution (if I understood correctly) as the ground truth for training, except in the case of synthetic datasets. Therefore, this method is only as good as RLD deconvolution for experimental datasets. The authors have not provided justification or insight into why this was used as the ground truth for the experimental data. Could authors have explored other ground truth mechanisms? Or, maybe the objective is to perform Richardson Lucy Deconvolution without using it and not necessarily to obtain sharper or more optically cleared images. If this is the case, there is not a flaw in the objective except a significantly limited impact. It also conflicts with the authors claiming that RLN performs better than RLD for synthetic datasets. On the other hand, it is commendable that authors have explored using synthetic datasets for training, but both synthetic and experimental datasets for testing. Why not then use simulation/synthetic data-supervision as the core solution, in order to be not limited by the lack of ground truth in the experimental data?

Re: We appreciate the reviewer's comments and questions. Yes, in our submitted manuscript, in many cases we used RLD of high SNR raw data as the ground truth for training. As RLN incorporates the image formation process into the neural network and is motivated directly by RLD with unmatched back projectors, an important question is whether the network result can accurately reproduce the RLD result on high SNR data (this is usually the regime in which RLD excels). As shown in **Fig. 3** (ground truth: high SNR deconvolved iSIM data; input data: raw iSIM data), clearly RLN can reproduce the RLD result with high fidelity. At the same time, RLN can produce even better results than RLD on single-view input when using dual-view deconvolved data or synthetic data as ground truth (e.g., **Fig. 1b-g, Fig. 2, Fig. 5**), particularly in axial views. Presumably this is because of the isotropic resolution of the ground truth in these cases. Partially in response to this reviewer's comments, we have now additionally gathered super-resolution ground truth data (e.g., STED microscopy and 3D SIM, Figure 8 and 9 in this letter, **Supplementary Figs.**

17, 18 in the revision), which is presumably considerably closer to the object structure than deconvolved data – again finding that when using this ground truth, RLN outperforms RLD. In our revision, we have also expanded **Supplementary Table 1** to make it very clear what ‘ground truth’ we used to train the network for every dataset presented in the paper.

Regarding the question about the use of synthetic datasets as the core solution – on the one hand, RLN does indeed generalize very well (e.g., on the widefield volumes presented in **Figs. 4, 5, Supplementary Figs. 14, 15**), often producing apparently better reconstructions than RLD even when there is no experimental ground truth available. On the other hand, we often found that RLN performed better under a conventional testing regime (training on data similar to the input) vs. generalization testing (training using dissimilar data): (1) in **Supplementary Fig. 7**, we tested RLN performance on simulated bead samples, comparing RLD, conventional testing, and generalization, and we found that “the generalization result slightly distorts and sharpens bead shapes compared to the ground truth and conventional testing result”; (2) in **Supplementary Fig. 11**, we compared the conventional testing (trained on experimental data) and generalization (trained on synthetic data) on images of *C. elegans* embryos acquired with diSPIM, and quantitative assessments show that the conventional testing results (SSIM-membrane: 0.80, SSIM-nuclei: 0.85) is closer to the dual-view joint deconvolution ground truth than the generalization results (SSIM-membrane: 0.75, SSIM-nuclei: 0.76). Based on these findings, we suggest the use of synthetic data for establishing training models only when it is challenging to obtain high quality experimental ground truth.

2. No benchmarking beyond simulation results, RLD, and data driven methods. We do not know what artifacts RLN produces, which ones are attributed to the quality of ground truth, which ones to the architecture, and which ones to the learning process. We also do not see a discussion on the limitation of the technique or failure analysis in order to understand the scope in which the method creates utility. Experimental benchmarking using methods that support super-resolution like structured illumination microscopy and/or methods known as gold standard for benchmarking such as SEM is needed, even if for thin samples.

Re: We appreciate this suggestion and agree that additional experimental benchmarking would strengthen our paper, particularly with structured illumination or other super-resolution ground truth. To address this point, and taking the direct suggestion of this reviewer, we have (1) restored confocal data (U2OS cells stained with a primary antibody against Tomm20 and an anti-rabbit secondary antibody conjugated with Alexa Fluor 594, marking mitochondria) using RLN trained on STED ground truth (**Figure 8, Supplementary Fig. 17** in the revision); (2) restored widefield data (Jurkat cells expressing EMTB-3XGFP, a microtubule marker; and U2OS cells expressing Lamp1-EGFP, a lysosomal marker) using RLN trained on 3D SIM ground truth (**Figure 9, Supplementary Fig. 18** in the revision). In both cases, RLN improves resolution, but not to the extent of the ground truth. For example, some dim microtubule filaments evident in the 3D SIM ground truth are not recovered in RLN (e.g., magnified insets shown in **Figure 8a, Supplementary Fig. 18a** in the revision). This is unsurprising, given that the ground truth contains fine

features that are likely not easily recoverable given single diffraction-limited input images, and that RLN was designed primarily to deconvolve, not super-resolve, input data. Another failure case – as highlighted in our manuscript in **Supplementary Fig. 16** and again unsurprising – arises when attempting to process extremely low SNR input. We highlight this point again in Figure 11 (also in **Supplementary Fig. 16a** in the revision). To address the challenging case of noisy input at least partially, in our manuscript we investigated a two-step training procedure (**Supplementary Fig. 16d, e**), which outperforms the single RLN step. In the revision, we have emphasized both the ‘resolution’ and ‘SNR’ limitations in the Discussion section of our paper (page 10, line 364-367): “Like any denoising method, RLN’s performance degrades if presented with ultralow SNR input data (**Supplementary Fig. 16a, b**), although a multistep network approach may help (**Supplementary Fig. 16d**). Also, although RLN can provide some resolution enhancement, it cannot restore fine detail to the extent present in the super-resolution ground truth (**Supplementary Figs. 17, 18**).”

Figure 11. (**Supplementary Fig. 16a** in the revision) Another example of the failure of RLN when challenged with very noisy input. Scalebar: 10 pixels.

3. The recently released Thunder Imaging system of Leica also claims superior deconvolution using deep learning (in fact better deconvolution or optical clearance in comparison to conventional RLD, if I understood this correctly). How does RLN compare with Thunder Imaging in terms of quality of results and/or approach?

Re: We thank the reviewer for bringing up the Leica Thunder method as an example of a state-of-the-art deconvolution method that might be worth benchmarking our technique against. We would, however, like to respectfully correct the reviewer on one point. The Thunder software is not based on deep learning methods, as is clear from the technical note released by Leica describing Thunder (<https://7157e75ac0509b6a8f5c-5b19c577d01b9ccfe75d2f9e4b17ab55.ssl.cf1.rackcdn.com/EQZZKYFC-PDF-1-450123-4426040265.pdf>) and as we have additionally verified directly with Leica engineers. Thunder is designed for one purpose: to deblur widefield volumes. Also, as the Thunder software only supports the Leica file format (.lif) and there is no way to convert other image formats (e.g., the

commonly-used tif format) to a (.lif) for processing with Thunder, it is unfortunately much less of a 'general purpose' tool than one might be led to believe.

Nevertheless, to address this reviewer's question we acquired data on two new samples (fixed U2OS cells stained with Alexa Fluor 568 Phalloidin, marking actin; and fixed Cos-7 cells immunolabeled with a primary mouse anti-Nup clone Mab414 and goat-anti-mouse IgG secondary antibody conjugated with Star635P, marking nuclear pore complexes), first with a state-of-the-art Leica widefield microscope (LAS X, DM18, 63x/1.40 OIL UV) for Thunder processing; then (on the same samples) with Leica confocal microscopy (HC PL APO CS2 63x/1.40 OIL), to provide ground truth. We then compared the raw widefield input data, RLD on the widefield input, the Thunder results, the RLN prediction (using synthetic mixed structures for training), and the registered confocal data as a ground truth reference (Figure 12, **Fig. 4** in the revision). As shown, RLD, Thunder, and RLN all improve effective contrast and resolution when compared to the raw widefield input data. In some XY planes (e.g., Figure 12a, **Fig. 4a** in the revision), the deblurring ability of Thunder and RLN appear visually similar. However, RLD and Thunder both produce obvious artifacts in the axial views (Figure 12a, **Fig. 4a** in the revision) that are suppressed in RLN. Additionally, the Thunder imaging system also introduces artifacts in the NPC images (Figure 12b, **Fig. 4b** in the revision) that are absent in RLN. RLN results are obviously visually closest to the confocal results, an impression confirmed with quantitative PSNR and SSIM analysis (see the caption in Figure 12). To summarize, while Thunder may be reasonably suited to deblurring widefield images acquired on Leica microscopes, RLN provides obviously superior results and is clearly far more versatile, as it can be immediately applied to data acquired on a far greater number of microscope types (light-sheet, SIM, confocal) and on microscopes from any vendor.

In sum, in lines 277-291 on page 8 in the revised manuscript, we have added the discussion of the new results: "....., we examined multiple samples imaged with widefield microscopy. First, we evaluated the generalization ability of RLN trained on purely synthetic data (**Supplementary Fig. 2**) to widefield images of fixed U2OS cells stained with Alexa Fluor 568 Phalloidin, marking actin; and fixed Cos-7 cells immunolabeled with a primary mouse anti-Nup clone Mab414 and goat-anti-mouse IgG secondary antibody conjugated with Star635P, marking nuclear pore complexes (NPC). We then compared the RLN predictions to the widefield input data, RLD on the widefield input, the Leica Thunder computational clearing method (a state-of-the-art commercial deconvolution software package), and confocal images of the same structures to provide ground truth (**Fig. 4**). As shown, RLD, Thunder, and RLN all improve effective contrast and resolution when compared to the raw widefield input data. In some XY planes in the actin data (e.g., **Fig. 4a**), RLD, Thunder and RLN provide visually similar output. However, RLD and Thunder both produce obvious artifacts in the axial views (**Fig. 4a**) that are suppressed in RLN. Additionally, the Thunder imaging system also introduces artifacts in the NPC images (**Fig. 4b-d**) that are absent in RLN. The RLN output was visually closest to the confocal data, an impression confirmed with quantitative PSNR and SSIM analysis (**Supplementary Table 2**).” In lines 889-895 on page 30 in the revised manuscript, we have added a few sentences to describe how we acquired and processed the data: “ Wide-field fixed U2OS and Cos-7 cell images (**Fig. 4**) were acquired with a Leica widefield microscope (LAS X, DM18, 63x/1.40 OIL UV, 102 nm pixel size) and processed with the Leica Thunder computational

clearing method (commercial deconvolution software designed for deblurring widefield volumes, using the small volume computational clearing method with default settings, Strategy: Adaptive; Thunder Strength: 60; Thunder Regularization 5.05×10^6), then the same samples were acquired with Leica confocal microscopy (HC PL APO CS2 63x/1.40 OIL, a pixel size of 102 nm). The same field of view of widefield- and confocal data were found manually, then finely registered with an affine transformation⁸.”

Figure 12. (Fig. 4 in the revision) Comparisons between RLN, RLD and Leica Thunder Imaging system. a) Lateral and axial planes from images of a fixed U2OS cells stained with Alexa Fluor 568 Phalloidin, comparing widefield raw data, RLD with 100 iterations, the Thunder output, the RLN result, and the registered confocal data as a ground-truth reference. RLN was trained with synthetic mixed structures.

RLN predictions show better restoration than RLD and Thunder, particularly along the axial dimension. PSNR and SSIM analysis using the confocal data as ground truth confirm this result (Raw widefield: SSIM 0.57 ± 0.03 , PSNR 28.7 ± 0.9 ; Richardson-Lucy deconvolution: SSIM 0.67 ± 0.03 , PSNR 30.0 ± 0.7 ; Thunder: SSIM 0.63 ± 0.05 , PSNR 30.0 ± 0.7 ; RLN: SSIM 0.73 ± 0.02 , PSNR 30.9 ± 0.9 , N = 4 volumes). b) Lateral and axial planes of nuclei pore complexes in a fixed Cos-7 cell immunolabeled with primary mouse anti-Nup clone Mab414 and goat-anti-mouse IgG secondary antibody conjugated with Star635P, comparing widefield input, RLD with 100 iterations, Thunder output, RLN prediction, and registered confocal data as a ground-truth reference. c) Magnified views of the blue rectangle in b). d) Line profiles across the red and magenta lines in the lateral and axial views in the left-most panels of b). RLN was trained with synthetic mixed structures. Both visual analysis (e.g., red arrows) and line intensity profiles demonstrate that RLN restoration outperform Thunder (obvious artifacts indicated by orange arrows) and RLD in both lateral and axial view, with the RLN restoration approaching the confocal reference. PSNR and SSIM analysis using the registered confocal results as ground truth confirm this result (Raw widefield input: SSIM 0.78 ± 0.04 , PSNR 34.5 ± 0.4 ; Richardson-Lucy deconvolution: SSIM 0.79 ± 0.02 , PSNR 36.7 ± 0.5 ; Thunder: SSIM 0.80 ± 0.04 , PSNR 36.7 ± 0.4 ; RLN: SSIM 0.86 ± 0.01 , PSNR 37.5 ± 0.3 , N = 4 volumes). Scalebar: a) $10 \mu\text{m}$; b) $10 \mu\text{m}$; c) $3 \mu\text{m}$.

4. The authors discuss a lot about interpretability in the introduction, but do not address the interpretability of the method beyond using algorithm unrolling. Indeed the outputs of RLN FP1, BP1, FP2, BP2 are given in multiple figures and discussed in one paragraph on page 3, it is not a sufficient insight into interpretability. It would be more useful to explain how learning progresses through training, and what gets learnt or unlearnt in the process, how dependent it is on the data used for training, and why it generalizes over samples and imaging systems.

Re: We appreciate this suggestion and agree that better ‘interpretability’ would be desirable. Interpretability refers to the extent of a human’s ability to understand the model, but it appears quite difficult to reach a consensus on the exact meaning of the term [5, 6]. For example, some researchers explore post-hoc explanations for models, while others try to explore the interplay between the internal components / machinery of a model. RLN was motivated by, and is based on, Richardson-lucy deconvolution with unmatched backprojectors [3]. As the convolution operation plays key roles in both classic deconvolution algorithms and convolutional neural networks (CNN), RLN uses the convolutional layers of a CNN to replace the traditional convolution operation, solving the unmatched forward/backward projector design problem by combining data-driven training with an architecture that mimics the underlying RLD model. From this starting point, RLN can be naturally interpreted as a projector-design algorithm and the role of different convolutional layers in RLN also can be explained. This level of ‘interpretability’ certainly meets the bar presented in the two papers [1, 2] mentioned by Reviewer #1. From the feature analysis perspective, we showed the internal features of layers (**Fig. 1b** in the main text and **Supplementary Figs. 3-5**) and provided qualitative insights on what kinds of features were

learned by RLN. To further investigate 'how learning progresses through training', we have now added all the intermediate RLN outputs and linked them to specific steps in RLD (Figure 13). Although the H3 module represents purely 'data-driven' learning, H1 and H2 modules display both data-driven learning and behavior characteristic of RLD. As shown in Figure 13, the FP submodule provides gradually smoother features, the DV step enhances dim signals (yellow arrows), the final results of BP provide the update factor and the resolution is enhanced after the update step. We acknowledge that it is difficult to explain what the first few layers of BP (images with red borders) are doing, perhaps because there is no obvious link to RLD, but it appears that the last layers of BP in both H1 and H2 provide update factors that enhance contrast for fine features (e.g., edges).

Regarding the reviewer's comment about "why it generalizes over samples and imaging systems", there is a large effort to understand why neural networks do or do not generalize, yet there still does not appear to be a satisfactory explanation [7, 8]. Thus, we admit it is hard for us to explain how/why RLN generalizes so well, but also feel that explaining this mystery is a research problem unto itself and in our view goes far beyond the scope of this initial paper. The following points may offer some insight: (1) although the network does not directly learn the PSF kernel used in forward and backward projection steps, the summed effect of the convolutional layers mimic an effective PSF analogous to that used in RLD, perhaps explaining the robustness to different types of data (as shown in **Figs. 3-5** in the main text and **Supplementary Figs. 5, 14, 15, 17, 18**); (2) the RLD formula embedded in the network structure acts to regularize training, helping to guide non-content-based feature learning; (3) the number of learning parameters in RLN is much less than in CARE and RCAN, which might reduce over-fitting. We have added this interpretability and generalization discussion as **Supplementary Note 2** in the revision.

Figure 13. Further links between RLN and RLD. a) The decomposition of RLD and the function of each step, illustrated with synthetic mixed structures. b) Comparisons between RLD and RLN at each intermediate output. Five iterations of RLD are shown (top row), compared with all intermediate feature maps generated by RLN (other rows), to illustrate where RLN does and does not agree with RLD. H1 and H2 display both data-driven behavior and more interpretable output analogous to RLD. The FP submodule provides gradually smoother features, the DV step enhances dim signals (yellow arrows), the final results of BP provide the update factor and the resolution is enhanced after the update step. The first few layers of BP are not easy to understand because of the learning-based characteristics of convolutional layers (indicated by the red borders), but the last layer of BP provides an update factor that enhances contrast for fine features. H3 part is based purely on data-driven learning and is used to merge the output of H1 and H2 part. Scalebar: 10 pixels.

[1] Deep-URL: A Model-Aware Approach to Blind Deconvolution Based on Deep Unfolded Richardson-Lucy Network, Agarwal et al., 2020

[2] Deep Unfolding Network for Image Super-Resolution, Zhang et al., 2020

[3] Guo, M. et al. Rapid image deconvolution and multiview fusion for optical microscopy. *Nature Biotechnology* 38, 1337-1346 (2020).

[4] Wu, Y. et al. Reflective imaging improves spatiotemporal resolution and collection efficiency in light sheet microscopy. *Nat. Commun.* 8, 1452 (2017).

- [5] F. -L. Fan, J. Xiong, M. Li and G. Wang, "On Interpretability of Artificial Neural Networks: A Survey," in IEEE Transactions on Radiation and Plasma Medical Sciences, vol. 5, no. 6, pp. 741-760, Nov. 2021, doi: 10.1109/TRPMS.2021.3066428.
- [6] Y. Zhang, P. Tiño, A. Leonardis and K. Tang, "A Survey on Neural Network Interpretability," in IEEE Transactions on Emerging Topics in Computational Intelligence, vol. 5, no. 5, pp. 726-742, Oct. 2021, doi: 10.1109/TETCI.2021.3100641.
- [7] Chatterjee, S., & Zielinski, P. (2022). On the generalization mystery in deep learning. <https://arxiv.org/pdf/2203.10036v1.pdf>
- [8] Zhang, C.Y., et al., Understanding Deep Learning (Still) Requires Rethinking Generalization. Communications of the Acm, 2021. 64(3): p. 107-115.

5. A minor comment on readability. This manuscript is replete with acronyms, right from the abstract. Quite a few acronyms are so close to each other, the readability is compromised in discerning one acronym from the other and recalling which one meant what.

Re: The abstract contains a single acronym, RLN. While we appreciate the reviewer's point and are sympathetic – it is not our fault that every new method has a new acronym. We follow the standard in this field, which is to define the full name once and then refer to it by its acronym.

6. While deconvolution is an important topic, especially for thick samples, I believe that the scope of the work is quite narrow, the novelty limited, and the validity and reliability needing more work. The robustness is demonstrated and the principle behind the work is interesting. I do acknowledge the extensive efforts of the authors, but I feel that this work is more suited to other avenues, which are looking for new ways of making deep learning more powerful and application specific. My own suggestions would be Nature Machine Intelligence and/or IEEE TMI/TCl/TIP.

Re: We hope the additional data and testing we have performed have helped to convince this reviewer of the broad and useful applicability of RLN on diverse microscopes and samples, and that the paper is indeed well-suited to *Nature Methods*.

Decision Letter, first revision:

Dear Yicong,

Thank you for submitting your revised manuscript "Incorporating the image formation process into deep learning improves network performance" (NMETH-A48597B). It has now been seen by the original referees and their comments are below. The reviewers find that the paper has improved in revision, and therefore we'll be happy in principle to publish it in Nature Methods, pending minor revisions to satisfy the referees' final requests and to comply with our editorial and formatting guidelines.

Please have a look at the remaining comments of reviewer 3 and let me know how you might update the manuscript.

TRANSPARENT PEER REVIEW

Thank you again for your interest in Nature Methods Please do not hesitate to contact me if you have any questions.

Sincerely,
Rita

Rita Strack, Ph.D.
Senior Editor
Nature Methods

ORCID

Reviewer #1 (Remarks to the Author):

The authors have responded to the reviewers' concerns in great and exhausting detail. The quality of the manuscript has increased substantially in the revised version, and I would definitely rate it as ready for publication. The rigorousness of the authors' analysis is very impressive.

However, my main concern remains: I do not think this manuscript to be sufficiently novel and impactful for the targeted journal Nature Methods. I would rather expect to read a manuscript of this sort in one of the more technical journals, such as IEEE TCI, TIP, TMI, or Scientific Reports.

Reviewer #2 (Remarks to the Author):

In this manuscript Li et al. demonstrate a novel approach to image deconvolution in which a convolutional neural network is applied to the forward and backward projection processes used in classical iterative deconvolution. The authors present a very rigorous, thoroughly documented case for the superiority of the "Richardson-Lucy Network" (RLN) over conventional Richardson-Lucy Deconvolution (RLD), with respect resolution, signal linearity and generality.

In my original review, I questioned several aspects of the approach, in particular the overall advantages over existing, which I saw as incremental. The authors have done an outstanding job of thoroughly addressing each of my questions and criticisms. The revised manuscript includes new data and clearer explanations that better exemplify and delineate the advantages of the RLN technique. While the differences in performance are often visually subtle, in each case the RLN system provides results that are quantitatively superior. To the degree that these differences are not compelling to potential users, I believe that the major advantages of the system are that (1) it is effective in 3D images, (2) is faster than other comparable techniques (sometimes profoundly so), (3) eliminates the need for parameter tuning, preventing user error/manipulation and, perhaps most significantly, (4) copes well with spatially variant point-spread functions, a characteristic of many, if not most image volumes. The revised manuscript describes a novel process that is rigorously characterized, likely to be widely interesting and useful to a broad range of investigators.

Reviewer #3 (Remarks to the Author):

I commend the authors for the extensive rebuttal. I do see that authors have put efforts in showing

some empirical support regarding the scientific concerns raised by me. I do not see any crisp insight except some indicative results for points 1 and 4. Regarding point 1, the authors argue that although the results of model trained using simulated data only does generalize well, they also argue that quantitative metrics when using conventional (not generalized) setting are better. And I agree with both the arguments. However, it appears that the advantage in terms of quantitative metrics is just slightly better and one may argue that the cost of generating experimental data of required quality can be quite expensive. Further, it is arguable that maybe the simulated dataset could be better designed for the purpose resulting in much better results, for example through one or more of the following:

- (a) the target images in simulated training dataset need not be just the original phantom with sharp boundaries, but they can be slightly blurred images (for example through a narrower PSF than the microscope) which will take the target image space of simulated datasets closer to the experimental ones (including similar to iSIM, STED, etc.)
- (b) Training using input simulated images with a wide range of SNR or extremely low SNR may help
- (c) More complex simulated phantoms may be used (brain phantom is sufficiently complicated, but spheres etc. are quite simple) to better replicate the real structures.

While I appreciate the extended discussion on interpretability in Supplementary Note 2. However, I suggest reducing too much emphasis in the main manuscript the mention of it. I still think that the question of interpretability is dealt with in a shallow manner and experts on interpretability of deep learning may widely disagree. I suppose 'interpretability' is better replaced by 'insight into specific components of RLN and their connection to the physics of the problem of deconvolution'.

Having commented on the replies of scientific comment, I still believe that the paper does not match the following requirements of Nature Methods "The potential significance of the results — whether these results will be important to the field and advance understanding in a way that will move the field forward."

The work does not advance understanding in the field and does not necessarily move the field forward except making deconvolution process faster (albeit after enough effort has been put in generating the training dataset, training the model, and testing it rigorously). I therefore am still not convinced that this work is a match to the expectations of Nature Methods.

Author Rebuttal, first revision:

Dear Reviewers,

Thank you for your helpful feedback and comments. Below is our point-by-point response to the issues you raised, followed by the **Re:** callouts and highlighted the changes with **dark red**. In the revised manuscript, we also mark the changes with **dark red**.

Thank you for reconsidering our revised manuscript.

Sincerely,
Yicong Wu and Hari Shroff (on behalf of all the authors)
NIBIB/NIH

Reviewer #1:

Remarks to the Author:

The authors have responded to the reviewers' concerns in great and exhausting detail. The quality of the manuscript has increased substantially in the revised version, and I would definitely rate it as ready for publication. The rigorousness of the authors' analysis is very impressive.

Re: We are glad that the reviewer recognizes that the quality of our revised paper is significantly improved and it is ready for publication.

However, my main concern remains: I do not think this manuscript to be sufficiently novel and impactful for the targeted journal Nature Methods. I would rather expect to read a manuscript of this sort in one of the more technical journals, such as IEEE TCI, TIP, TMI, or Scientific Reports.

Re: We disagree that the manuscript is not sufficiently novel. Our novel contributions are well summarized by Review #2. We think our revised manuscript now matches the expectation of *Nature Methods*.

Reviewer #2:

Remarks to the Author:

In this manuscript Li et al. demonstrate a novel approach to image deconvolution in which a convolutional neural network is applied to the forward and backward projection processes used in classical iterative deconvolution. The authors present a very rigorous, thoroughly documented case for the superiority of the "Richardson-Lucy Network" (RLN) over conventional Richardson-Lucy Deconvolution (RLD), with respect resolution, signal linearity and generality.

In my original review, I questioned several aspects of the approach, in particular the overall advantages over existing, which I saw as incremental. The authors have done an outstanding job of thoroughly addressing each of my questions and criticisms. The revised manuscript includes new data and clearer explanations that better exemplify and delineate the advantages of the RLN technique. While the differences in performance are often visually subtle, in each case the RLN system provides results that are quantitatively superior. To the degree that these differences are not compelling to potential users, I believe that the major advantages of the system are that (1) it is effective in 3D images, (2) is faster than

other comparable techniques (sometimes profoundly so), (3) eliminates the need for parameter tuning, preventing user error/manipulation and, perhaps most significantly, (4) copes well with spatially variant point-spread functions, a characteristic of many, if not most image volumes. The revised manuscript describes a novel process that is rigorously characterized, likely to be widely interesting and useful to a broad range of investigators.

Re: We thank the reviewer for the summary of our novel contributions and are glad that they found the revised paper addressed the concerns raised in the previous review.

Reviewer #3:

Remarks to the Author:

I commend the authors for the extensive rebuttal. I do see that authors have put efforts in showing some empirical support regarding the scientific concerns raised by me. I do not see any crisp insight except some indicative results for points 1 and 4. Regarding point 1, the authors argue that although the results of model trained using simulated data only does generalize well, they also argue that quantitative metrics when using conventional (not generalized) setting are better. And I agree with both the arguments. However, it appears that the advantage in terms of quantitative metrics is just slightly better and one may argue that the cost of generating experimental data of required quality can be quite expensive. Further, it is arguable that maybe the simulated dataset could be better designed for the purpose resulting in much better results, for example through one or more of the following:

(a) the target images in simulated training dataset need not be just the original phantom with sharp boundaries, but they can be slightly blurred images (for example through a narrower PSF than the microscope) which will take the target image space of simulated datasets closer to the experimental ones (including similar to iSIM, STED, etc.)

(b) Training using input simulated images with a wide range of SNR or extremely low SNR may help

(c) More complex simulated phantoms may be used (brain phantom is sufficiently complicated, but spheres etc. are quite simple) to better replicate the real structures.

Re: We thank the reviewer for the positive feedback and appreciate the suggestion about how the simulated dataset could be better designed. We agree that this is an important issue to discuss. In the revision (line 364-366, pages 10, second paragraph in Discussion section), we have addressed this point, "Further modification of the synthetic data would likely improve performance, perhaps by using a blurring kernel or noise level closer to the experimental test data or by incorporating more complex phantoms that better resemble real biological structures."

While I appreciate the extended discussion on interpretability in Supplementary Note 2. However, I suggest reducing too much emphasis in the main manuscript the mention of it. I still think that the question of interpretability is dealt with in a shallow manner and experts on interpretability of deep learning may widely disagree. I suppose 'interpretability' is better replaced by 'insight into specific components of RLN and their connection to the physics of the problem of deconvolution'.

Re: We agree with the reviewer's suggestion. In the revision, we have reduced the number of times we mentioned 'interpretability'. Specifically, (1) in the abstract, we changed the sentence "...., improving network interpretability and robustness" to "**....., establishing a connection to the image formation process and thereby improving network performance**", (2) in line 106, we changed "To demonstrate the interpretability of RLN" to "**To provide further insight into the connection between RLN and traditional deconvolution**"; (3) in line 342, we deleted the term 'interpretability' so it reads as "**thereby improving network performance**"; (4) in the caption of Supplementary Fig. 3 (previous Supplementary Fig. 5), we changed the sentence "Interpretability and generalization of RLN as assessed on human brain phantom." to "**Insight into the intermediate stages of RLN and generalization of RLN as assessed on human brain phantom.**", and the sentence "... indicates interpretability of RLN" to "**... indicates connections between RLN and RLD**".

Having commented on the replies of scientific comment, I still believe that the paper does not match the following requirements of Nature Methods "The potential significance of the results — whether these results will be important to the field and advance understanding in a way that will move the field forward." The work does not advance understanding in the field and does not necessarily move the field forward except making deconvolution process faster (albeit after enough effort has been put in generating the training dataset, training the model, and testing it rigorously). I therefore am still not convinced that this work is a match to the expectations of Nature Methods.

Re: After substantially addressing the reviewers' previous concerns, we do not agree that our innovation is solely in making the deconvolution process faster. We have shown more general applicability of RLN beyond deconvolution: (1) RLN outperforms direct deconvolution and Thunder on widefield volumes; (2) RLN provides class-leading performance when using super-resolved ground truth (like STED and 3D SIM data) that is closer to the object structure than deconvolved raw data. Now we think our revised manuscript matches the expectations of *Nature Methods*.

Final Decision Letter:

Dear Yicong,

I am pleased to inform you that your Article, "Incorporating the image formation process into deep learning improves network performance", has now been accepted for publication in Nature Methods. Your paper is tentatively scheduled for publication in our November print issue, and will be published online prior to that. The received and accepted dates will be March 7, 2022 and September 16, 2022. This note is intended to let you know what to expect from us over the next month or so, and to let you know where to address any further questions.

Over the next few weeks, your paper will be copyedited to ensure that it conforms to Nature Methods style. Once your paper is typeset, you will receive an email with a link to choose the appropriate

publishing options for your paper and our Author Services team will be in touch regarding any additional information that may be required.

Your paper will now be copyedited to ensure that it conforms to Nature Methods style. Once proofs are generated, they will be sent to you electronically and you will be asked to send a corrected version within 24 hours. It is extremely important that you let us know now whether you will be difficult to contact over the next month. If this is the case, we ask that you send us the contact information (email, phone and fax) of someone who will be able to check the proofs and deal with any last-minute problems.

If, when you receive your proof, you cannot meet the deadline, please inform us at rjsproduction@springernature.com immediately.

Once your manuscript is typeset and you have completed the appropriate grant of rights, you will receive a link to your electronic proof via email with a request to make any corrections within 48 hours. If, when you receive your proof, you cannot meet this deadline, please inform us at rjsproduction@springernature.com immediately.

Once your paper has been scheduled for online publication, the Nature press office will be in touch to confirm the details.

Content is published online weekly on Mondays and Thursdays, and the embargo is set at 16:00 London time (GMT)/11:00 am US Eastern time (EST) on the day of publication. If you need to know the exact publication date or when the news embargo will be lifted, please contact our press office after you have submitted your proof corrections. Now is the time to inform your Public Relations or Press Office about your paper, as they might be interested in promoting its publication. This will allow them time to prepare an accurate and satisfactory press release. Include your manuscript tracking number NMETH-A48597C and the name of the journal, which they will need when they contact our office.

About one week before your paper is published online, we shall be distributing a press release to news organizations worldwide, which may include details of your work. We are happy for your institution or funding agency to prepare its own press release, but it must mention the embargo date and Nature Methods. Our Press Office will contact you closer to the time of publication, but if you or your Press Office have any inquiries in the meantime, please contact press@nature.com.

If you are active on Twitter, please e-mail me your and your coauthors' Twitter handles so that we may tag you when the paper is published.

Please note that Nature Methods is a Transformative Journal (TJ). Authors may publish their research with us through the traditional subscription access route or make their paper immediately open access through payment of an article-processing charge (APC). Authors will not be required to make a final decision about access to their article until it has been accepted. Find out more about Transformative Journals

Authors may need to take specific actions to achieve compliance with funder and institutional open access mandates. If your research is supported by a funder that requires immediate open access (e.g. according to Plan S principles) then you should select the gold OA route, and we will direct you to the compliant route where possible. For authors selecting the subscription publication route, the journal's standard licensing terms will need to be accepted, including self-archiving policies. Those licensing terms will supersede any other terms that the author or any third party may assert apply to any version of the manuscript.

To assist our authors in disseminating their research to the broader community, our SharedIt initiative provides you with a unique shareable link that will allow anyone (with or without a subscription) to read the published article. Recipients of the link with a subscription will also be able to download and print the PDF. As soon as your article is published, you will receive an automated email with your shareable link.

Please note that you and your coauthors may order reprints and single copies of the issue containing your article through Nature Portfolio's reprint website, which is located at <http://www.nature.com/reprints/author-reprints.html>. If there are any questions about reprints please send an email to author-reprints@nature.com and someone will assist you.

Best regards,
Rita